# Uropathogenic *Escherichia coli* invade luminal prostate cells via FimH–PPAP receptor binding

Maria Guedes[1,7], Simon Peters [1,7], Amruta Joshi[1], Sina Dorn[1], Janina Rieger [1], Kimberly Klapproth [1], Tristan Beste[2], Alexander M. Leipold[2,3], Mathias Rosenfeldt [4], Antoine-Emmanuel Saliba [2,3], Ulrich Dobrindt [5], Charis Kalogirou[6] & Carmen Aguilar [1]✉

Bacterial prostatitis caused by uropathogenic *Escherichia coli* (UPEC) strains is a highly prevalent and recurrent infection responsible for significant morbidity in men. The molecular pathogenesis of prostatitis remains poorly understood, partly due to a lack of suitable in vitro models. Here we developed a 2D mouse stem cell-derived prostate epithelial organoid model. In the organoid model, 5α-dihydrotestosterone promoted differentiation of basal into luminal cells, while transcriptomic analyses validated the model in comparison to 3D models and mouse prostate tissue. Infection analyses revealed that UPEC preferentially attached to, invaded and replicated within luminal prostate cells. Experiments with a UPEC mutant strain lacking the bacterial adhesin, FimH, alongside immunoprecipitation, mass spectrometry, biochemistry and infection experiments with host gene knockouts revealed that FimH–prostatic acid phosphatase (PPAP) binding interactions promote UPEC invasion of luminal prostate cells. D-Mannose competitively inhibited FimH–PPAP interactions. Findings were validated using ex vivo human prostate tissue. These data highlight the adaptability of FimH in engaging host receptors and the potential for FimH-targeting strategies to reduce bacterial prostatitis.

Urinary tract infections (UTIs) are among the most common bacterial infections worldwide, with over 400 million cases per year[1]. Despite adequate antibiotic treatments, these infections are highly recurrent, with over 50% of patients experiencing a recurrence within a year, highlighting limitations of current treatment options. While women have a higher incidence of UTIs, infections in men are harder to treat, often requiring prolonged and higher antibiotic doses[2,3]. A common complication in male UTIs is the spread of infection to the prostate gland[4].

Bacterial prostatitis affects nearly 1% of men worldwide, with a bimodal distribution among younger and older men[5]. Due to the proximity of the prostate to the bladder, shared infection route, common causative infective agents and overlapping risk factors, bacterial prostatitis is increasingly considered a form of UTI[6]. As in the bladder, *Escherichia coli* strains, predominantly uropathogenic *E. coli* (UPEC), are the leading cause of bacterial prostatitis[6–10]. However, little is known regarding *E. coli*'s pathogenesis in prostate tissue. In the bladder, UPEC uses type 1

[1]Host Pathways in Urinary Tract Infections Group, Institute of Molecular Infection Biology, University of Würzburg, Würzburg, Germany. [2]Helmholtz Institute for RNA-based Infection Research, Helmholtz-Centre for Infection Research, Würzburg, Germany. [3]Institute of Molecular Infection Biology, Faculty of Medicine, University of Würzburg, Würzburg, Germany. [4]Institute of Pathology, University of Würzburg, Würzburg, Germany. [5]Institute of Hygiene, University of Münster, Münster, Germany. [6]Department of Urology and Pediatric Urology, University Hospital Würzburg, Würzburg, Germany. [7]These authors contributed equally: Maria Guedes, Simon Peters. ✉e-mail: carmen.aguilar@uni-wuerzburg.de

pili with the FimH adhesin at their tips to bind to the membrane protein uroplakin 1a (UPK1A) on the superficial urothelial cells. After entering these cells, UPEC starts to replicate and forms intracellular bacterial communities (IBCs)[11,12]. Superficial cells containing IBCs are quickly exfoliated, exposing the underlying immature transitional epithelial cells[11]. FimH can also bind to α3β1 integrins on these cells and facilitate invasion to form quiescent intracellular reservoirs (QIRs)[13,14], which are responsible for antibiotic tolerance and recurrences. Understanding whether these infection characteristics in the bladder also occur in the prostate tissue is fundamental, as it influences the design and implementation of treatments.

The prostate is a male sex gland made up of ducts lined with pseudostratified epithelium composed of three main types of cell: secretory luminal cells expressing keratin (KRT) 18 and androgen receptors (AR); basal cells, characterized by their expression of KRT5, KRT14 and p63; and rare neuroendocrine cells, positive for chromogranin A (CHGA)[15].

Despite their clinical relevance, prostate infections remain substantially understudied, partly due to technical challenges in infecting in vivo mouse models and the lack of suitable in vitro models that reproduce prostatic epithelial heterogeneity. A few studies have examined prostate infections using mouse models, and to our knowledge, none have utilized primary cells from human or mouse tissues to investigate these infections. Collectively, mouse studies have shown that *E. coli*, either prostate isolates or cystitis isolates, can establish acute and chronic infection in the murine prostate gland, inducing a robust proinflammatory response, immune cell infiltration and hyperplasia in the tissue[16–23]. Interestingly, one study reported UPEC tropism for the prostate versus the bladder in C57BL/6 male mice[22], and found that, similar to the male bladder, the murine prostate fails to induce a protective immune response, leading to chronic prostatitis. However, how the bacteria interact with epithelial cells and whether they invade and/or replicate within these host cells remain unknown.

Over the past decade, organoids and organoid-based models have become powerful in vitro systems that replicate key physiological and functional characteristics of their corresponding organs. Derived from tissue-resident adult stem cells, they can differentiate into the various epithelial cell types found within the tissue, closely mimicking the structure, function and cellular complexity of the organ. Prostate organoids, in particular, are valuable tools for studying prostate development and cancer[24,25]. However, a suitable model for investigating host–pathogen interactions is still lacking.

Here we use organoids to develop a model of the prostate epithelium to study *E. coli* pathogenesis in the context of bacterial prostatitis. The model closely mimics the cellular heterogeneity of the epithelial compartment in the real organ. Using this model, we found that *E. coli* adheres to, invades and replicates within luminal prostate cells. This interaction was dependent on FimH binding to the herein-identified prostate-specific membrane receptor prostatic acid phosphatase (PPAP).

## Results

### An organoid-based model mimics epithelial heterogeneity in prostate

A suitable in vitro infection model should reflect the cellular complexity of the tissue it represents. In case of a prostate model, this should include the main cell types (that is, luminal gland and basal stem cells) present in the prostate epithelium. To set this up, we generated mouse prostate three-dimensional (3D) organoids from C57BL/6 wild-type (WT) mice following a previously published protocol[26]. These organoids were then seeded onto a two-dimensional (2D) surface to provide apical accessibility for subsequent *E. coli* infections (Fig. 1a), using the standard prostate organoid media previously described[26]. However, this condition did not achieve a very high degree of differentiation, as observed by the low number of differentiated luminal cells (CD24a⁺) and the high number of KRT5⁺ basal cells (Fig. 1b, 1 nM, middle, and

Extended Data Fig. 1). Given that activation of the androgen receptor pathway is the primary driver of prostate epithelium development and differentiation, we hypothesized that increasing androgen levels in the medium could improve cellular differentiation. A 10-fold increase of 5α-dihydrotestosterone (DHT) was sufficient to differentiate basal into luminal cells (Fig. 1b, 10 nM, right, and Extended Data Fig. 1). In contrast, removing DHT enriched the model for basal stem cells (KRT5⁺ cells; control condition; Fig. 1b, left, and Extended Data Fig. 1).

To obtain a more in-depth characterization of the model, we applied single-cell RNA sequencing (scRNA-seq) to the 2D model and 3D organoids grown in both culture conditions (control and 10 nM DHT). Altogether, we analysed 9,992 single-cell transcriptomes (Supplementary Table 1), and quality control indicated good dataset quality and sequencing depth in all samples (Supplementary Fig. 1). Unsupervised clustering identified three clusters in both 3D organoids and the 2D model, corresponding to basal cells (*Krt5*⁺, *Trp63*⁺, *Krt14*⁺), luminal cells (*Cd24a*⁺, *Krt8*⁺, *Krt18*⁺, *Psca*⁺) and highly proliferative basal cells (*Mki67*⁺, *Birc5*⁺; Fig. 1c–e, Extended Data Fig. 2 and Supplementary Table 2). However, we observed a much higher percentage of luminal cells in the 2D model, independent of the absence or presence of DHT. This indicates an overall higher degree of differentiation in this model, achieved just by seeding the 3D organoid cells onto a 2D surface (Fig. 1f and Supplementary Table 1). Although media supplementation with 10 nM DHT increased the number of luminal cells in both models, the effect was more prominent in 2D than in 3D (Fig. 1f and Supplementary Table 1). Moreover, the expression of luminal markers (*Cd24a*, *Krt8*, *Epcam*, *Cd113/Prom1*, *Krt18*) was higher in 2D compared with 3D (Fig. 1d and Extended Data Fig. 2c–f). Although luminal cells expressed characteristic luminal markers (that is, *Cd24a*, *Krt18*, *Epcam* and *Cd113/Prom1*), transcripts for other luminal markers such as *Pbsn* and *Nkx3.1* were not detected in either the 3D or 2D models. Importantly, a neighbourhood analysis using murine prostate tissue data[27,28] as reference showed that the 2D model grown with DHT had the highest transcriptional similarity at the single-cell level to primary tissue (Extended Data Fig. 3), and the proportion of luminal cells in this model also closely matched that in murine tissue[27] (Supplementary Table 1).

While scRNA-seq also showed increased transcription of cell polarization and epithelial barrier integrity markers in the 2D model under the DHT condition (for example, *Ocln*, Cdh1, *Tjp1-3* and claudins; Supplementary Fig. 2), these markers did not show a clear increase in cell polarization in 3D (Supplementary Fig. 3). Cell polarization upon differentiation with DHT in 2D was further confirmed by immunostaining of the zonula occludens 1 (ZO-1) protein and actin filaments at cell junctions (apical ring of actin filaments; Fig. 1g). In addition, the presence of DHT in the culture medium led to higher transepithelial electrical resistance (TEER) in the 2D model (Fig. 1h), indicating enhanced epithelial barrier function. Together, our data demonstrate that the 2D organoid-based model grown with 10 nM DHT mimics the epithelial compartment of the murine prostate.

### *E. coli* preferentially invades luminal prostate cells

Invasion and replication within host cells are crucial steps for many pathogenic bacteria, allowing them to evade immune clearance and antibiotics. For *E. coli*, different strategies have been reported: in the bladder, it invades and replicates in umbrella cells[12]; in the kidney, it forms extracellular biofilm-like communities[29]. Its fate in the prostate remains unknown. Therefore, we first aimed to characterize *E. coli*'s ability to infect prostate cells using the 2D model. Since most *E. coli* pathotypes isolated from bacterial prostatitis are UPEC strains[9,10], we used UTI89, one of the most common reference UPEC strains, originally isolated from a cystitis case[30].

To distinguish between extra- and intracellular bacteria, we generated a GFP-expressing UTI89 strain and stained extracellular bacteria with an anti-lipopolysaccharide (anti-LPS) antibody without cell permeabilization. This showed rapid UPEC invasion of the 2D model,

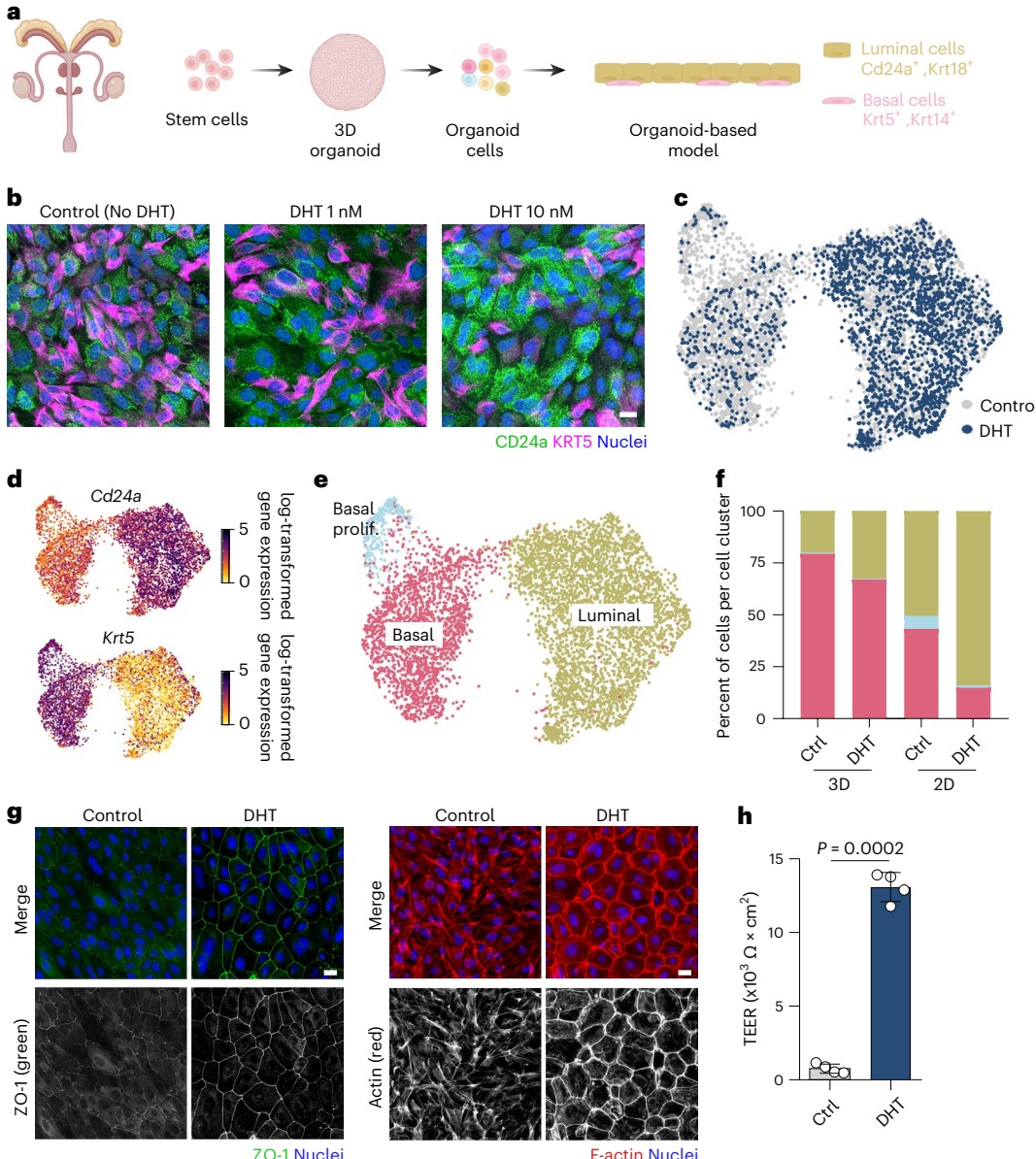

**Fig. 1 | An organoid-based model of the murine prostate reflects epithelial cell heterogeneity. a**, Scheme of the experimental setup. Prostate organoids were generated from murine prostate tissue and seeded in 2D. **b**, Confocal microscopy images of the organoid-based model grown in the presence or absence of DHT (1 or 10 nM) and stained for CD24a (green) and KRT5 (magenta) using immunofluorescence. Nuclei were counterstained using Hoechst 33342 (blue). Scale bar, 25 μm. *n* = 3 independent experiments, 3 organoid lines. **c**, Single-cell transcriptomes from the 2D organoid-based models grown in control (3,602 cells, grey) or DHT (10 nM, 2,127 cells, dark blue) medium were integrated and projected using UMAP. **d**, Expression of known markers specific for luminal prostate cells (*Cd24a*) and basal cells (*Krt5*) colour coded and projected on top of the UMAP displayed in **c**. **e**, Using known prostate marker genes for the specific prostate cell types, cells were clustered on the UMAP and assigned to their identities (see Methods and Extended Data Fig. 2). **f**, Bar plot depicting the cell percentage per cluster identified in **e** and colour coded as in **e** (basal proliferative cells shown in light blue, basal in pink and luminal in khaki). **g**, Representative confocal microscopy images of the organoid-based model grown without or with 10 nM DHT and stained for ZO-1 (green, left) and F-actin (red, right). Nuclei were counterstained using Hoechst 33342 (blue, *n* = 5 biological replicates). Scale bar, 25 μm. **h**, TEER analysis of the 2D organoid-based model grown in the presence or absence of DHT (10 nM; *n* = 4 biological replicates). Data are presented as mean ± s.d., and a two-tailed Student's *t*-test was used. Panel **a** created with BioRender.com.

---

with more bacteria invading cells grown with DHT (Fig. 2a). Of note, only live bacteria were able to invade cells, as PFA-killed, heat-killed bacteria or fluorescent beads were not observed intracellularly (Supplementary Fig. 4), indicating an active invasion process by the bacteria rather than a passive uptake by the host cells. To test whether increased bacterial invasion was due to the higher number of luminal cells, we measured infection within the luminal (CD24⁺) and basal (CD49f⁺) populations in the presence/absence of DHT. Results showed a consistent preference of *E. coli* for luminal over basal cells (Fig. 2b

and Supplementary Fig. 5). Similar results were observed using the *E. coli* isolate CP1, originally isolated from a chronic prostatitis patient[16] (Extended Data Fig. 4).

We next addressed the intracellular fate of the bacteria. Interestingly, we observed that UPEC replicates more efficiently in the DHT model than in the Ctrl model, strengthening UPEC preference for luminal cells (Fig. 2c). Most probably, the increased bacterial load at later time points in the DHT model was a result of increased replication within the luminal cells rather than an increased initial invasion (Fig. 2d).

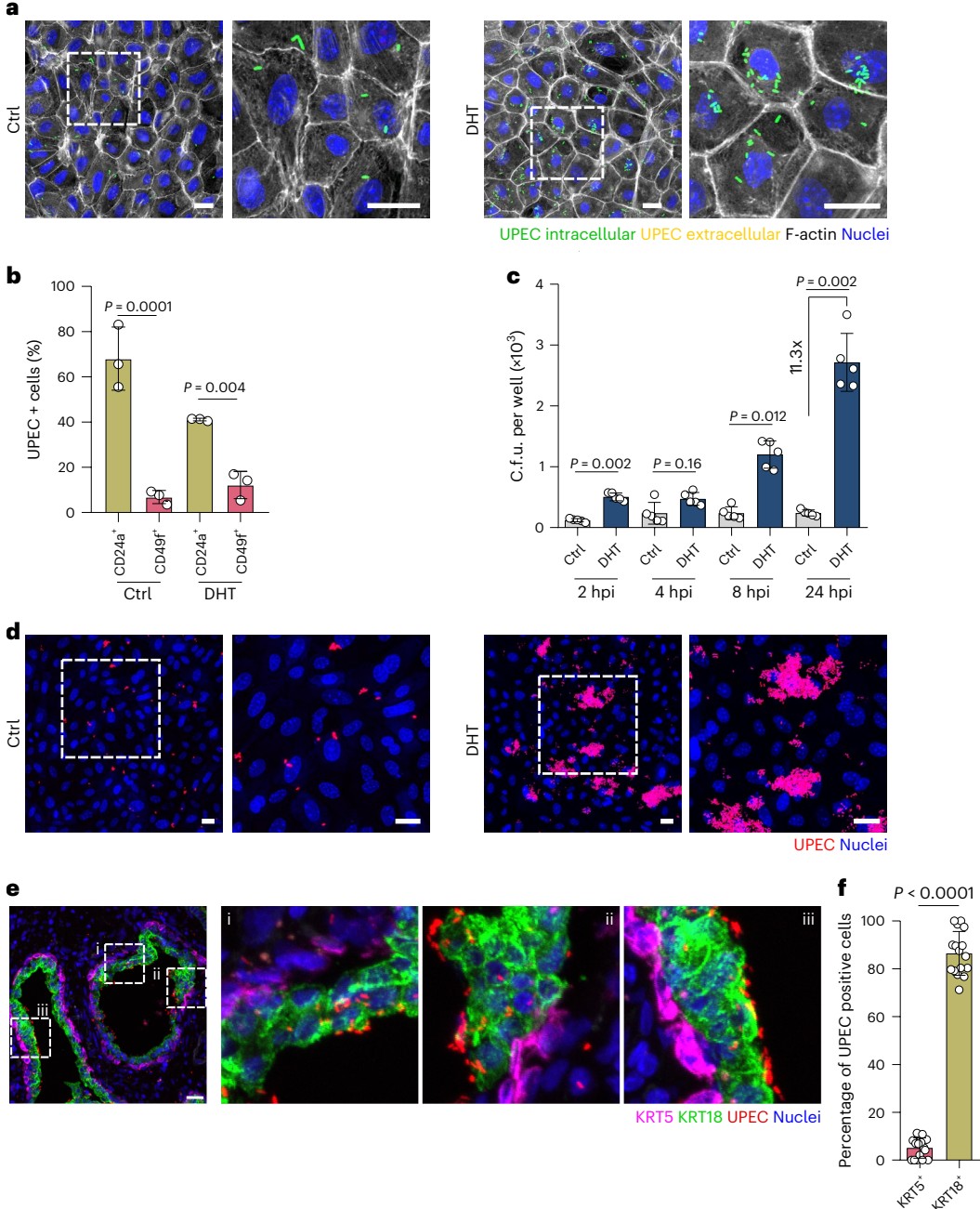

**Fig. 2 | UPEC preferentially invades and replicates in cells within the 2D organoid-based model grown in the presence of DHT. a**, Confocal microscopy images of the prostate organoid-based models at 1 hpi with UPEC UTI89 (green). The model was grown without (left, Ctrl) or with 10 nM DHT (right) and stained for F-actin (grey). Extracellular bacteria were stained for LPS (red), and nuclei were counterstained with Hoechst 33342 (blue). To differentiate extracellular from intracellular bacteria, cells were stained without permeabilization, which results in immunostaining of only extracellular bacteria. Extracellular bacteria are shown in yellow (green + red), and intracellular bacteria in green. Scale bar, 25 μm. *n* = 3 biological replicates. **b**, Flow cytometry analysis of UPEC-infected cells (mCherry⁺). Cells were first gated into luminal (CD24⁺, khaki) and basal (CD49f⁺, pink) populations. The percentage of mCherry⁺ (infected) cells was then quantified within each population. *n* = 3 biological replicates. Gating strategy is shown in Supplementary Fig. 5. **c**, UPEC UTI89 c.f.u. were quantified at 2, 4, 8 and 24 hpi in the organoid-based models. *n* = 5 biological replicates. **d**, Representative confocal images of the organoid-based models at 24 hpi. UPEC is shown in red. Scale bar, 25 μm. *n* = 3 biological replicates. **e,f**, Representative confocal images (**e**) and image quantification graph (**f**) of human prostate tissue incubated with UPEC UTI89 and stained for KRT5 (magenta) and KRT18 (green). Nuclei were counterstained using Hoechst 33342 (blue). Scale bar, 25 μm (*n* = 2 independent experiments, 3 donors). Data in **f** are shown as percentage of infected cells per field of view (*n* = 2 independent experiments, 3 donors; 2–3 images were quantified per condition). Absolute numbers are shown in Extended Data Fig. 6c. Data in **b**, **c** and **f** are presented as mean ± s.d. One-way ANOVA with Tukey's post hoc test was used in **b** and **c**, and a two-tailed Student's *t*-test in **f**.

Similar results were observed with the CP1 strain, as well as two additional *E. coli* isolates from acute prostatitis patients (P10 and P16 (ref. 9); Extended Data Fig. 5a–f), indicating that this pattern is not specific to UTI89 but also applies to *E. coli* strains associated with prostatitis.

Since DHT had no effect on bacterial growth (and was absent during infection assays; Supplementary Fig. 6), this indicates a strong preference for *E. coli* isolates to invade and replicate in luminal prostate cells.

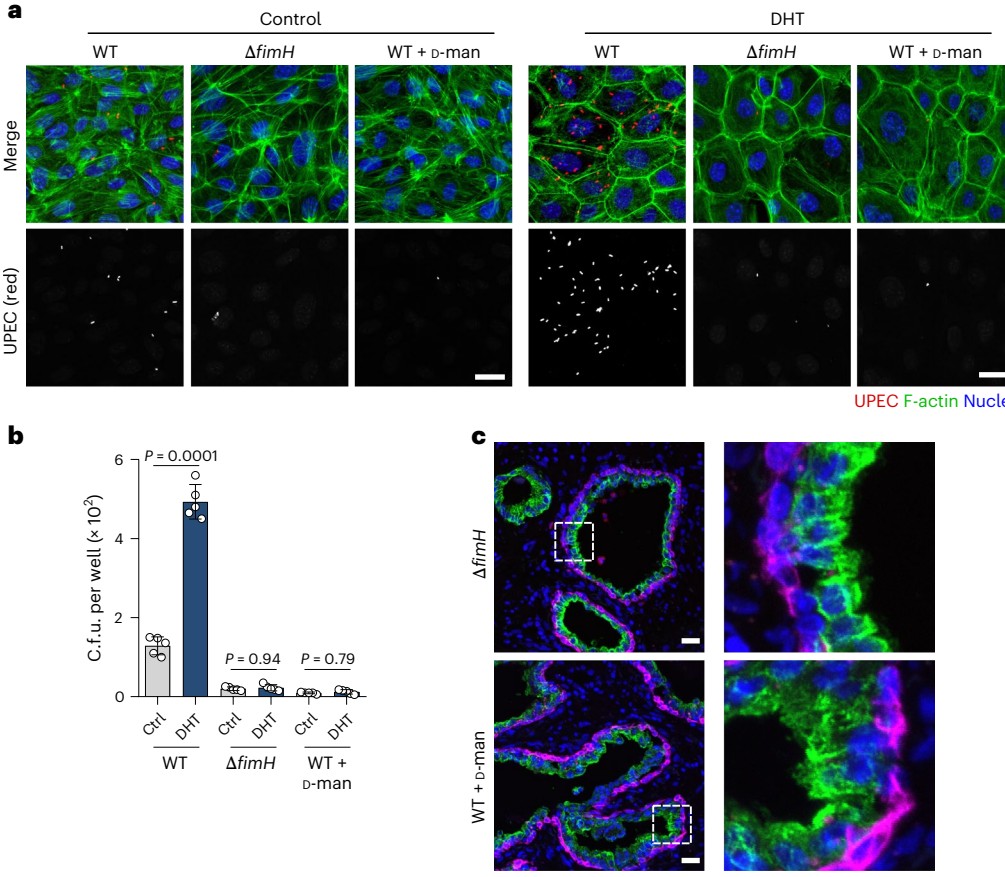

**Fig. 3 | FimH is necessary for maximal invasion into differentiated prostate cells. a,b,** Representative images (**a**) and c.f.u. (**b**) of the prostate organoid-based models infected with WT UTI89, UTI89 Δ*fimH* mutant and WT UTI89 in the presence of 2.5% ᴅ-mannose for 1 h. Bacteria are shown in red. Cells were stained for F-actin (green), and nuclei were counterstained with Hoechst 33342 (blue). Scale bar, 25 μm (*n* = 3 (**a**) or 5 (**b**) biological replicates). Data are presented as mean ± s.d., and a one-way ANOVA with Tukey's post hoc test was used. **c,** Representative confocal images showing the binding of Δ*fimH* mutant and WT treated with 2.5% ᴅ-mannose (red) to human prostate tissue. The tissue was stained for KRT18 (green) and KRT5 (magenta), and nuclei were counterstained using Hoechst 33342 (blue). Scale bar, 25 μm. Image quantification graph for **c** is shown in Extended Data Fig. 6c. Data are representative results from three donors and two independent experiments.

To validate this preference in human tissue, human prostate tissue slides were incubated with *E. coli* strains (UTI89 and CP1). Microscopy analysis showed a higher number of both *E. coli* strains attached to luminal prostate cells (KRT18⁺) than to basal cells (KRT5⁺; Fig. 2e,f and Extended Data Fig. 5g,h). Overall, our results suggest that *E. coli* invades and replicates effectively in a murine prostate model enriched with luminal cells and that bacterial binding is also more efficient in human luminal prostate cells.

### FimH is necessary for *E. coli* invasion into luminal cells

*E. coli* is a versatile pathogen that uses various adhesins to infect different tissues (for example, type 1 pili in bladder versus P pili in kidney[31,32]). Since the interaction between *E. coli* and prostatic cells has thus far not been characterized, we aimed to identify *E. coli* adhesin(s) responsible for bacterial adhesion/invasion into luminal prostate cells. Given their essentiality in bladder colonization, proximity between the bladder and prostate, and the high prevalence of the type 1 pili in prostatitis isolates[9,33], we initially investigated the role of the type 1 pili, specifically the tip adhesin FimH, in this interaction. Deletion of FimH in UTI89 impaired bacterial invasion into both Ctrl and DHT model (Δ*fimH*; Fig. 3a,b). Moreover, allosteric blockage of FimH's mannose-binding pocket with ᴅ-mannose resulted in a similar invasion defect in the WT (Fig. 3a,b). ᴅ-Mannose also blocked invasion by all three *E. coli* prostatitis strains (CP1, P10 and P16; Extended Data Fig. 6a). Similarly, incubation of human prostate tissue slides with the *fimH* deletion mutant (UTI89)

or WT (UTI89 and CP1) in the presence of ᴅ-mannose showed a striking bacterial adhesion impairment (Fig. 3c and Extended Data Fig. 6b,c). Since all three prostatitis isolates expressed FimH under the culture conditions similar to UTI89 (Extended Data Fig. 6d), this indicates FimH to be the main adhesin utilized by *E. coli* to interact with prostate cells.

### FimH binds to the prostate-specific membrane receptor PPAP

We next sought to determine which host cell receptors are involved in the interaction with FimH. Since the 2D model grown with DHT exhibited increased bacterial invasion, we opted to continue using this model for subsequent experiments. So far, four receptors have been described to directly interact with FimH in the bladder: UPK1A[34], uromodulin (UMOD)[35], integrin beta-1 (ITGB1) and alpha-3 (ITGA3)[13]. In addition, desmoglein 2 (DSG2) has recently been described as a FimH receptor in kidney cells[36]. Given that UPK1A and UMOD are not expressed in human prostate tissue, ITGB1 is very lowly expressed in prostate cells, and ITGA3 is only expressed by basal prostate cells[37] (Extended Data Fig. 7a), we hypothesized that FimH might not bind to any of these known receptors but rather to a prostate-specific receptor(s) yet to be identified. To test this, we used a previously described protocol to block FimH binding to ITGB1 and ITGA3 with antibodies[13]. Binding to DSG2 could not be tested since there are no antibodies available against its extracellular domain. While blocking of ITGB1 and ITGA3 in the bladder cell line 5637 resulted in strongly reduced bacterial invasion, UPEC invasion into the prostate model showed no difference between blocked and unblocked

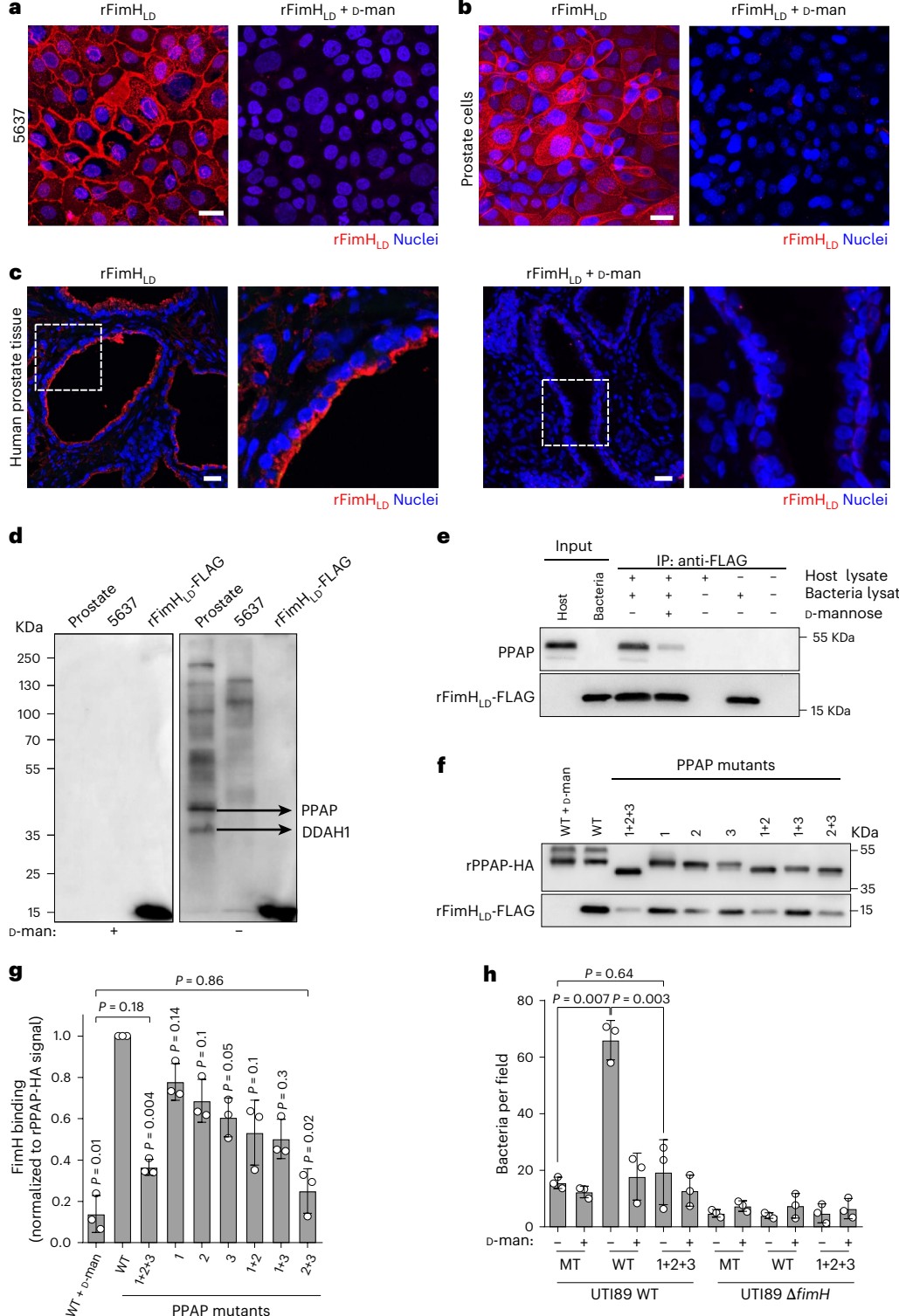

**Fig. 4 | FimH binds to the prostate-specific membrane protein PPAP.**
**a**–**c**, Representative confocal microscopy images of recombinant FimH$_{LD}$ (rFimH$_{LD}$) binding to the 5637 bladder cell line (**a**), organoid-based model grown with 10 nM DHT (**b**) and human prostate tissue (**c**). rFimH$_{LD}$ signal (anti-FLAG antibody) is shown in red. Nuclei were counterstained using Hoechst 33342 (blue). Scale bar, 25 µm ($n$ = 3 biological replicates). **d**, Overlay assay: prostate, 5637 and the rFimH$_{LD}$-FLAG protein samples were resolved in an SDS–PAGE, transferred to PVDF membrane and incubated with the rFimH$_{LD}$-FLAG in the presence or absence of 2.5% of D-mannose. Prostate protein candidates were detected using an anti-FLAG antibody. The two most prominent bands were

excised and identified as PPAP and DDAH1. $n$ = 2 biological replicates. **e**, Co-immunoprecipitation of PPAP using rFimH$_{LD}$-FLAG as bait ($n$ = 3 biological replicates). **f**,**g**, Representative blot (**f**) and quantification (**g**) of rFimH$_{LD}$-FLAG binding to rPPAP-HA WT or mutants (mutant 1 position 94, mutant 2 position 220 and mutant 3 position 333; fold change over PPAP-HA signal; $n$ = 3 biological replicates, one-way ANOVA with Dunnett's post hoc test). **h**, Binding array showing UTI89 WT or UTI89 Δ$fimH$ mutant binding to rPPAP-HA (WT and the triple mutant 1+2 +3). Mock-transfected (MT) cell lysate was used as control ($n$ = 3 biological replicates, one-way ANOVA with Tukey's post hoc test). Data in **g** and **h** are presented as mean ± s.d.

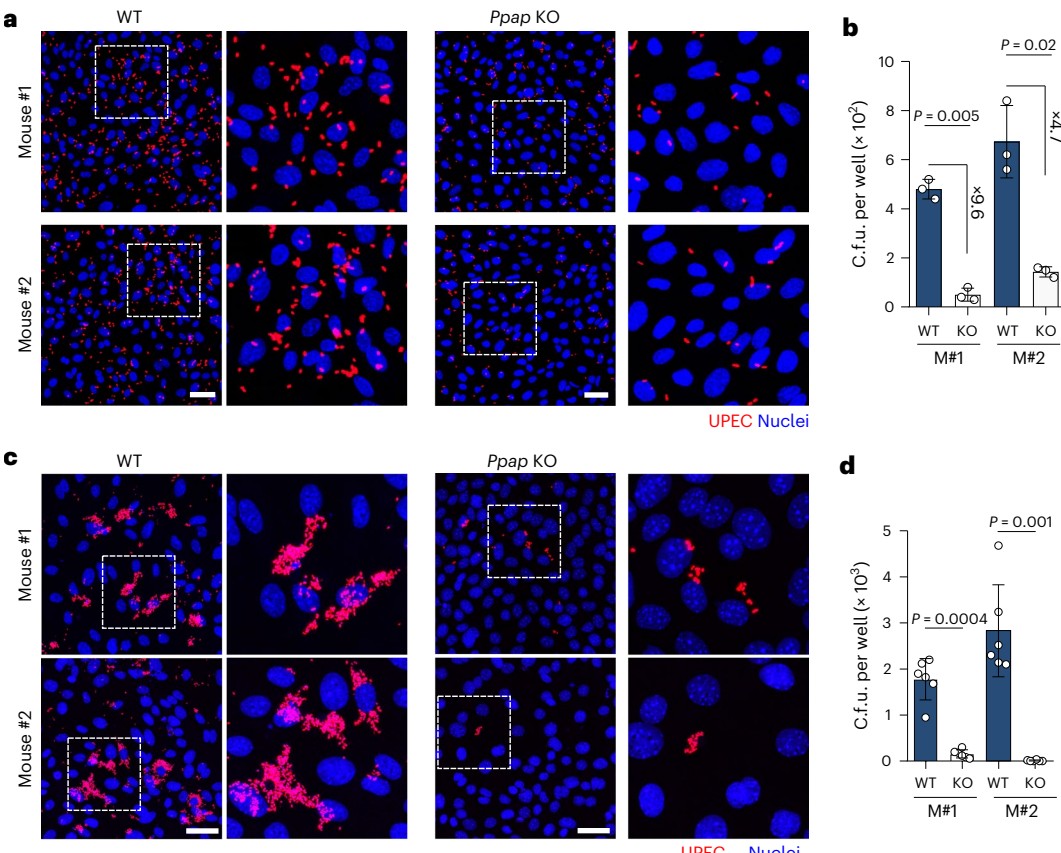

**Fig. 5 | PPAP is necessary for maximal UPEC invasion into prostate cells.**
**a,b**, Representative confocal microscopy images (**a**) and c.f.u. quantification
(**b**) of two *Ppap* KO mouse organoid lines (grown in 10 nM DHT) infected with
UTI89 WT (red) for 1 h. Scale bar, 50 μm (*n* = 3 biological replicates, two-tailed
Student's *t*-test). **c,d**, Representative confocal microscopy images (**c**) and c.f.u.

quantification (**d**) of two *Ppap* KO mouse organoid lines infected with UTI89 WT
(red) for 24 h. Nuclei were counterstained using Hoechst 33342 (blue). Scale bar,
50 μm (*n* = 6 biological replicates, two-tailed Student's *t*-test). Data in **b** and **d** are
presented as mean ± s.d.

cells (Extended Data Fig. 7b). This suggests that these two integrins are
not necessary for FimH attachment to luminal prostate cells.

To identify a potential FimH receptor in prostate cells, we produced
a recombinant version of the lectin domain (LD) of FimH (rFimH$_{LD}$)
with a 3× FLAG tag fused to the C terminus[38]. Sequence comparison
of FimH from UTI89, the three prostatitis isolates and other refer-
ence UPEC strains revealed no mutations in the binding pocket and
only common UPEC-specific single-nucleotide polymorphisms[39]
(Supplementary Fig. 7). Incubation of rFimH$_{LD}$ and subsequent stain-
ing with an anti-FLAG antibody showed that the recombinant protein
was able to bind to 5637 bladder cells with the expected binding pattern
to intercellular junctions, where most ITGB1 and ITGA3 are located
(Fig. 4a). Adding D-mannose to the incubation medium abrogated the
binding of rFimH$_{LD}$ (Fig. 4a). In comparison, rFimH$_{LD}$ bound to the pros-
tate cells with a signal not only specific to the cell–cell junctions but
also on the apical side of these cells (Fig. 4b). Moreover, we observed
that rFimH$_{LD}$ preferentially bound to some cells in the prostate model.
Importantly, these observations were similarly reflected in human
prostate tissue slides, where rFimH$_{LD}$ showed a clear co-localization
with the apical side of the prostate luminal cells (Fig. 4c). This strongly
suggests that FimH receptor(s) might be confined to a specific cell type.

To identify potential FimH receptor(s) in prostate cells, we used a
similar approach previously applied to the identification of ITGA3 and
ITGB1 as FimH receptors[13]. For that, host membrane proteins isolated
from human prostate tissue were resolved by SDS–PAGE, transferred
to PVDF membrane, blocked and then incubated with rFimH$_{LD}$. The
detection of multiple bands using an anti-FLAG antibody indicated

that rFimH$_{LD}$ bound to several host proteins (Fig. 4d). D-Mannose addi-
tion abolished rFimH$_{LD}$ binding to the membrane, as expected. Mass
spectrometry analysis of the two most prominent bands (35/45 kDa)
identified them as *N*(G),*N*(G)-dimethylarginine dimethylaminohydro-
lase 1 (DDAH1) and PPAP, respectively. Because DDAH1 is a cytoplasmic
protein and PPAP is either secreted or a plasma membrane protein, we
focused on PPAP as a potential FimH receptor.

First, we validated interaction between rFimH$_{LD}$ and PPAP by
immunoprecipitation using rFimH$_{LD}$ as bait (Fig. 4e). In general, FimH
binds via its lectin domain to high-mannose type *N*-glycans of host
glycoproteins. Thus, we tested whether mannosylated residues in
PPAP were required for FimH binding. PPAP has three *N*-glycosylated
asparagine residues (positions 94, 220 and 333) with different gly-
cosyl units[40]. To modify these residues, we generated an HA-tagged
PPAP recombinant protein, which also served to validate PPAP bind-
ing to rFimH$_{LD}$ (Supplementary Fig. 8). Then, we exchanged all three
asparagines for alanines and tested their respective contribution to
FimH binding. Single mutations only showed a small reduction in
rFimH$_{LD}$ binding (Fig. 4f,g). However, a strong reduction was observed
in the double mutant for positions 220 and 333. A similar effect was
observed in the triple mutant (Fig. 4f,g), suggesting that glycosylation
in position 94 is not as crucial as in positions 220 and 333. The addition
of D-mannose to the reaction drastically reduced FimH capacity to
bind to PPAP (Fig. 4f,g) to a greater extent than the 220 + 333 or triple
mutant, although the difference was not statistically significant. To
confirm PPAP binding to the full-length FimH, we performed a binding
array assay with WT UTI89 bacteria, where rPPAP was bound to a glass

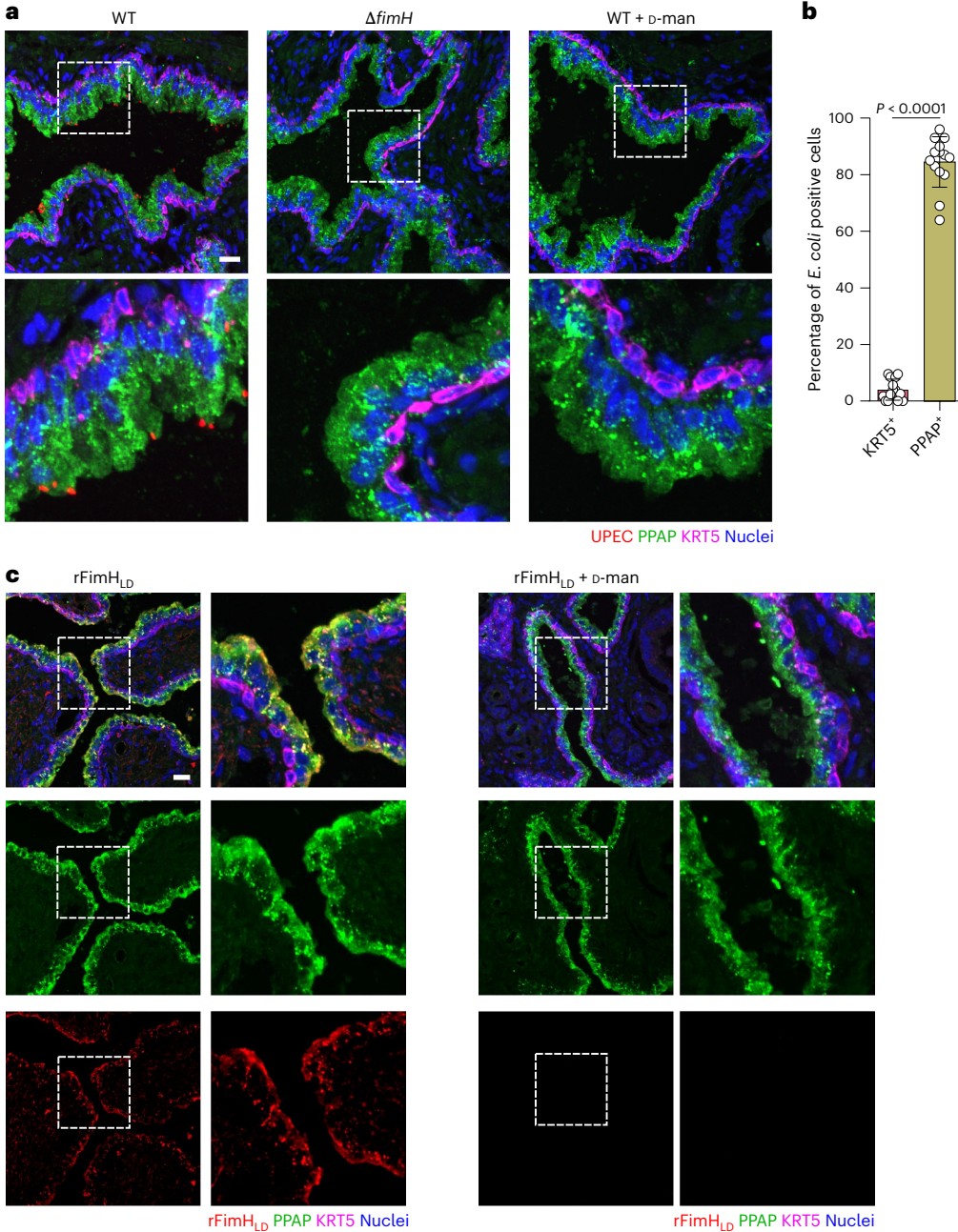

**Fig. 6 | rFimH$_{LD}$ and UPEC co-localize with PPAP$^+$ cells in human prostate tissue.**
**a**, Representative confocal microscopy images of the binding of UTI89 WT, Δ*fimH*
mutant and WT in the presence of 2.5% D-mannose to human prostate tissue.
Bacteria are shown in red, PPAP in green and KRT5 in magenta (*n* = 2 independent
experiments, 3 donors). **b**, Image quantification graph of the binding of UTI89
WT. Data are shown as percentage of infected cells per field of view (*n* = 2
independent experiments, 3 donors; 2–3 images were quantified per condition).

Absolute numbers are shown in Extended Data Fig. 9c. Data are presented as
mean ± s.d., and a two-tailed Student's *t*-test was used. **c**, Representative confocal
microscopy images of rFimH$_{LD}$ (red) binding to human prostate tissue and
stained for PPAP (green) and KRT5 (magenta). Nuclei were counterstained using
Hoechst 33342 (blue). Scale bar, 25 μm. Data are representative results from two
donors and two independent experiments.

microscopy slide and incubated with live bacteria. This showed that
bacteria bound to WT PPAP in higher numbers than to the triple mutant
PPAP (Fig. 4h). As expected, the presence of D-mannose or deletion
of FimH abrogated binding to both WT and mutated PPAP (Fig. 4h).
Together, these data show that *E. coli* FimH binds to the prostate recep-
tor PPAP and that PPAP glycosylation is important for this interaction.

**PPAP is required for maximal *E. coli* invasion into prostate cells**
Given that FimH assists bacterial invasion into the bladder cells[41] and
that FimH is also necessary for invasion into mouse prostate cells, we

hypothesized that PPAP aids the invasion process. To test this, we gen-
erated two murine *Ppap* knockout (KO) organoid lines using CRISPR/
Cas9 technology as previously described[42]. Sanger sequencing and FISH
microscopy confirmed the successful knockout of the *Ppap* receptor
in murine organoid cells (Supplementary Fig. 9). Infection of *Ppap* KO
cells with UTI89 and the other three prostatitis isolates resulted in a
significant decrease in bacterial invasion at 1 h post infection (hpi) and
intracellular replication at 24 hpi compared to WT cells (Fig. 5a–d and
Extended Data Fig. 8). This further confirms the essential role of PPAP
in supporting maximal *E. coli* invasion.

### FimH co-localizes with PPAP in human prostate tissue

After observing that FimH binds to PPAP in vitro, we tested whether live WT bacteria prefer PPAP+ cells ex vivo. To assess this, human prostate tissue slides were incubated with live WT UTI89 or CP1 (±D-mannose) and stained for PPAP. Confocal microscopy showed both bacteria mainly near PPAP+ cells (Fig. 6a,b and Extended Data Fig. 9). Validating our in vitro data, the Δ*fimH* mutant or the WT with D-mannose failed to bind the prostate tissue (Fig. 6a and Extended Data Fig. 9). In addition, using the same approach as in Fig. 4a, rFimH$_{LD}$ protein clearly co-localized with PPAP in luminal cells (Fig. 6c). Overall, these results demonstrate that *E. coli* binds to PPAP+ prostate luminal cells via FimH.

## Discussion

*E. coli* is a versatile bacterial pathogen that can infect various organs of the urogenital tract, using distinct strategies for each tissue. In this work, we established an adult stem cell organoid-based model of the mouse prostate to study the main hallmarks of *E. coli* infection in the prostate epithelium, an understudied tissue susceptible to *E. coli* infection. Using this model, we show that different urogenital *E. coli* strains are able to adhere to, invade and replicate within prostate luminal cells and that this is mediated by FimH binding to the glycosylated prostate-specific membrane receptor PPAP (Extended Data Fig. 10).

Recent advances in organoid technology have significantly improved infection biology by enabling more accurate in vitro studies of pathogens in primary cells[43–46]. These primary cells can differentiate into the various epithelial cell types found in tissues, under suitable conditions. This is crucial for studying infectious diseases, as pathogens interact differently with distinct host cell subtypes. Therefore, ensuring that an in vitro model accurately mimics epithelial heterogeneity is essential for effective research. Here, using mouse prostate adult stem cell-derived organoids, we developed a model reproducing the cell composition of the prostate epithelium, except for the rare neuroendocrine cells. Our data show marked differences between the 3D organoid and the 2D organoid-based model: while the 3D model is enriched with undifferentiated basal stem cell-like cells, the 2D model mainly consists of differentiated luminal cells. This aligns with our previous study in gastric organoids, where 2D cultures exhibited higher differentiation towards pit cells than 3D cultures under identical culture conditions[42]. The fact that simply seeding the cells on a 2D surface leads to increased cell differentiation, despite using the same culture medium, suggests that other factors may be at play. For instance, differences in physical and mechanical cues, such as the low stiffness of the extracellular matrix in 3D culture compared with the high stiffness of tissue culture plastic, and the presence of extracellular matrix components and growth factors in 3D cultures, may influence cell fate decisions. Further work is needed to define their contribution. Furthermore, in our model, supplementing the medium with 10-fold higher DHT concentration induced basal-to-luminal differentiation, resembling cell proportions in the native tissue[25,27]. In this context, previous work on mouse organoids has shown that androgen deprivation decreases luminal markers, while androgen restoration regenerates the luminal cell layer[25,28]. Conversely, DHT removal from the medium shifted cell differentiation towards a basal-like phenotype. Such a model, enriched with basal stem cell-like cells, could hold additional benefits in studying, for example, prostate basal cell carcinomas or basal responses to specific stimuli.

In our study, *E. coli* showed a clear preference for luminal cells over basal cells. Previous studies have reported bacterial invasion and limited replication in human prostate-transformed cell lines[16,47–49]. However, cell differentiation was not addressed in those studies. Although in vivo data on prostate invasion are lacking, our findings mirror bladder infections, where bacteria replicate more efficiently in superficial umbrella cells than in intermediate cells. While adhesion to differentiated umbrella cells can be explained by the presence of specific host receptors (for example, UPK1A)[34], the higher intracellular replication rate in these cells remains poorly understood. Several hypotheses, including differences in cytoplasmic space or actin filaments organization, have been proposed[50,51], but none have been proven. Nonetheless, bacterial invasion into bladder cells allows the pathogens to escape the immune response, antibiotic treatment and elimination via micturition. Although the prostate gland is not voided as frequently as the bladder, it seems plausible that invading prostate luminal cells would benefit the bacteria in a similar manner.

Several studies have shown that testosterone increases susceptibility to and worsens bladder and kidney infections, mainly through its immunosuppressive effects[29,52–55]. However, none of those studies examined its effect on epithelial cells. In our model, DHT increased cell differentiation towards *E. coli* target cells, facilitating infection. Notably, DHT had no effect on *E. coli* growth, and it was removed during bacterial incubation to avoid any potential influence. Other studies have reported reduced *E. coli* invasion and replication in prostate cell lines after DHT treatment. However, those studies used much higher concentrations (20–40 μg ml⁻¹, equivalent to 68.8–137.6 μM[54] or 5–20 μg ml⁻¹, equivalent to 17.2–68.8 μM[49]) compared with our physiological level of 10 nM[56].

Bacterial adhesion to host cells is critical for infection and precedes invasion. *E. coli* possesses a versatile array of adhesins or pili, including type 1 pili, pyelonephritis-associated pilus (Pap), S pili, Dr adhesins and others[56], enabling colonization of diverse niches. This is not trivial for the bacteria: while *E. coli* uses PapG adhesins on P pili to bind to galactose-rich membrane glycolipids on kidney cells[57], it uses type 1 pili to bind to mannosylated residues on the surface of bladder urothelial cells[41] and vaginal epithelial cells (although other adhesins also contribute to female genital tract colonization)[58,59]. Our data show that the FimH adhesin is the main adhesin responsible for *E. coli* binding to prostate cells. However, a small fraction of FimH mutant bacteria still adhered, suggesting the potential presence of other adhesin/s or other surface structures contributing to it. FimH binds to different host receptors in the bladder, including UPK1A (present in umbrella cells)[34], ITGA3 and ITGB1 (present in umbrella and intermediate cells)[13], and UMOD (secreted by kidney cells into the urine)[35]. Since these are either absent or lowly expressed in the prostate, it raised the question of whether additional prostate-specific FimH receptors exist. Recombinant FimH binding to the DHT-enriched luminal cell model showed a distinct apical signal pattern, unlike the basolateral signal observed in bladder cells. Moreover, the small, punctate, round structures observed on the central apical side may suggest binding to prostatic extracellular vesicles or prostasomes. Our results indicate that the PPAP protein functions as a FimH receptor in the prostate epithelium. PPAP, a non-specific tyrosine phosphatase exclusively expressed by luminal prostate cells, dephosphorylates multiple substrates under acidic conditions[60–63], which helps sperm motility, although the exact mechanism remains unclear[64]. PPAP also inhibits prostate cancer via ERBB2 dephosphorylation and MAPK signalling deactivation[62]. Both human and murine PPAP exist in two isoforms produced through alternative splicing: isoform 1 is secreted (soluble), and isoform 2 is a type-I transmembrane protein[65]. Since only isoform 2 could function as a cellular receptor, we focused on it. Nevertheless, secreted PPAP could diffuse within the prostate lumen and potentially influence infection beyond its cell of origin. In particular, secreted PPAP may act as a decoy receptor by binding to FimH, thereby blocking bacterial adhesion to membrane-bound PPAP and promoting bacterial clearance, similar to the role described for uromodulin in the bladder[66]. Alternatively, it could facilitate bacterial aggregation or biofilm formation, enhancing bacterial survival. These scenarios underscore the dual potential role of secreted PPAP to either protect the host or facilitate bacterial colonization. Further work will be necessary to determine their relevance during infection.

Our knockout organoid lines showed that PPAP is required for maximal *E. coli* invasion in differentiated cells. However, a small proportion of intracellular *E. coli* was still found in the KO cells, suggesting

the potential presence of other host receptor/s. This aligns with FimH binding observed in other tissues, where multiple receptors can be engaged within the same tissue. A decrease in bacterial intracellular replication was also observed in the *Ppap* KO cells, which could not be explained by reduced invasion alone. This suggests that PPAP may affect bacterial replication or, more likely, that in its absence, bacteria invade less permissive cell types via alternative receptors, where intracellular replication may not occur.

All known FimH receptors are heavily mannosylated glycoproteins. Similarly, human PPAP contains three *N*-glycosylation motifs (Asn94, Asn220 and Asn333)[40,67] that could serve as anchors for the FimH lectin binding pocket. Of note, since both isoforms (secreted and membrane-bound) share the same glycosylation pattern, this supports the hypothesis that soluble PPAP could act as a soluble FimH ligand. The three positions have different numbers of glycosyl units, with Asn333 being the most glycosylated[40]. We hypothesized that substituting asparagine with alanine to remove PPAP glycosylation would affect FimH binding to PPAP. Our results showed that glycosylation of Asn220 and Asn333 was essential, while Asn94 was less important.

In our study, both the rFimH protein and live *E. coli* bacteria demonstrated a clear preference for PPAP+ luminal cells in human prostate tissue. This is presumably the first observation of *E. coli* interaction in ex vivo human prostate tissue. Although co-localization of the PPAP signal with *E. coli* or rFimH$_{LD}$ in the tissue does not directly confirm binding, it strongly suggests that our in vitro data reflect a more physiologically relevant context, thereby reinforcing the significance of our findings.

Importantly, we observe no significant difference in pathogenesis between the reference UPEC strain UTI89 and the three prostatitis isolates in our study, supporting the notion that bacterial prostatitis should be classified as a form of urinary tract infection[6].

The present work underscores the significance of type 1 pili and the mannosylated epithelial receptor PPAP in *E. coli* infection of prostate cells. Since FimH is already targeted by small-molecule inhibitors and vaccines in bladder infection patients[68,69], these therapeutics could also be effective in preventing or treating bacterial prostatitis.

## Methods

### Reagent and resource sharing
Further information and requests for resources should be directed to C.A. (carmen.aguilar@uni-wuerzburg.de).

### Organoids generation
Mouse prostate organoids were generated from four 8-month-old WT male C57BL/6 mice as previously described[26,28], with minor modifications. Briefly, prostate tissue was minced into small pieces and digested in 5 mg ml$^{-1}$ collagenase (C9891-100MG, Sigma-Aldrich) with 10 μM RHOKi (Y-27632; M1817, AbMole) for 1 h at 37 °C on a shaking platform. Tissue fragments were then washed once with advanced Dulbecco's modified Eagle medium (adDMEM)/F12 (12634028, Thermo Fisher) supplemented with 10 mmol l$^{-1}$ HEPES (15630056, Thermo Fisher), 1× GlutaMAX (35050-038, Thermo Fisher; adDMEM/F12+/+) and centrifuged at 150 × *g* for 5 min at 4 °C. The cell pellet was resuspended in TrypLE Express (12605028, Gibco) supplemented with 10 μM RHOKi (Y-27632, M1817, AbMole) for 15 min at 37 °C. The cell suspension was then washed, filtered with a 100-μm strainer (43-10100, pluriSelect) and centrifuged at 350 × *g* for 5 min at 4 °C. Cells were seeded in 50-μl drops of Matrigel (356231, Corning). Culture medium contained: adDMEM/F12 (12634028, Thermo Fisher), 10 mmol l$^{-1}$ HEPES (15630056, Thermo Fisher), 1× GlutaMAX (35050-038, Thermo Fisher), 1× B27 (12587010, Thermo Fisher), 1 mM *N*-acetylcysteine (A9165-25G, Sigma-Aldrich), 10% Noggin-conditioned medium, 10% R-spondin 1 conditioned medium, 50 ng ml$^{-1}$ EGF (AF-100-15, Peprotech), 1 nM dihydrotestosterone (5α-Androstan-17β-ol-3-one, DHT; A8380, Sigma-Aldrich), 0.2 μM TGFβi (A-83-01, Tocris) and 100 ng ml$^{-1}$ Primocin (Ant-pm-1, Invivogen). For propagation, organoids were split weekly at a 1:4 ratio.

To adhere to the 3R principles (replacement, reduction, refinement), we utilized only leftover material from mice previously used for other experiments. No ethics approval was required. No additional animals were killed for this study.

### Organoid-based model seeding
One-week-old mouse prostate organoids were used to seed the organoid-based model. Organoids were collected and mechanically disrupted. Organoid fragments were washed with adDMEM/F12+/+ and centrifuged at 450 × *g* for 5 min. Organoid fragments were then resuspended in TrypLE Express (12605028, Gibco) and incubated at 37 °C for 15 min. The single-cell suspension was washed and centrifuged at 450 × *g* for 5 min. Single cells were counted and seeded at 12,500 cells per well in a 48-well plate (833923, Sarstedt) using the organoid culture medium without Primocin and allowed to grow for a week. RHOKi (Y-27632; 10 μM, M1817, AbMole) was added at the seeding time. To induce cell differentiation, 10 nM DHT was added to the medium on the day of splitting or seeding and kept throughout the experiment. Ethanol was used as a vehicle control in the control condition.

### Bacterial culture and infection
A list of bacterial strains used in this study is provided in Supplementary Table 3. UPEC strain UTI89 WT, *E. coli* strains CP1 (ref. 16), P10[9] or P16[9] expressing mCherry or GFP were obtained by transforming the bacteria with the plasmid pFPV or pXG10, respectively. Chromosomal deletion of *fimH* in UPEC strain UTI89 was achieved by λ Red recombineering[70]. Briefly, a DNA fragment including the cat cassette flanked by FRT sites and 60 bp homologous regions to the 5′ and 3′ end of fimH was PCR-amplified from pKD3 using the primers fimH:cat_npt_ff (ATGAAACGAGTTATTACCCTGTTTGCTGTACTGCT-GATGGGCTGGGTGTAGGCTGGAGCTGCTTC) and fimH:cat_npt_rev (TTGATAAACAAAAGTCACGCCAATAATCGATTGCA CATTCCCTGC-CATATGAATATCCTCCTTAGTTCC). All primer sequences are listed in Supplementary Table 4. UPEC UTI89 was then transformed with 250 ng of the obtained PCR product, and mutants were selected on LB agar plates supplemented with 12.5 μg ml$^{-1}$ chloramphenicol. The deletion of *fimH* was verified by PCR, and the mutant was cured from pKD46 by incubation at 42 °C.

For in vitro infection, bacteria were grown in LB broth overnight at 37 °C under static conditions. A 24-h culture was inoculated with optical density at 600 nm (OD$_{600}$) of 0.05 from the overnight culture and grown under the same conditions. Bacteria were collected by centrifugation and resuspended in organoid culture medium. Bacteria were then diluted to a multiplicity of infection of 100 in the organoid culture medium and added to the wells. D-Mannose (M6020, Sigma-Aldrich) at a final concentration of 2.5% was added to the medium simultaneously with the bacteria when indicated. No DHT was added to the medium during infection. After 1-h incubation, the unbound bacteria were washed, and the culture medium was exchanged with fresh medium supplemented with 50 μg ml$^{-1}$ gentamicin (G1272, Sigma-Aldrich). After 30 min, the medium was changed to a lower concentration of gentamicin (10 μg ml$^{-1}$; G1272, Sigma-Aldrich) and left until analysis. At the indicated time, the cells were washed with PBS and collected for further analysis. For OD$_{600}$ measurements over time (growth curves), 24-h cultures were inoculated in a 96-well plate to an initial OD$_{600}$ of 0.01 in the presence of 10 nM DHT or ethanol (vehicle).

### Colony-forming units
A colony forming unit (c.f.u.) assay was used to quantify intracellular bacteria. At the indicated time, organoid-infected cells were washed with PBS and lysed with 0.1% Triton X-100 (T9284, Sigma-Aldrich) in PBS. Cell lysates were then serially diluted and plated on LB agar plates and grown overnight.

For the antibody blocking, experiments were done as previously

published[13]. Briefly, cells were pre-incubated with purified monoclonal antibodies for 30 min before adding bacteria. The final concentration of each antibody was 1.5 µg per well. The following antibodies were used: β1 integrin (6S6; sc-53711, Santa Cruz Biotechnology) and α3 integrin (P1B5; sc-13545, Santa Cruz Biotechnology). PBS was used as control in the 'No Ab' condition.

## Flow cytometry

Cells were seeded as described above. For analysis of naïve cells, 7-day-old 2D models were dissociated into single cells with TrypLE Express (12605028, Gibco) and washed with FACS buffer (PBS, 10% FBS, 50 mM EDTA). Single-cell suspensions were then stained with anti-mouse CD24 (FITC, 1:500; 101805, Biolegend) and anti-mouse CD49f (Alexa647, 1:750; 313610, Biolegend) or isotype control (FITC, 1:500; 400633; Alexa 647, 1:750, 400526, Biolegend) antibodies for 30 min at 4 °C. Cells were then measured with the NovoCyte Quanteon flow cytometer and analysed with NovoExpress software (Agilent; see Supplementary Fig. 5). For infected cells, cells were infected as described above, washed 3 times with PBS after 1 hpi and then stained following the same protocol.

## Staining and immunofluorescence

Cells seeded on a µ-Slide 18-well slide (81816, Ibidi) were washed with PBS and fixed with 4% paraformaldehyde (PFA) for 15 min at room temperature, permeabilized with 0.5% Triton X-100 (T9284, Sigma-Aldrich) in PBS for 15 min and then blocked with blocking buffer (1% BSA in PBS) for 30 min at room temperature. Primary antibodies were diluted in blocking buffer and incubated overnight at 4 °C, followed by 1 h at room temperature. Anti-cytokeratin 5 (ab52635, Abcam), anti-cytokeratin 5 Alexa Fluor 647 antibody (ab193895, Abcam), anti-CD24a (10600-1-AP, Proteintech), anti-ACPP (PPAP, LS-C292593, LSBio) and anti-KRT18 (ab133263, Abcam) antibodies were used at 1:100. Anti-ZO-1 polyclonal antibody (21773-1-AP, Proteintech) was used at 1:750. Cells were washed with PBS and incubated with the secondary antibodies anti-rabbit IgG Alexa Fluor 488 (A21441, Invitrogen) or anti-mouse IgG Alexa Fluor 594 (A21201, Invitrogen) at a dilution of 1:500. To stain F-actin, cells were stained with Flash Phalloidin Green 488 (424201, Biolegend), Red 594 (424203, Biolegend) or Alexa 647 (A30107, Invitrogen) at 1:250 dilution for 1 h. For differential staining of intracellular versus extracellular bacteria, after fixation with 4% PFA, the permeabilization step was omitted. After blocking with 1% BSA, extracellular bacteria were stained with anti-*E. coli* LPS (ab35654, Abcam) and anti-mouse IgG Alexa Fluor 594 (A21201, Invitrogen) at a dilution of 1:500. Nuclei were counterstained with Hoechst 33342 (1:5,000; H3570, Life Technologies).

To stain and image the human prostate tissue, tissue explants were obtained from patients undergoing surgery (biopsy) at the University Hospital of Würzburg. Given that a biopsy is not a common procedure in bacterial prostatitis or male UTI patients, we utilized prostate tissue blocks from patients who had undergone biopsies for other reasons, such as cancer screening. Only tissues identified as non-cancerous by the pathologist were included in the analysis. This study was approved by the Ethics Committee of the University of Würzburg (Approval 168/22), and informed consent was obtained from all donors. No discrimination based on age, race or genotypic information was applied. Samples from three donors (45–80 years old) were used. No financial compensation was provided. Tissue pieces were fixed with 4% PFA, dehydrated overnight with 30% sucrose and then embedded in O.C.T (Tissue-Tek O.C.T., TTEK, Hartenstein). Tissue slides were then rehydrated with PBS for 10 min, and immunostaining was performed as described for the µ-Slide 18-well slides.

Confocal microscopy images, shown as maximum projected Z-stack images, were acquired with a Leica SP5 laser scanning confocal microscope or with a Leica Stellaris 5 confocal microscope and LAS AF Lite software (Leica Microsystems). The images were analysed with ImageJ.

## Bacterial incubation of human prostate tissue slides

Prostate tissue slides were obtained as described above. After rehydration, slides were incubated with UPEC UTI89 WT or Δ*fimH* ($OD_{600}$ = 0.02) for 30 min on a shaking platform in the presence or absence of D-mannose (2.5%, M6020, Sigma-Aldrich). The slides were then washed with PBS, fixed with 4% PFA and further processed for immunofluorescence as described above.

## Data mining from 'The Human Protein Atlas'

Immunohistochemistry images of human prostate tissue samples stained for UPK1A, UMOD, ITGB1, ITGA3 and DSG2 were obtained by datamining the human protein atlas database (www.proteinatlas.org)[37].

## TEER measurement

An ECIS 8-well transfilter array connected to ECIS Z-Theta (Applied Biophysics) was used to measure the TEER of the organoid-based model. Cells were seeded as described above on an 8-well PET slide with 40 electrodes per well (8W10E+, Applied Biophysics). TEER values were measured at 400 Hz for 48 h (5–7 days after seeding). The TEER of a blank well (only culture medium) was used as control and subtracted from all TEER values.

## rFimH$_{LD}$ production and far western blot overlay assays

The lectin domain of *fimH* (1–156 amino acids) of UTI89 was amplified from genomic DNA using the forward primer 5′-CCCCGAATTCAGGAGGAGATTGTAATGAAACGTGTTATTACCCTG-3′ and the reverse primer 5′-GCGCGGATCCTTACTTATCGTCGTCATCCTTGTAGTCGCCGCCAGTAGGCACCAC-3′, cloned into the EcoRI-BamHI sites of plasmid pQE-30 and transformed into *E. coli* T4 Express competent cells (C3037I, New England BioLabs). Expression of fimH-FLAG was induced with 1 mM IPTG (2316.3, Carl Roth) for 5 h. Bacterial cells were collected and resuspended in cold 20 mM Tris/20% sucrose solution along with the protease inhibitor cocktail (05892791001, Sigma-Aldrich) and 200 µM PMSF. The cells were then sonicated and incubated with 75 µg ml$^{-1}$ of lysozyme for 30 min. Cell debris was removed by centrifugation at 15,000 × *g* at 4 °C for 20 min. Supernatant was filtered twice through a 0.45-µm filter to remove smaller debris. Proteins were then concentrated and washed with 20 mM Tris using a 10-kDa molecular weight cut-off centrifugal filter (88527, Thermo Scientific). The presence of rFimH$_{LD}$ protein was confirmed by western blot analysis using the antibodies anti-FimH (CSB-PA362349ZA01ENV, Bio Trend, 1:1,000) and anti-FLAG (1:1,000; 66008-4-Ig, Proteintech).

Far western blot overlay assays were done as previously described[13] with minor modifications. Human prostate tissue from four patients was lysed using the Proteoextract Native Membrane Extraction kit (Sigma-Aldrich, 444810-1KIT) to enrich for membrane proteins. Pooled protein lysates were then separated in a 10% SDS–PAGE gel. The proteins were then transferred on a PVDF transfer membrane and blocked with 1% BSA/1% milk in TBS-T (TBS 0.1% Tween-20, pH 7.4) for 20 min. The membrane was incubated with the bacterial lysate containing the rFimH$_{LD}$ at a final concentration of 100 µg ml$^{-1}$, alone or in the presence of 2.5% D-mannose (M6020, Sigma-Aldrich) diluted in the blocking buffer for 90 min. After TBS-T washes, the membrane was incubated with an anti-FLAG antibody (66008-4-Ig, Proteintech, 1:1,000) for 1 h, washed and incubated with the secondary anti-mouse IgG-HRP conjugate (GENA931, Sigma-Aldrich, 1:10,000). Signals were detected using SuperSignal West Dura Extended Duration Substrate (Pierce, 34075) using an ImageQuant LAS 4000 CCD camera (GE Healthcare). Host protein bands that bound to rFimH$_{LD}$ were excised from the duplicate SDS–PAGE gel that was stained with Coomassie blue, and identified by mass spectrometry at the Mass Spectrometry Facility of the Rudolf Virchow Center for Integrative and Translational B (University of Würzburg). Mass spectrometry data are provided in the Supplementary Table 5. Uncropped images of immunoblots are included as Source Data.

## FimH sequencing

The *fimH* DNA sequence from the strains UTI89, CP1, P10 and P16 was amplified by PCR using the Phusion enzyme (NEB, M0530) and specific primers (forward 5′-CAGGCAGTGATTAGCATCACCTA-3′; reverse 5′-AGGCTCTGTTCGGATTGTCG-3′). Purified PCR products were then Sanger sequenced using either the forward or reverse primer. Nucleotide sequences were then translated in silico using the ExPASy translation tool (https://web.expasy.org/translate/) and compared with each other and with the amino acid sequences of FimH from other UPEC reference strains (NU14, CFT073 and 536, sequences obtained from UniProt; www.uniprot.org). DNA sequences have been deposited in GenBank (ID 3019276) through BankIt.

## FimH western blot

Bacterial strains were grown as described above for in vitro infections. One ml of cells at $OD_{600} = 1$ was collected by centrifugation (12,000 × $g$ for 5 min), lysed in Laemmli's buffer, sonicated and separated on SDS–PAGE gels, followed by electro-transfer to nitrocellulose membranes. Antibodies used for western blot analysis were anti-FimH antibody (LSBio, CSB-PA362349ZA01ENV; 1:1,000), anti-GroEL antibody (Sigma, G6532; 1:5,000) and HRP-conjugated anti-rabbit secondary antibody (Proteintech, SA00001-2; 1:10,000). Membranes were developed using ECL substrate and imaged using the ImageQuant LAS 4000 system. Uncropped images of immunoblots are included as Source Data.

## rPPAP protein expression

The PPAP isoform 2 (membrane protein) protein tagged with HA in the C terminus was expressed using a commercially available plasmid from Sino Biologicals (HG10959-CY). The plasmid was transfected into HeLa 229 cells using X-tremeGENE HP DNA Transfection Reagent (6366236001, Merck) at a 1:2 ratio. Cells were collected at 48 h post transfection in lysis buffer containing 1% NP-40, 5% glycerol, 25 mM Tris, 150 mM NaCl, 1 mM EDTA and protease inhibitor cocktail (05892791001, Sigma-Aldrich). Cell lysis was completed by incubation on ice for 1 h, followed by centrifugation at 15,000 × $g$ for 1 h. Pellets were discarded and PPAP expression was examined by western blot using anti-HA polyclonal antibody (51064-2-AP, Proteintech, 1:5,000) and the secondary anti-rabbit IgG-HRP conjugate (GENA934, Sigma-Aldrich, 1:10,000). Uncropped images of immunoblots are included as Source Data.

## Co-immunoprecipitation

Co-immunoprecipitations were performed using the Pierce Anti-DYKDDDDK Magnetic Agarose (A36797, Thermo Fisher) or the Pierce anti-HA magnetic beads (13464229, Thermo) to pull down rFimH-FLAG or PPAP-HA-binding partners, respectively, and following manufacturer instructions. Briefly, for the co-immunoprecipitation with the Pierce Anti-DYKDDDDK Magnetic Agarose (A36797, Thermo Fisher), bacteria protein lysates were obtained as previously described and incubated with beads (100 µl lysate, 30 µl beads) for 30 min at 37 °C. Beads were then washed with wash buffer (20 mM Tris-HCl, 150 mM NaCl, pH 7.2) and incubated with 200 µl of the human prostate protein lysates for 30 min at 37 °C. Beads were then washed with wash buffer, eluted by boiling in 50 µl of Laemmli buffer and analysed by western blot assay. An anti-PPAP antibody (HPA063916, Atlas Antibodies, 1:300) was used in a western blot assay to validate the co-immunoprecipitation of PPAP from the human prostate lysate with the $rFimH_{LD}$-FLAG beads.

For co-immunoprecipitation using the Pierce anti-HA magnetic beads (13464229, Thermo Scientific), PPAP-expressing HeLa 229 cell (and mock control) lysates were incubated with beads (150 µl lysate, 20 µl beads) for 30 min at room temperature. Beads were then washed with wash buffer (TBS-T, 0.05% Tween-20) and incubated with 40 µl of bacterial lysate for 30 min at room temperature. Beads were then washed with wash buffer, eluted by boiling in 50 µl of Laemmli buffer and analysed by western blot assay. For D-mannose blocking experiments,

D-mannose (M6020, Sigma-Aldrich) to a final concentration of 2.5% was added to the bacterial lysate. For quantification, band intensities from western blots were measured using ImageJ. The $rFimH_{LD}$-FLAG signal (FLAG blot) was normalized to the rPPAP-HA (HA blot) for each WT or mutant sample to account for potential differences in bait loading to the beads. Uncropped images of immunoblots are included as Source Data.

## $rFimH_{LD}$ binding assays

To confirm that $rFimH_{LD}$ could bind to host cells, we first used the 5637 cell line (HTB-9, ATCC) as control. Cells of the 5637 cell line were cultured in RPMI 1640 GlutaMAX (Thermo Fisher, 72400047). A day before the experiment, 18,000 cells per well were seeded on µ-Slide 18-well slides (81816, Ibidi). The next day, fresh medium containing 10% of the bacterial lysate ($rFimH_{LD}$) was incubated with the cells for 1 h at 37 °C. Then, cells were washed, fixed with 4% PFA, blocked with 1% BSA and stained with an anti-FLAG antibody (1:1,000; 66008-4-Ig, Proteintech) for 1 h at room temperature. Then, cells were incubated with anti-mouse IgG Alexa Fluor 594 (1:500; A21201, Invitrogen) for 1 h at room temperature. D-Mannose (M6020, Sigma-Aldrich) at a final concentration of 2.5% was added to the medium simultaneously with the bacterial lysate, as indicated. The same procedure was performed on the organoid-based model (DHT condition). In this case, organoid cells were seeded as described above. Nuclei were counterstained with Hoechst 33342 (1:5,000; H3570, Life Technologies).

For the incubation of the human prostate tissue slides, O.C.T-embedded tissue slides were rehydrated for 10 min in PBS at room temperature, blocked with 1% BSA and incubated with blocking buffer containing 10% protein lysate in the presence or absence of D-mannose (2.5%, M6020, Sigma-Aldrich) for 1 h at room temperature. Then, slides were washed and fixed in 4% PFA. Staining of $rFimH_{LD}$-FLAG was done as described for the µ-Slide 18-well slides above.

## Site-directed mutagenesis

Mutants of *ACPP* (*PPAP*) putative binding sites were generated by site-directed mutagenesis. A pair of complementary primers carrying the desired point mutation at positions 94 and 220 (94: 5′-AAGAGATATAGAAAATTCTTGGCTGAGTCCTATAAACATGAACAG-3′ and 5′-CTGTTCATGTTTATAGGACTCAGCCAAGAATTTTCTATATCTCTT -3′; 220: 5′-TATATTGTGAGAGTGTTCACGC-3′ and 5′-CAGGAGGGT AAAGTGAAAGC-3′) were used to introduce a point mutation that mutated the asparagine to alanine by PCR. All primer sequences are listed in Supplementary Table 4. This approach was unsuitable for generating the same point mutation for position 333 due to the high GC content of the sequence around the position. Hence, a DNA sequence containing the point mutation in position 333 was obtained from Integrated DNA Technologies (TGCGCATGACACTACTGTGAGTGGCCTACAGATGGCGC TAGATGTTTACAACGGACTCCTTCCTCCCTATGCTTCTTGCCACTT-GACGGAATTGTACTTTGAGAAGGGGGAGTACTTTGTGGAGATGTAC-TATCGGGCTGAGACGCAGCACGAGCCGTATCCCCTCATGCTACCTGGCT-GCAGCCCCAGCTGTCCTCTGGAGAGGTTTGCTGAGCTGGTTGGCCCT-GTGATCCCTCAAGACTGGTCCACGGAGTGTATGACCACAAACAGCCAT-CAAGTTCTAAAGGTCATCTTTGCTGTTGCCTTTTGCCTGATATCTGCT-GTCCTAATGGTACTACTGTTTATCCACATTCGCCGTGGACTCTGCTG-GCAGAGAGAATCCTATGGGAACATCGGGGGTGGAGGCTCTTATCCTTAC-GACGTGCCTGACTACGCCTAAACTCGAGTCTAGA). The sequence was cloned into the ACPP plasmids (WT or mutants 93, 220 or both) between the XbaI and FspI restriction sites. Correct plasmid DNA sequences were confirmed by Sanger sequencing.

## Bacterial binding to immobilized PPAP assays

For bacterial adhesion assays on immobilized PPAP-HA WT and mutants, cell lysates from PPAP-HA-expressing HeLa 229 cells were immobilized on PDL-coated glass slides (J2800AMNZ, Epredia) using a

modified protocol already described[71]. Briefly, the Grace Bio-Labs Pro-Plate microarray system for 64 wells (GBL246865, Grace Bio-Labs) was used on PDL-coated glass slides. Slides were incubated with 12.5 µg ml$^{-1}$ of anti-HA (51064-2-AP, Proteintech) or anti-LPS (ab35654, Abcam) antibody in array buffer (3% BSA in PBS) for 2 h at room temperature. Afterwards, wells were washed with array buffer and incubated with 50 µl of cell lysate overnight at 4 °C. Wells were then washed with array buffer before the addition of 50 µl (OD$_{600}$ = 0.05) of UPEC strain UTI89

WT or Δ*fimH* mutant, in the presence or absence of D-mannose (2.5%, M6020, Sigma-Aldrich). After incubation with bacteria, slides were washed, fixed with 4% PFA and imaged using the ECHO Revolve microscope. Numbers of bacteria attached to the slide were measured in four different images per experiment and condition using ImageJ software.

### Single-cell RNA-seq and data analysis

To generate single-cell suspensions, organoids and organoid-based model cells were washed with PBS and dissociated with TrypLE Express Enzyme (12605028, Gibco) for 15 min at 37 °C. Samples were washed with adDMEM/F12+/+, centrifuged at 400 × *g* for 5 min and resuspended in PBS with 0.1% BSA. Cell viability was measured with trypan blue. Only samples with over 90% viability were used. Cells from two mice (grown in control and DHT media) were multiplexed in two pools (2D and 3D organoids) using CellPlex technology (10x Genomics). Samples were resuspended in CellPlex Tag Buffer and incubated with individual CellPlex oligonucleotide tags for 15 min at room temperature. After labelling, cells were washed twice with PBS containing 0.04% BSA and pooled together in equal proportions. Cell concentration and viability were determined using an automated cell counter (Countess II, Thermo Fisher).

The pooled cells were subsequently loaded onto a Chromium Controller (10x Genomics) to generate Gel Bead-In-Emulsions (GEMs). Libraries were quantified using a Qubit dsDNA HS Assay kit (Thermo Fisher) and their size distribution was assessed using an Agilent 2100 Bioanalyzer. Sequencing libraries were then loaded onto an Illumina NovaSeq 6000 platform using an S2 flow cell with 100 cycles.

Raw sequencing data were demultiplexed, quality checked and subsequently aligned and quantified using the Cellranger software suite (10x Genomics, v.7.0.1) against the mm10 (Ensembl98) mouse genome assembly as reference.

Obtained count matrices of 3D organoids and 2D models were loaded into R (v.4.3.0) and analysed analogously using the Seurat R package (v.4.3.0). Cells were quality filtered on the basis of standard quality metrics, including the number of detected transcripts per cell (>10,000 2D, >5,000 3D organoids), the number of detected genes per cell (>200 < 9,000 2D, >200 < 8,000 organoids) and mitochondrial count fraction (<10%). Raw counts were normalized and log transformed (NormalizeData, default settings). Highly variable genes (HVGs) were identified (FindVariableFeatures, using nfeatures = 3,000) and scaled (ScaleData, default settings). Principal component analysis (PCA) was performed on the basis of HVGs (RunPCA, default settings). The uniform manifold approximation and projection (UMAP) algorithm was used to compute a two-dimensional embedding for visualization based on the 30 first principal components (RunUMAP, dims = 1:30). A nearest-neighbour graph was constructed (FindNeighbors, dims = 1:30), and unsupervised clustering was performed using the Louvain algorithm (FindClusters) with the resolution parameter set to 0.4. Cell types were assigned on the basis of the expression of known marker genes (Luminal: *Krt8*, *Krt18*, *Cd24a*, *Psca*, *Ly6a*; Basal: *Krt5*, *Trp63*, *Krt14*; Proliferating: *Mki67*, *Birc5*).

The neighbourhood graph correlation-based similarity analysis[72] was applied to compare cell states in the 2D and 3D scRNA-seq data to an scRNA-seq primary mouse prostate tissue reference. To construct a primary tissue reference, publicly available data from refs. [25,27] were obtained, loaded into R and analysed using Seurat. Low-quality transcriptomes (<1,000 transcripts per cell; <300 and >5,000 genes

per cell; >10% mitochondrial count fraction) were removed. Gene counts were log normalized and 3,000 HVGs were selected. Datasets were integrated by applying the Seurat reciprocal principal component analysis (RPCA) workflow. Nearest-neighbour graph construction and computation of a two-dimensional UMAP embedding were based on the 30 first RPCA components. The Louvain algorithm was used for unsupervised clustering at a resolution of 0.7. Cell types were assigned on the basis of marker genes curated from refs. [25,27]. Independent neighbourhood graphs based on transcriptional similarity were constructed for each condition (2D Ctrl, 2D DHT, 3D Ctrl, 3D DHT), as well as the reference using the miloR package (v.1.6.0)[73]. For the 2D and 3D organoid data, *k* = 20 nearest neighbours and 30 PCA dimensions, and for the reference, *k* = 50 nearest neighbours and 50 RPCA dimensions were used. The gene expression profile of each neighbourhood was calculated as the mean expression of the genes within all cells constituting a neighbourhood. For each neighbourhood, the Pearson correlation of gene expression with all neighbourhoods in the reference was computed using the scrabbitr R package (v.0.1.0)[72]. Pearson correlation calculations were performed on the 3,000 most variable genes identified within basal and luminal cells in the reference. Similarity was determined as the maximum correlation between each neighbourhood in a 2D/3D condition and a reference neighbourhood.

### *Acpp* (*Ppap*) knockout generation

Mouse prostate organoids were edited with the CRISPR–Cas9 system using a previously described protocol[42]. Briefly, two guide RNAs against *Acpp* (sgRNA 1: 5′-TGAGCCGGACAGCAAGCCTC-3′, sgRNA 2: 5′-GAGCCGGACAGCAAGCCTCA-3′) were designed, using Benchling (www.benchling.com), to target the first exon of *Acpp*. gRNAs were cloned into BbsI linearized pSpCas9(BB)-2A-Puro (PX459) V2.0 plasmid (Addgene, 62988). Then the plasmid was transfected into the organoid cells via electroporation using the protocol from ref. 74. Briefly, 2 days before the electroporation, Primocin was removed and Noggin-conditioned medium was replaced with recombinant Noggin (100 ng ml$^{-1}$, 250-38, Peprotech). A day before the electroporation, mouse prostate organoids were treated with 1.25% dimethylsulfoxide. On the day of the electroporation, organoids were shredded mechanically and made into single cells using TrypLE Express Enzyme (12605036, Gibco) as described above. For each plasmid, a total of 1 × 10$^5$ cells were resuspended in 100 µl of BTXpress buffer (45-0805, BTX Molecular Delivery Systems) supplemented with RHOKi (Y-27632; 10 µM, M1817, AbMole). Cells were then electroporated with 20 µg of pSPCas9(BB)−2A-Puro V2.0 plasmid (Addgene, 62988), containing either gRNA1 or gRNA2, in a nucleofection cuvette (EC-002S, Nepagene) using the NEPA 21 Super Electroporator (Nepagene) with the following settings: poring pulse: 175 V, pulse length 5 ms, pulse interval 50 ms, number of pulses 2, decay rate 10% and positive polarity; transfer pulse: 20 V, pulse length 50 ms, pulse interval 50 ms, number of pulses 5, decay rate 40% and positive/negative polarity.

After electroporation, cells were washed with Opti-MEM (31985070, Gibco) containing RHOKi (Y-27632; 10 µM, M1817, AbMole) and seeded in Matrigel (356231, Corning). The organoid culture medium was supplemented with RHOKi (Y-27632; 10 µM, M1817, AbMole) and DHT (1 nM, A8380, Sigma-Aldrich). *Acpp*-KO organoids were selected with puromycin (1 µg ml$^{-1}$, sc-108071B, Santa Cruz Biotechnologies) for 72 h 1 day after electroporation. Individual organoids were isolated and expanded until molecular characterization. Genomic DNA from potential KO organoid lines was isolated and used to amplify a fraction of *Acpp* containing the sgRNAs (sgRNA 1: 5′-TGAGCCGGACAGCAAGCCTC-3′, sgRNA 2: 5′-GAGCCGGACAGCAAGCCTCA-3′) using the following primer pair: forward 5′-AGGCTATGCATCACCATCCG-3′; reverse 5′-GCTGCCTGTTCTCCTGAAGT-3′. *Acpp* KO was confirmed by Sanger sequencing (5′-ACCTGCCAACCAAAGCGAGAAC-3′) and hybridization chain reaction RNA-fluorescence in situ hybridization (HCR RNA-FISH) (see Supplementary Table 4).

## HCR RNA-FISH

HCR 3.0 Probe Maker v.0.3.2 Jupyter Notebook, developed by ref. 75 and publicly available for non-commercial use at https://github.com/rwnull/insitu_probe_generator, was utilized to design the HCR RNA-FISH 3.0 Probes. The sequence of the relevant RNA was used as input by the Python 3 notebook. The mouse genome was blasted with split probes to decrease off-target alignment. Splitprobe couples were eliminated from the pool if the second-highest match was on the same gene and the splitprobe had a match rate greater than 60%. The target sequences were obtained from NCBI (*Acpp*, NM_207668.2).

DNA probe sets were reconstituted to a final concentration of 1 µM in TE-Buffer (12090015, Invitrogen). HCR RNA-FISH was performed according to the protocol for mammalian cells from Molecular Instruments, and HCR Amplifier and HCR Buffers sets (Molecular Instruments) were used. Briefly, cells were fixed with 4% PFA for 15 min and then overnight permeabilized with 70% ethanol. After 30 min of prehybridization at 37 °C (incubation with hybridization buffer; HCR Buffers set, Molecular Instruments), hybridization took place overnight with a 1:100 dilution of the probe in hybridization buffer (HCR Buffers set, Molecular Instruments). Samples were washed with heated wash buffer (HCR Buffers set, Molecular Instruments) and three washes with 5× SCCT (20× sodium chloride sodium citrate; 7732-18-5, Invitrogen; +0.1% Tween in double-distilled $H_2O$). Samples were pre-incubated with amplification buffer (HCR Buffers set, Molecular Instruments) for 30 min. The preferred amplifier (B1 - 488, B2 - 594, B3 - 647; HCR Amplifier set, Molecular Instruments) was snap cooled to 95 °C and allowed to cool down in the dark for 30 min, then diluted 1:100 in amplification buffer (HCR Buffers set, Molecular Instruments). Amplification was carried out overnight at room temperature. The samples were stained with DAPI (62247, Thermo Fisher; 1:1,000) in 5× SCCT and rinsed in 5× SCCT. Confocal microscopy was used for imaging, as described above.

## Statistical analysis

Unless otherwise indicated, data are presented as mean ± s.d., with the exact number of experiments performed indicated in figure legends. Statistical analysis was performed using GraphPad Prism. Normality of data distribution was assessed using the Shapiro–Wilk test. For statistical comparison of datasets from two conditions, two-tailed Student's *t*-test was used; for data from three or more conditions/groups, one-way analysis of variance (ANOVA) with Tukey's or Dunnett's post hoc test was used. Statistical analyses and *P* values are detailed in Supplementary Table 6.

## Reporting summary

Further information on research design is available in the Nature Portfolio Reporting Summary linked to this article.

## Data availability

scRNA-seq data samples are publicly available in the GEO repository database under the accession number GSE275482. DNA sequences of *fimH* are publicly available in the GenBank BankIt repository under the submission ID 3019276 with accession numbers: UTI89_FimH PX570005, CP1_FimH PX570006, P10_FimH PX570007 and P16_FimH PX570008. Mass spectrometry data are provided in Supplementary Table 5. Any other relevant data are available from the corresponding author. Source data are provided with this paper.

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

## Acknowledgements

This work was supported by the BMFTR (FiRe-UPec, 01KI2107 to C.A.), the Deutsche Forschungsgemeinschaft (DFG GRK 2157; 3D Tissue Models for Studying Microbial Infections by Human Pathogens, Project 11, to C.A., Project 12 to A-E.S) and the Single-Cell Center Würzburg (Seed grant to C.A.). A-E.S and A.M.L. were supported by the DFG-funded project CRC1583 DECIDE—Decisions in Infectious Diseases; project Z02. U.D. was supported by the Interdisciplinary Centre for Clinical Research Münster (grant no. Dob2/010/22). We thank P. Thumbikat (Northwestern University) for the CP1 strain; F. Imdahl (Single-Cell Center Würzburg) for support with scRNA-seq experiments; N. Schall and F. Heim (Aguilar Lab) for general technical support; N. Burkard and N. Schlegel (University Hospital Würzburg) for support with ECIS1600R equipment; and S. Lamer (University of Würzburg) for mass spectrometry analysis. We thank the tissue donors, without whom this work would not have been possible. We acknowledge critical reading by V. Thacker (Heidelberg University) and M. Alzheimer (University of Würzburg). Schematics were created with BioRender.com.

## Author contributions

C.A. conceived of the study. C.A. wrote the paper with the input of all authors. M.G., S.P., A.J., S.D., J.R. and K.K. performed most of the experiments. T.B. performed the RNA-FISH experiments under the supervision of A-E.S. and C.A.; A.M.L. provided support in the analysis of scRNA-seq data under the supervision of A-E.S.; M.R. and C.K. provided technical and material support with the human prostate tissue; U.D. provided technical and material support with the UPEC mutant. All authors contributed to the revision of the paper and approved the final version.

## Funding

## Competing interests

The authors declare no competing interests.

## Additional information

**Extended data** is available for this paper at https://doi.org/10.1038/s41564-025-02231-0.

**Correspondence and requests for materials** should be addressed to Carmen Aguilar.

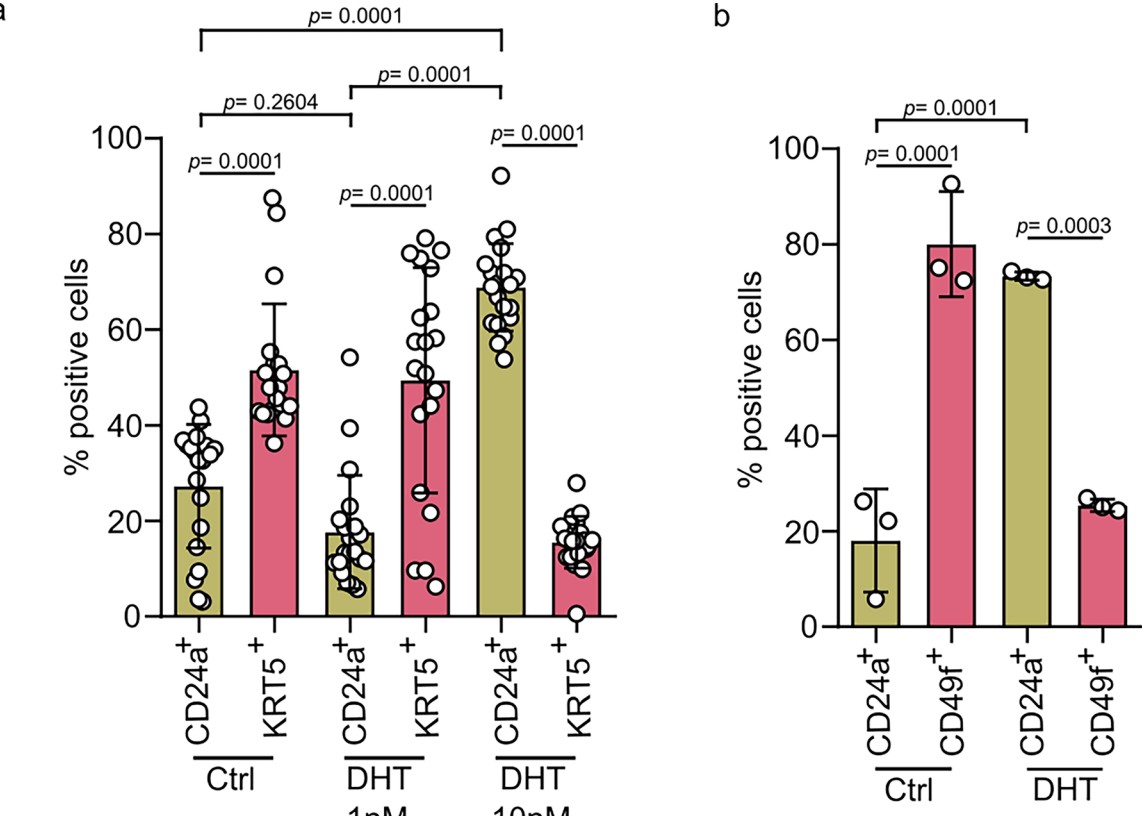

**Extended Data Fig. 1 | 10 nM DHT induces luminal differentiation in the organoid-based model. a.** Image quantification of CD24a⁺ (khaki, luminal cells) and KRT5⁺ (pink, basal cells) cells in the organoid-based model grown with no DHT (Ctrl), 1 nM or 10 nM. Data are shown as % of positive cells per field of view. Representative images are shown in Fig. 1b. *n* = 3 independent experiments, 3 organoid lines; 2–3 images were quantified per condition. **b.** Flow cytometry quantification of CD24a⁺ (khaki, luminal cells) and CD49f⁺ (pink, basal cells) cells in the organoid-based model grown with no DHT (Ctrl) or 10 nM DHT (DHT). *n* = 3 biological replicates. Data are presented as mean values ± s.d., and a one-way ANOVA with Tukey's post-hoc test was used in both panels.

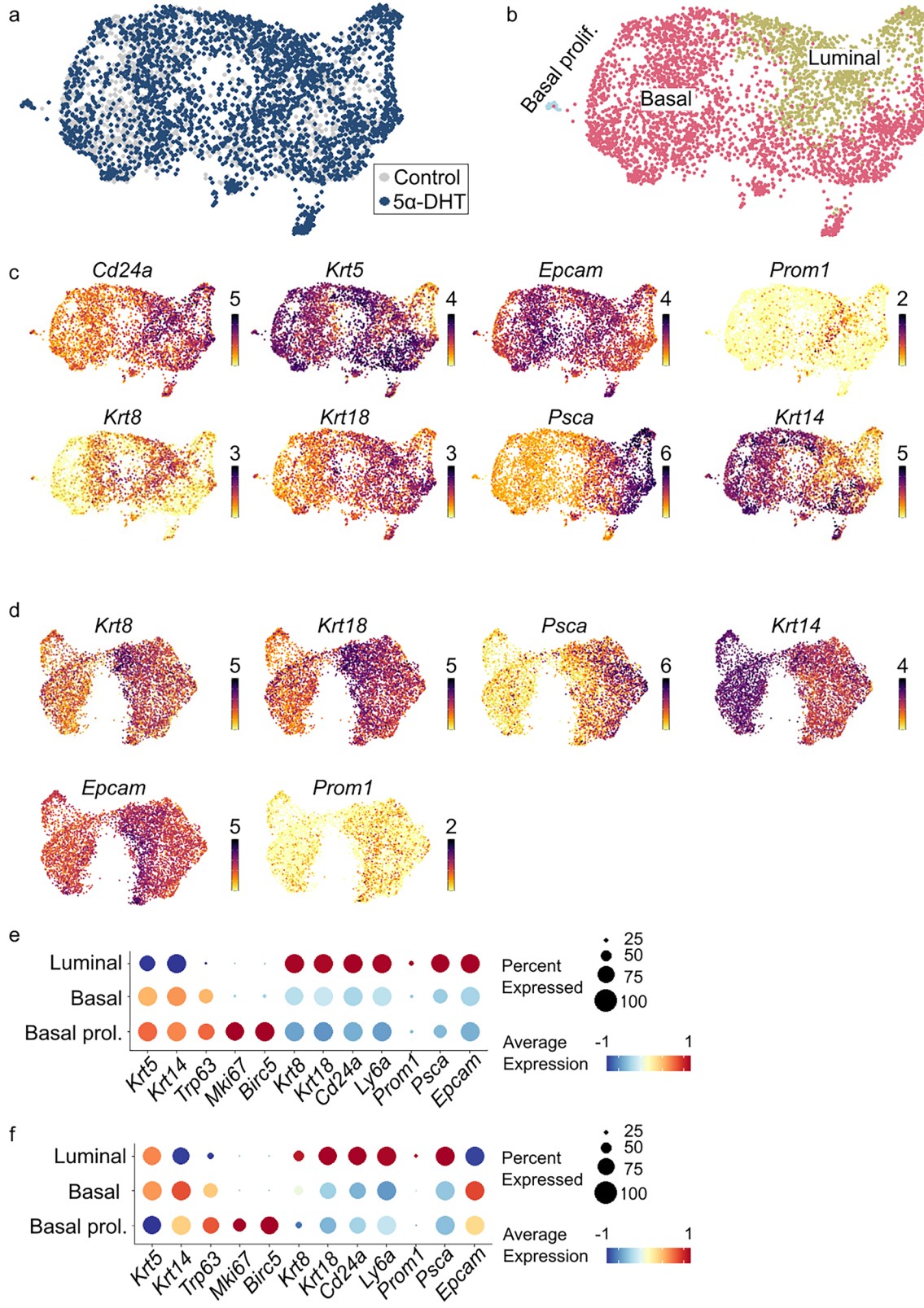

**Extended Data Fig. 2 | Prostate 3D organoids grown in the absence or presence of DHT (10 nM) contain a high number of basal prostate cells, in contrast to the 2D organoid-based model. a.** Single-cell transcriptomes from the 3D organoids grown in control (grey; 1,471 cells) or DHT (10 nM, dark blue; 2,792 cells) medium were integrated and projected using Uniform Manifold Approximation and Projection (UMAP). **b.** Using the same prostate cell type marker genes as the 2D organoid-based model, cell identities were assigned into three clusters (see **Methods** section). **c.** Expression level of markers specific for luminal prostate cells (*Cd24a, Krt8, Krt18, Psca, Epcam*, and *Prom1*) and basal cells (*Krt5, Krt14*) colour-coded and projected on the UMAP displayed in panels (**a**) and (**b**). **d.** Expression level of markers specific for luminal prostate cells (*Krt8, Krt18, Psca, Epcam, and Prom1*) and basal cells (*Krt14*) colour-coded and projected on top of the UMAP projection of 2D organoid-based model scRNA-seq data as displayed in Fig. 1c, e. **e-f.** Dot plot representation of Z-score of selected marker gene expression on 2D organoid-based model (**e**) and 3D organoids (**f**).

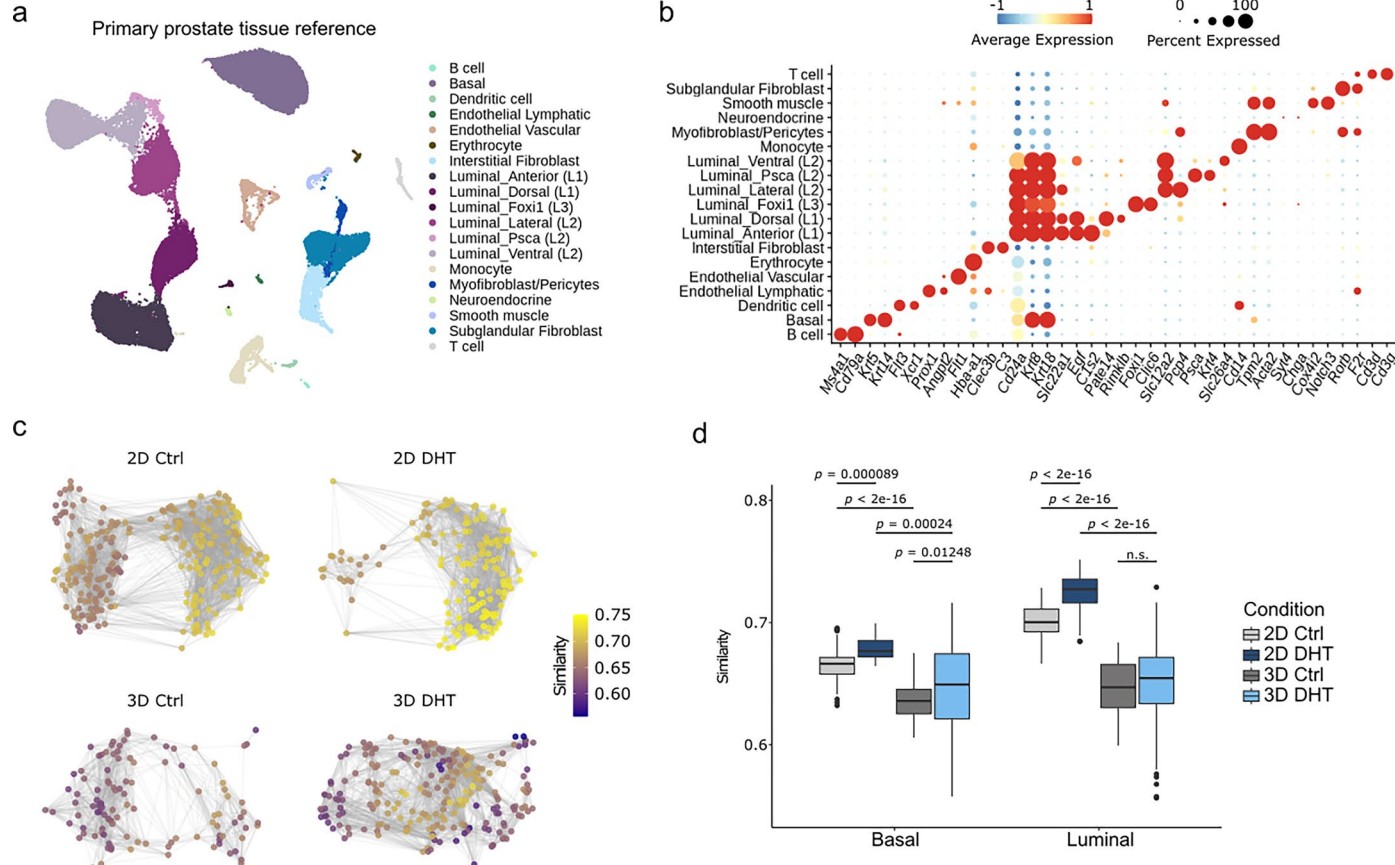

**Extended Data Fig. 3 | 2D prostate organoid-based model treated with DHT shows the highest similarity to primary prostate tissue on single-cell transcriptomic level. a**. UMAP representation of scRNA-seq transcriptomic data showing the cell types identified in primary prostate tissue scRNA-seq data from two publicly available datasets (Graham et al. 2024, Karthaus et al. 2020). **b**. Dot plot showing scaled mean expression (colour) and percentage of expressing cells (dot size) of selected marker genes used to delineate cell types in the primary tissue reference. **c**. Neighbourhood graphs of the four conditions profiled using scRNA-seq (2D Ctrl, 2D DHT, 3D Ctrl, 3D DHT) coloured by the maximum correlation value across primary prostate reference neighbourhoods. Neighbourhoods are positioned with respect to the UMAP embedding coordinates of the respective index cell. 2D neighbourhoods were constructed from cells across the two replicates in each condition. **d**. Box plot depicting neighbourhood similarities from (**c**) in basal and luminal neighbourhoods of 2D/3D. Centerline indicates the median, box limits indicate the upper and lower quantiles, and the whiskers indicate the 1.5x interquartile range. *n* = 2 biological replicates. A Wilcoxon-rank-sum test was used to assess significance.

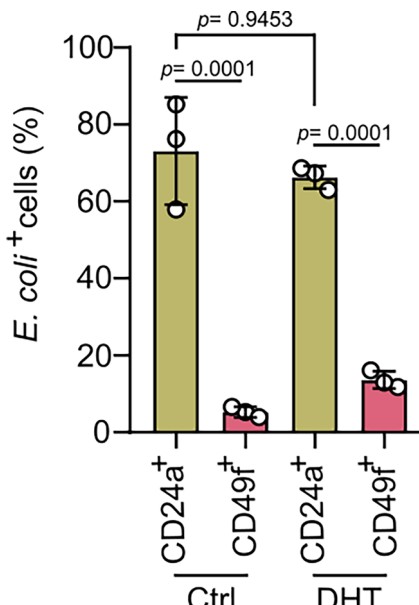

**Extended Data Fig. 4 | Flow cytometry quantification of *E. coli* isolate CP1-infected cells.** Same as in Fig. 2b for quantification of UTI89-infected cells, cells were first gated into luminal (CD24a⁺, khaki) and basal (CD49f⁺, pink) populations. The percentage of mCherry⁺ (infected) cells was then quantified within each population. *n* = 3 biological replicates. Gating strategy is shown in Supplementary Figure 5. Data are presented as mean values ± s.d., and a one-way ANOVA with Tukey's post-hoc test was used.

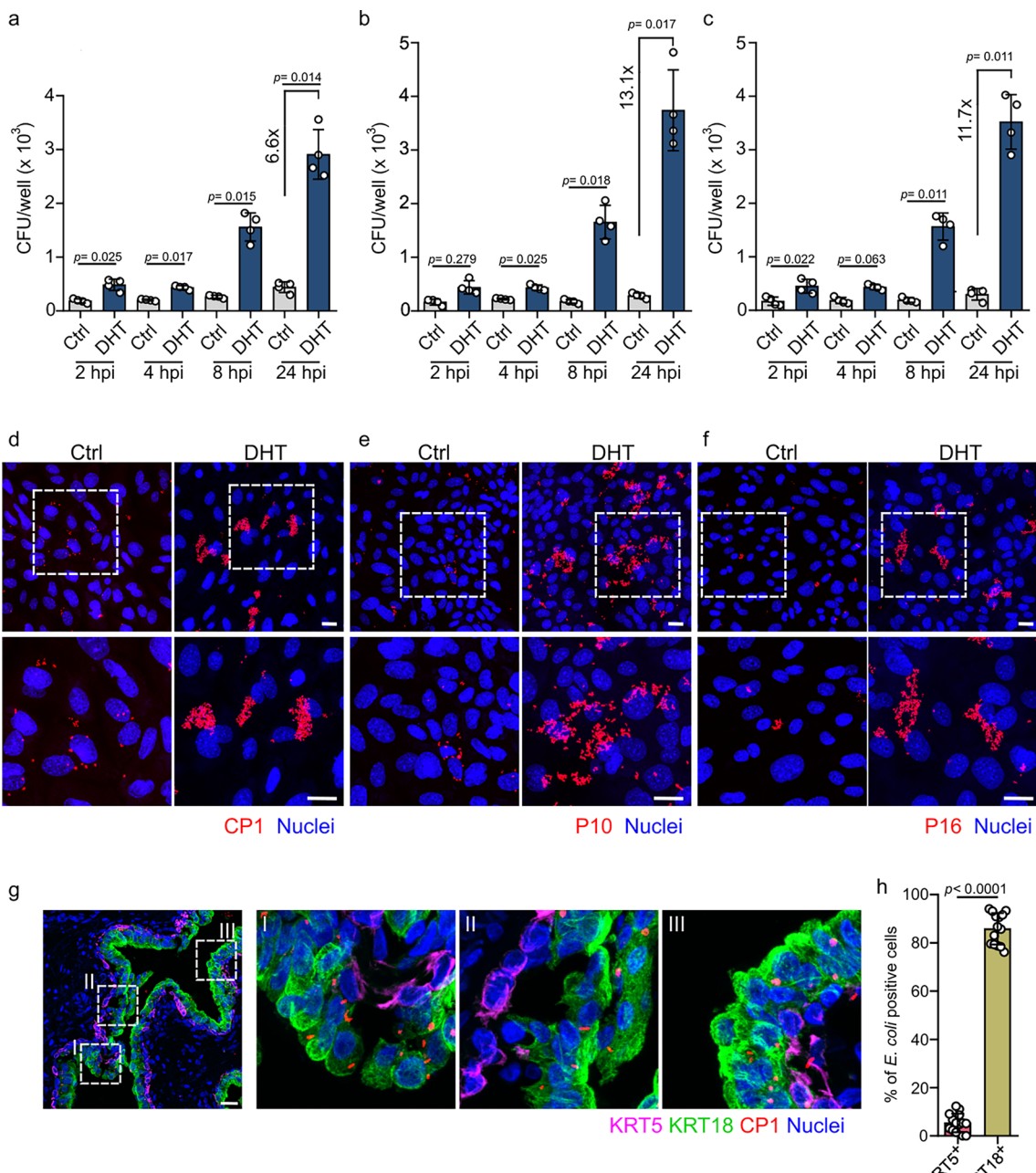

**Extended Data Fig. 5 | *E. coli* prostatitis isolates CP1, P10 and P16 preferentially invade and replicate in cells within the 2D organoid-based model grown in the presence of DHT. a-c.** *E. coli* CP1 (**a**), P10 (**b**), and P16 (**c**) CFUs were quantified at 2, 4, 8, and 24 hpi in the organoid-based model grown with and without DHT (n = 4 biological replicates). Data are presented as mean values ± s.d., and a one-way ANOVA with Tukey's post-hoc test was used. **d-f.** Representative confocal images of the organoid-based model 24 hpi (**d**, CP1; **e**, P10; **f**, P16). Bacteria is shown in red. Scale bar: 25 μm. *n* = 4 biological replicates.

**g-h.** Representative confocal images (**g**) and image quantification graph (**h**) of human prostate tissue incubated with *E. coli* CP1 and stained for KRT5 (magenta) and KRT18 (green). Nuclei were counterstained using Hoechst 33342 (blue). Scale bar: 25 μm. *n* = 2 independent experiments, 3 donors; 2–3 images were quantified per condition. Data in panel **h** are shown as % of infected cells per field of view and are presented as mean values ± s.d. A two-tailed Student's t-test was used. Absolute numbers are shown in Extended Data Fig. 6c.

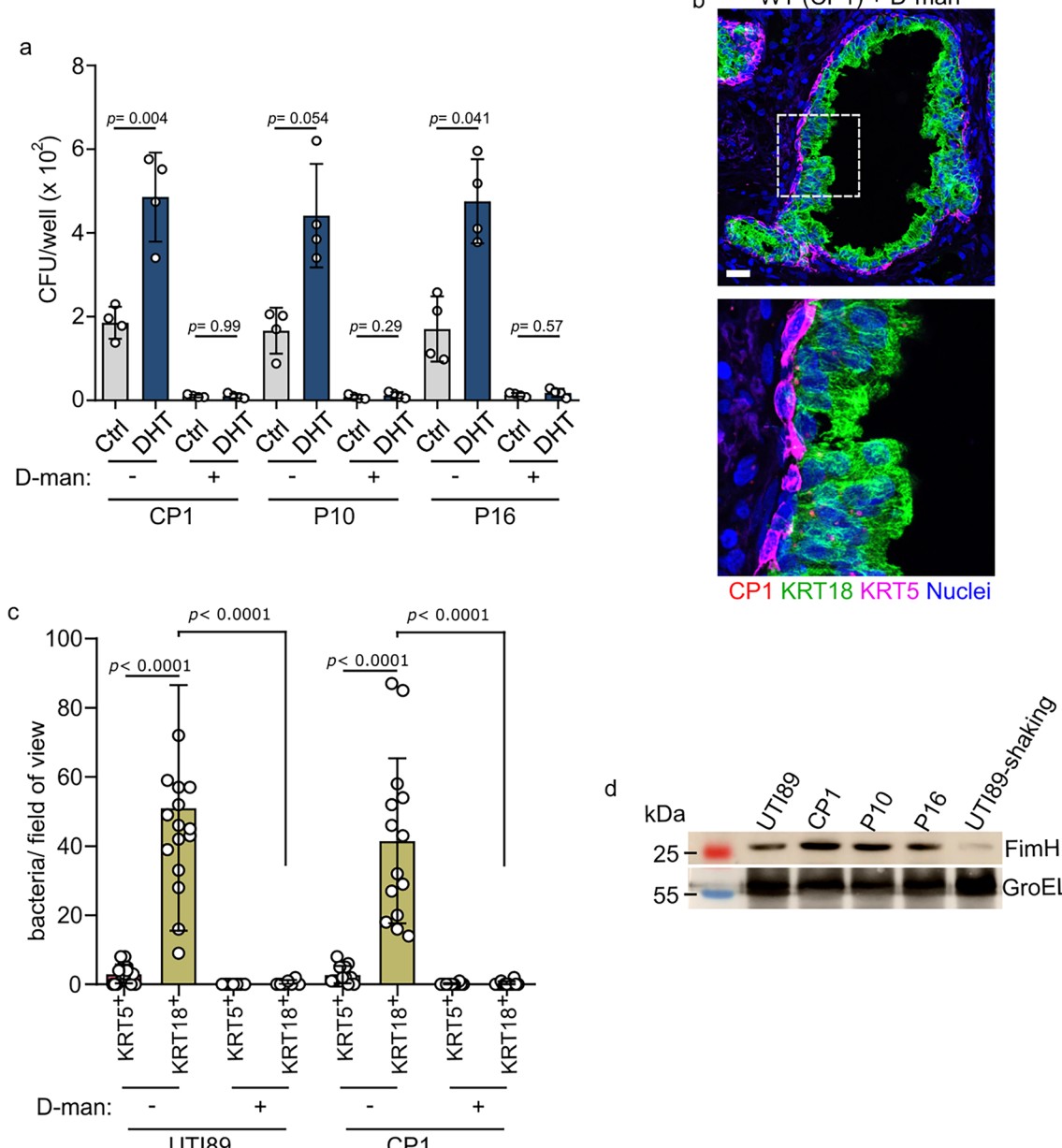

**Extended Data Fig. 6 | D-mannose blocks *E. coli* interaction with prostate cells. a**. CFUs of the prostate organoid-based model infected with *E. coli* CP1, P10, and P16 in the absence or presence of D-mannose ($n$ = 4 biological replicates, two-tailed Student's t-test). **b**. Representative confocal image showing binding of CP1 to human prostate tissue in the presence of D-mannose. Bacteria are shown in red, KRT18 in green, KRT5 in magenta, and nuclei in blue. Scale bar = 25 μm. Data are representative results from 3 donors and 2 independent experiments.

**c**. Image quantification of infected cells (absolute numbers) per field of view ($n$ = 2 independent experiments, 3 donors; 2–3 images were quantified per condition, one-way ANOVA with Tukey's post-hoc test). Data in panels **a** and **c** are presented as mean values ± s.d. **d**. Western blot analysis of FimH expression in UTI89, CP1, P10 and P16 in culture conditions pre-*in vitro* infection (24 h static), UTI89 in shaking conditions (reduced FimH expression) was used as control. GroEL was used as loading control ($n$ = 1).

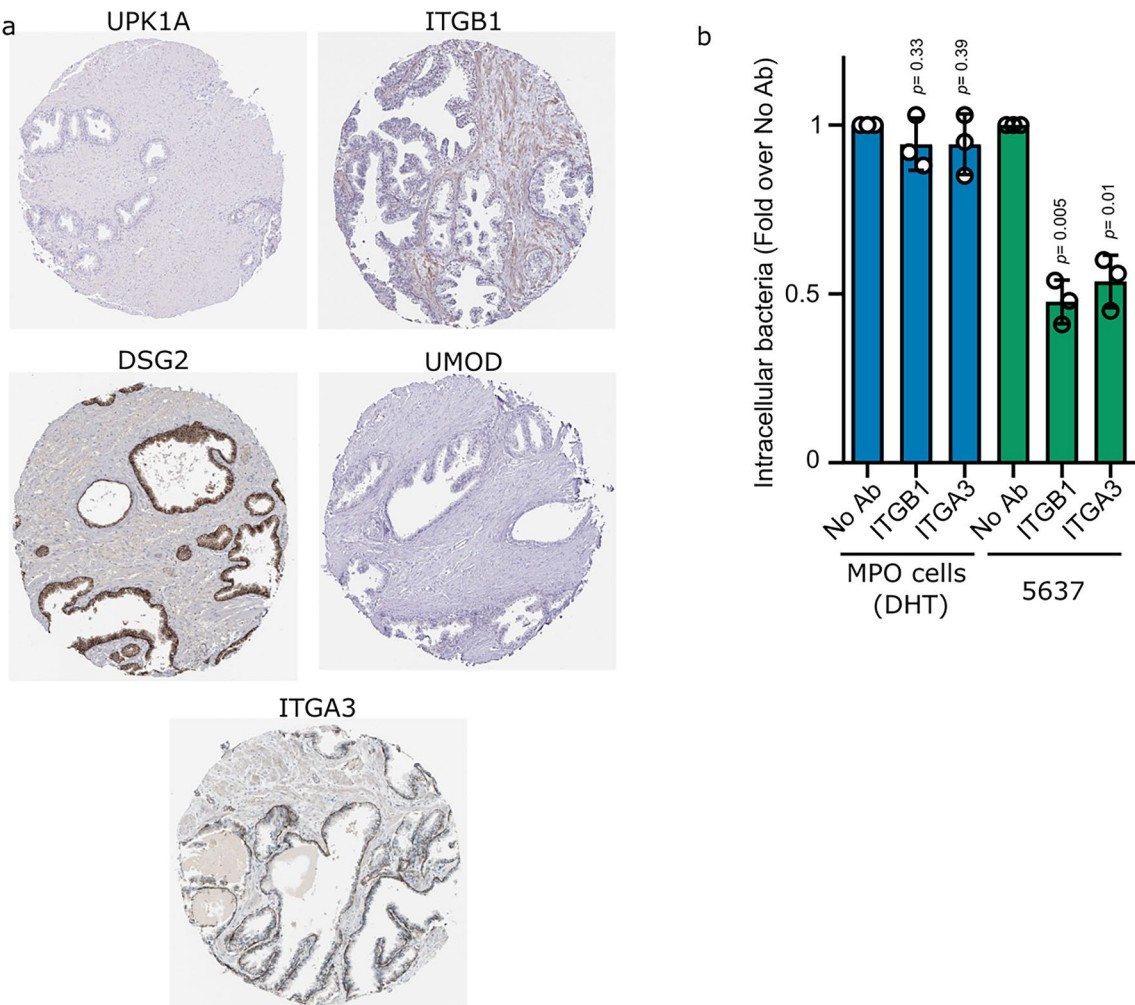

**Extended Data Fig. 7 | ITGB1 and ITGA3 blocking in murine prostate cells does not affect UPEC invasion. a**. Immunohistochemistry analysis of UPK1A, ITGB1, DSG2, UMOD, and ITGA3 protein expression in human prostate tissue (Data from The Human Protein Atlas Database). **b**. Quantification of intracellular bacteria after blocking ITGB1 and ITGA3 with antibodies in the organoid-based model (MPO: Mouse prostate organoids, DHT) or the bladder cell line 5637. PBS with no antibody was used as control (No Ab; $n$ = 3 biological replicates). Data are presented as mean values ± s.d., and a two-tailed Student's t-test was used.

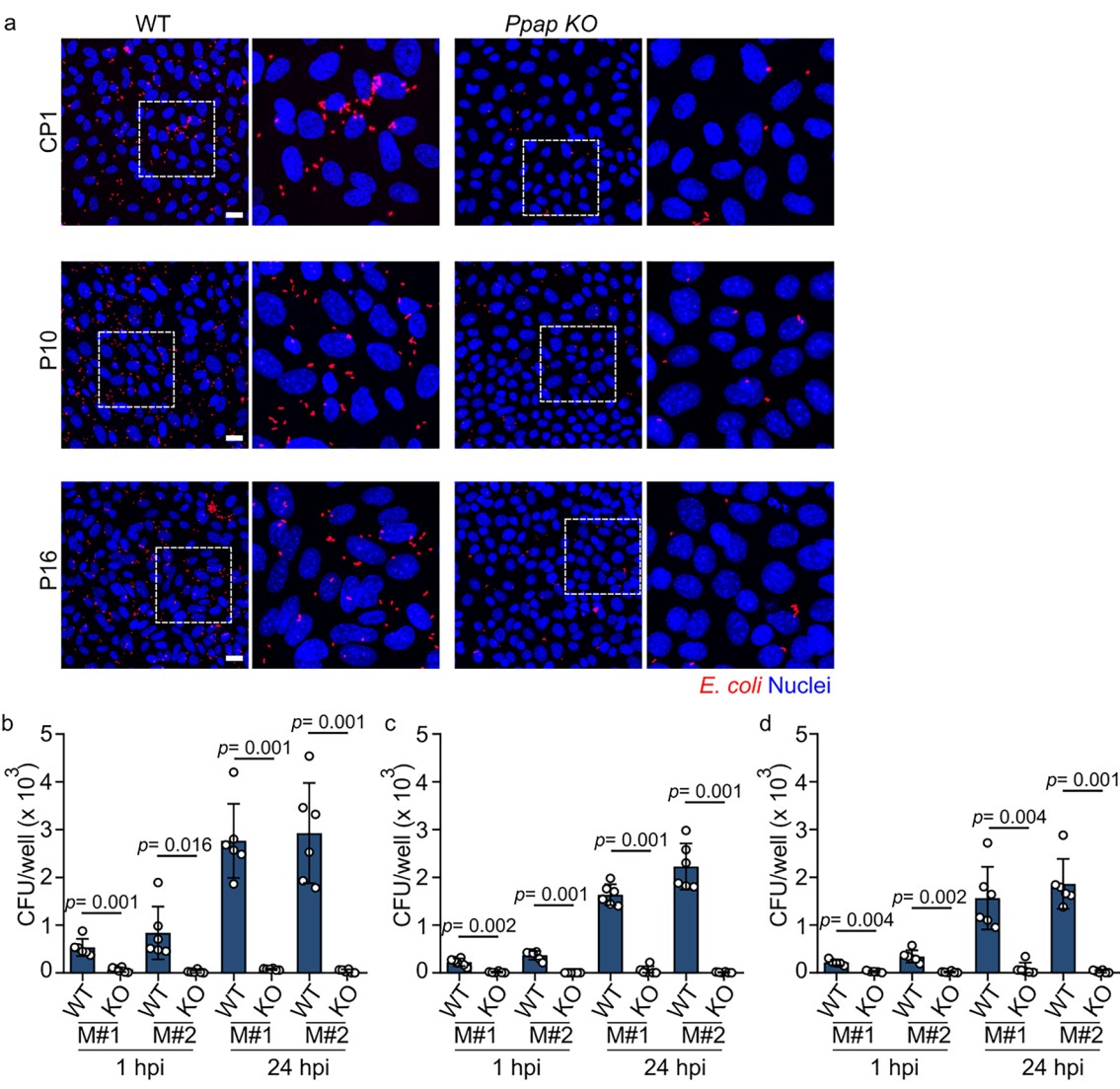

**Extended Data Fig. 8 | PPAP is necessary for maximal invasion of _E. coli_ CP1, P10 and P16 into prostate cells. a**. Representative confocal microscopy images of _Ppap_ KO mouse organoid cells (_Ppap_ KO M#1, grown in 10 nM DHT) infected with CP1, P10 or P16 (red) for 1 h. Nuclei were counterstained using Hoechst 33342 (blue). Scale bar: 25 μm. _n_ = 4 biological replicates. **b-d**. CFU quantification of two _Ppap_ KO mouse organoid lines infected with CP1 (**b**), P10 (**c**) or P16 (**d**) for 1 h and 24 h. _n_ = 6 biological replicates. Data are presented as mean values ± s.d., and a two-tailed Student's t-test was used.

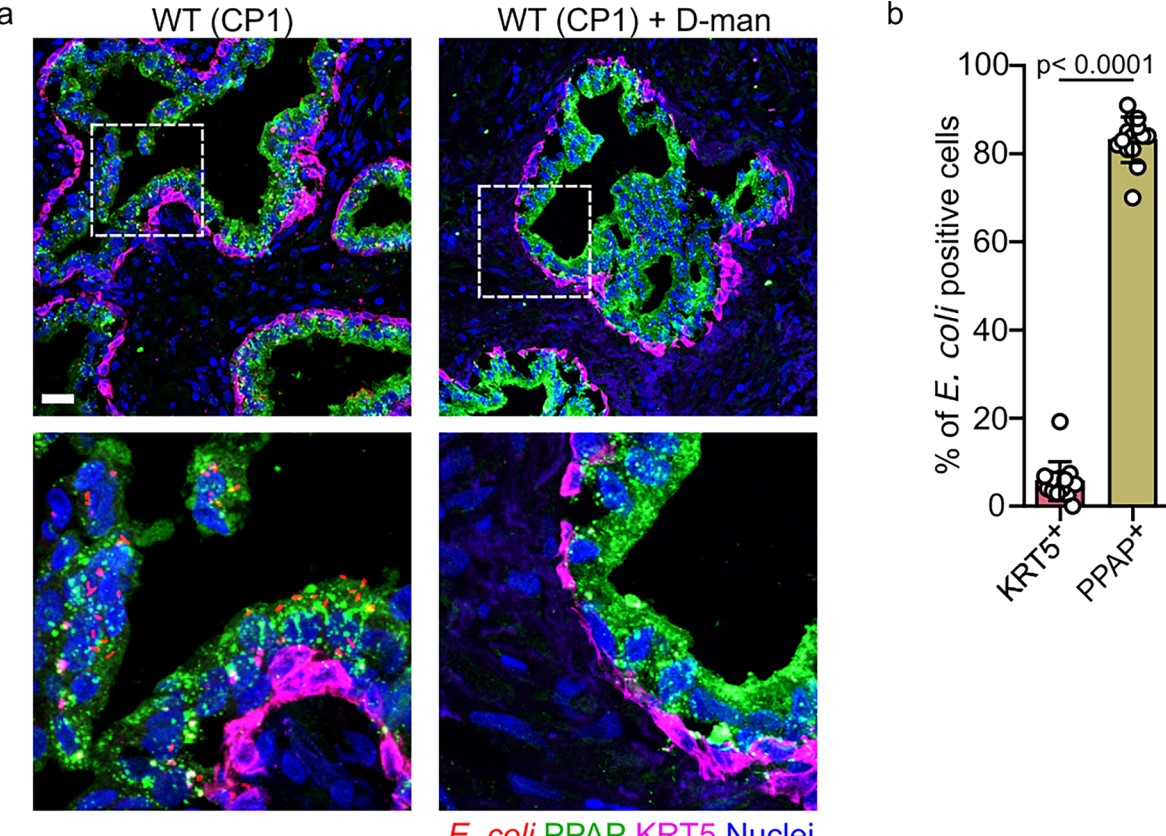

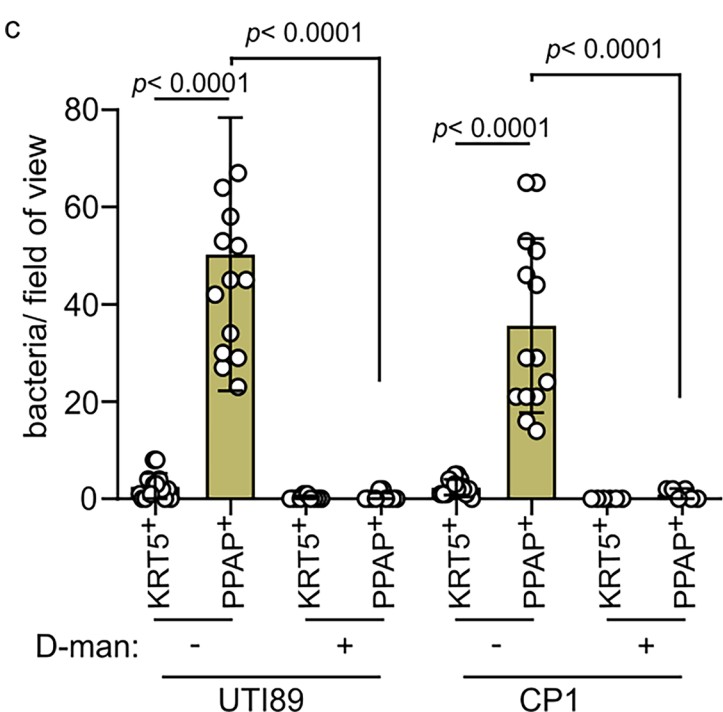

**Extended Data Fig. 9 | *E. coli* CP1 colocalises with PPAP-positive cells in human prostate tissue. a**. Representative confocal microscopy images showing the binding of CP1 in the absence (left image) or presence (right image) of 2.5% D-mannose to human prostate tissue. Bacteria are shown in red, PPAP in green and KRT5 in magenta. Nuclei were counterstained using Hoechst 33342 (blue). Scale bar: 25 μm. *n* = 2 independent experiments, 3 donors. **b**. Image quantification graph of WT CP1 binding. Data are shown as % of infected cells per field of view. Absolute numbers are shown in **c**. Data in panels **b** and **c** are presented as mean values ± s.d. *n* = 2 independent experiments, 3 donors; 2–3 images were quantified per condition (two-tailed Student's t-test in panel **b**, and a one-way ANOVA with Tukey's post-hoc test in panel **c**).

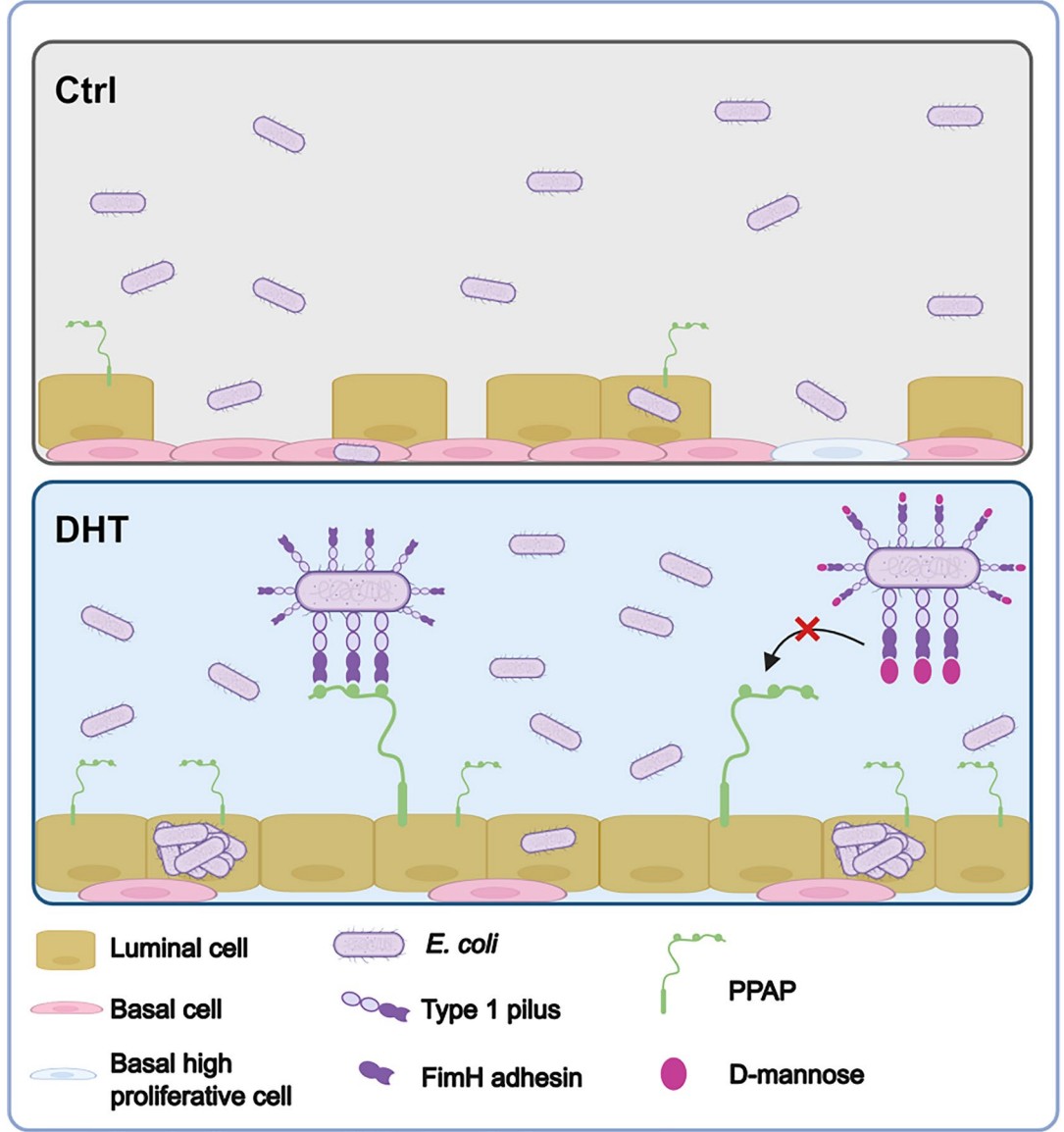

**Extended Data Fig. 10 | Model depicting the role of the PPAP protein in _E. coli_ infection of prostate cells.** In the organoid model exposed to 10 nM DHT, cells differentiate into luminal cells, while in the absence of DHT (control), they remain as basal cells. _E. coli_ targets prostate luminal cells by using FimH to bind to the PPAP receptor on their surface, facilitating bacterial invasion. D-mannose can inhibit this process by blocking FimH binding to PPAP, thereby preventing invasion. Created in BioRender. Aguilar, C. (2025) https://BioRender.com/m5hkkqn.

# Reporting Summary

## Statistics

For all statistical analyses, confirm that the following items are present in the figure legend, table legend, main text, or Methods section.

| n/a | Confirmed | |
|---|---|---|
| ☐ | ☒ | The exact sample size (*n*) for each experimental group/condition, given as a discrete number and unit of measurement |
| ☐ | ☒ | A statement on whether measurements were taken from distinct samples or whether the same sample was measured repeatedly |
| ☐ | ☒ | The statistical test(s) used AND whether they are one- or two-sided<br>*Only common tests should be described solely by name; describe more complex techniques in the Methods section.* |
| ☒ | ☐ | A description of all covariates tested |
| ☐ | ☒ | A description of any assumptions or corrections, such as tests of normality and adjustment for multiple comparisons |
| ☒ | ☐ | A full description of the statistical parameters including central tendency (e.g. means) or other basic estimates (e.g. regression coefficient) AND variation (e.g. standard deviation) or associated estimates of uncertainty (e.g. confidence intervals) |
| ☐ | ☒ | For null hypothesis testing, the test statistic (e.g. *F*, *t*, *r*) with confidence intervals, effect sizes, degrees of freedom and *P* value noted<br>*Give P values as exact values whenever suitable.* |
| ☒ | ☐ | For Bayesian analysis, information on the choice of priors and Markov chain Monte Carlo settings |
| ☒ | ☐ | For hierarchical and complex designs, identification of the appropriate level for tests and full reporting of outcomes |
| ☒ | ☐ | Estimates of effect sizes (e.g. Cohen's *d*, Pearson's *r*), indicating how they were calculated |

*Our web collection on statistics for biologists contains articles on many of the points above.*

## Software and code

Policy information about availability of computer code

| | |
|---|---|
| Data collection | Leica Application Suite Advance Fluorescence (LAS AF Leica Microsystems, v.2.7.3.9723), LAS X (v4.6.1.27508), ImageQuant LAS 4000, NovaSeq 6000 (Illumina), NovoCyte Quanteon flow cytometer. |
| Data analysis | Microsoft Excel (Microsoft Office, 2013), Prism (GraphPad, v.7.00), Inkscape (v1.3.2.), ImageJ (v.1.50b), R (v4.3.0), Rstudio (v3.53), Phyton v3.8, LAS AF Lite (Leica, 2.6.0 build 7266), ImageQuant LAS 4000 Control Software, Cell Ranger (version v7.0.1.2), Seurat (v4.3.0), BioRender, NovoExpress software (Agilent). |

For manuscripts utilizing custom algorithms or software that are central to the research but not yet described in published literature, software must be made available to editors and reviewers. We strongly encourage code deposition in a community repository (e.g. GitHub). See the Nature Portfolio guidelines for submitting code & software for further information.

## Data

Policy information about availability of data

All manuscripts must include a data availability statement. This statement should provide the following information, where applicable:

- Accession codes, unique identifiers, or web links for publicly available datasets
- A description of any restrictions on data availability
- For clinical datasets or third party data, please ensure that the statement adheres to our policy

> The data supporting the findings of this study have been deposited in Gene Expression Omnibus with the accession code GSE275482. All other relevant data are available from the corresponding author upon request.

## Research involving human participants, their data, or biological material

Policy information about studies with human participants or human data. See also policy information about sex, gender (identity/presentation), and sexual orientation and race, ethnicity and racism.

| | |
|---|---|
| Reporting on sex and gender | Only male patients tissue was used in this study. Ages 45-80. |
| Reporting on race, ethnicity, or other socially relevant groupings | Due to the low number of male patients used in this study, no discrimination based on age, race or genotypic information was applied. Samples from three donors were used (45-80 years old) were used. No financial compensation was provided. |
| Population characteristics | N/A |
| Recruitment | Patients that had planned a prostate surgery (e.g. biopsy for cancer screening or proctectomy) were informed by the doctor. Informed consent was obtained from all the donors. |
| Ethics oversight | Ethical committee of the University of Würzburg (Approval 168/22). |

Note that full information on the approval of the study protocol must also be provided in the manuscript.

# Field-specific reporting

Please select the one below that is the best fit for your research. If you are not sure, read the appropriate sections before making your selection.

☒ Life sciences          ☐ Behavioural & social sciences          ☐ Ecological, evolutionary & environmental sciences

For a reference copy of the document with all sections, see nature.com/documents/nr-reporting-summary-flat.pdf

# Life sciences study design

All studies must disclose on these points even when the disclosure is negative.

| | |
|---|---|
| Sample size | Mouse prostate organoids from 4 different mice (WT male C57BL/6 mice, 8-months-old). Experiments with human prostate tissue were done with samples from 3 donors. |
| Data exclusions | No data was excluded from the study |
| Replication | n for each experiment is stated in the figure legends. |
| Randomization | Human prostate tissue was chosen based on availability of tissue from patient at that time. Male mice for organoid generation were also randomly chosen from the pool of mice used at the moment for other experiments at the Institute. |
| Blinding | Blinding was not necessary in this study. |

# Reporting for specific materials, systems and methods

We require information from authors about some types of materials, experimental systems and methods used in many studies. Here, indicate whether each material, system or method listed is relevant to your study. If you are not sure if a list item applies to your research, read the appropriate section before selecting a response.

## Materials & experimental systems

| n/a | Involved in the study |
|---|---|
| ☐ | ☒ Antibodies |
| ☐ | ☒ Eukaryotic cell lines |
| ☒ | ☐ Palaeontology and archaeology |
| ☐ | ☒ Animals and other organisms |
| ☒ | ☐ Clinical data |
| ☒ | ☐ Dual use research of concern |
| ☒ | ☐ Plants |

## Methods

| n/a | Involved in the study |
|---|---|
| ☒ | ☐ ChIP-seq |
| ☐ | ☒ Flow cytometry |
| ☒ | ☐ MRI-based neuroimaging |

# Antibodies

| Antibodies used | All antibodies used in the study are commercially available and have been validated by the supplier.<br><br>Immunofluorescence:<br>Anti-Cytokeratin 5 (1:100, ab52635, Abcam)<br>Anti-Cytokeratin 5 Alexa Fluor® 647 antibody (1:100, ab193895, Abcam)<br>Anti-CD24a (1:100, 10600-1-AP, Proteintech)<br>Anti-ACPP (PPAP, 1:100, LS-C292593, LSBio)<br>Anti-KRT18 (1:100 ab133263, Abcam)<br>Anti-ZO-1 Polyclonal antibody (1:750, 21773-1-AP, Proteintech)<br>Anti-FLAG antibody (1:1,000; 66008-4-Ig, Proteintech)<br>Anti-E. coli LPS (1:500, ab35654, Abcam)<br>Flash Phalloidin™ Green 488 (1:250, 424201, Biolegend)<br>Flash Phalloidin™ Red 594 (1:250, 424203, Biolegend)<br>Phalloidin Alexa 647 (1:250, A30107, Invitrogen)<br>Hoechst 33342 (1:5,000, H3570, Life Technologies)<br>Anti-Rabbit IgG Alexa Fluor™ 488 (1:500, A21441, Invitrogen)<br>Anti-Mouse IgG Alexa Fluor™ 594 (1:500A21201, Invitrogen)<br><br>Western Blot:<br>Anti-mouse IgG-HRP conjugate (1:10,000, GENA931, Sigma Aldrich)<br>Anti-rabbit IgG-HRP conjugate (1:10,000, GENA934, Sigma Aldrich)<br>Anti-rabbit IgG-HRP conjugate (1:10,000, SA00001-2, Proteintech)<br>Anti-FLAG antibody (1:1,000; 66008-4-Ig, Proteintech)<br>Anti-HA Polyclonal Antibody (1:5,000, 51064-2-AP, Proteintech)<br>Anti-ACPP Antibody (1:300, HPA063916,Atlas Antibodies)<br>Anti-FimH Antibody (1:1,000, CSB-PA362349ZA01ENV, LSBio)<br>Anti-GroEL Antibody (1:5,000, G6532, Sigma)<br><br>Flow cytometry<br>Anti-mouse CD24 (FITC, 1:500, #101805, Biolegend)<br>Anti-mouse CD49f (Alexa647, 1:750, #313610, Biolegend)<br>Isotype controls Rat IgG2b, κ Isotype Ctrl (FITC, 1:500, #400633, Biolegend)<br>Isotype controls Rat IgG2a, κ Isotype Ctrl (Alexa 647, 1:750, #400526, Biolegend)<br><br>Bacterial Binding<br>Anti-HA (12.5 μg/mL, 51064-2-AP, Proteintech)<br>Anti-LPS (12.5 μg/mL, ab35654, Abcam)<br><br>Blocking assay<br>Anti-β1 integrin (1,5 μg/well, 6S6; sc-53711, Santa Cruz Biotechnology)<br>Anti-α3 integrin (1,5 μg/well, P1B5; sc-13545, Santa Cruz Biotechnology) |
|---|---|
| Validation | All antibodies used in this study were commercial antibodies available for Western-blot and immunofluorescence. Validation data are available on the manufacturer's websites and data sheets. |

# Eukaryotic cell lines

Policy information about cell lines and Sex and Gender in Research

| Cell line source(s) | ATCC HTB-9 (5637), ATCC CCL-2.1 (HeLa229) |
|---|---|
| Authentication | Performed by ATCC |
| Mycoplasma contamination | Mycoplasma contamination was tested routinely every month. |
| Commonly misidentified lines<br>(See ICLAC register) | None |

# Animals and other research organisms

Policy information about studies involving animals; ARRIVE guidelines recommended for reporting animal research, and Sex and Gender in Research

| | |
|---|---|
| Laboratory animals | C57BL/6 mice WT, 8 months old, male. |
| Wild animals | N/A |
| Reporting on sex | Only male mice were used to generate organoids (females do not have prostates). |
| Field-collected samples | N/A |
| Ethics oversight | To adhere to the 3R principles (Replacement, Reduction, Refinement), we utilised only leftover material from mice previously used for other approved experiments. |

Note that full information on the approval of the study protocol must also be provided in the manuscript.

# Plants

| | |
|---|---|
| Seed stocks | *Report on the source of all seed stocks or other plant material used. If applicable, state the seed stock centre and catalogue number. If plant specimens were collected from the field, describe the collection location, date and sampling procedures.* |
| Novel plant genotypes | *Describe the methods by which all novel plant genotypes were produced. This includes those generated by transgenic approaches, gene editing, chemical/radiation-based mutagenesis and hybridization. For transgenic lines, describe the transformation method, the number of independent lines analyzed and the generation upon which experiments were performed. For gene-edited lines, describe the editor used, the endogenous sequence targeted for editing, the targeting guide RNA sequence (if applicable) and how the editor was applied.* |
| Authentication | *Describe any authentication procedures for each seed stock used or novel genotype generated. Describe any experiments used to assess the effect of a mutation and, where applicable, how potential secondary effects (e.g. second site T-DNA insertions, mosiacism, off-target gene editing) were examined.* |

# Flow Cytometry

## Plots

Confirm that:

☒ The axis labels state the marker and fluorochrome used (e.g. CD4-FITC).

☒ The axis scales are clearly visible. Include numbers along axes only for bottom left plot of group (a 'group' is an analysis of identical markers).

☒ All plots are contour plots with outliers or pseudocolor plots.

☒ A numerical value for number of cells or percentage (with statistics) is provided.

## Methodology

| | |
|---|---|
| Sample preparation | For analysis of naïve cells, 7days-old 2D models were dissociated into single cells with TrypLE Express (12605028, Gibco) and washed with FACS buffer (PBS, 10% FBS, 50mM EDTA). Infected cells were treated washed 3 times with PBS and then stained following the same protocol. |
| Instrument | NovoCyte Quanteon flow cytometer. |
| Software | NovoExpress software (Agilent). |
| Cell population abundance | No sorting was performed. The total population after cell debris exclusion was separated into Cd24a+ or Cd49+ cells. Both populations were well represented in all the samples. |
| Gating strategy | Cells were first gated on SSC versus FSC to exclude debris, then gated based on APC-Cd49f (basal cells) or FITC-Cd24 (luminal cells) expression. mCherry-positive cells (infected cells) were subsequently measured. Gating strategy is shown in Supplementary Figure 5. |

☒ Tick this box to confirm that a figure exemplifying the gating strategy is provided in the Supplementary Information.

