## [Peer Review File · Nature Microbiology]

Uropathogenic *Escherichia coli* invade luminal prostate cells via FimH-PPAP receptor binding

Corresponding Author: Dr Carmen Aguilar

Version 0:

Reviewer comments:

Reviewer #1

(Remarks to the Author)

In this study, Guedes et al. developed a mouse 2D prostate organoid model for uropathogenic *E. coli* (UPEC) infection studies. The authors characterized the key ligand-receptor pair between the *E. coli* FimH and luminal prostate cell membrane protein PPAP, as the direct mediator of the infection. Given that FimH small-molecule inhibitors and vaccines were already applied to a bladder infection, this result suggested the possibility that the same treatment could benefit patients with a prostate infection. The result is quite interesting and can potentially benefit patients with bladder infection, I have the following critiques/suggestions and one major concern for the manuscript. If the authors could address my major concern (bulletin #2 below), I would recommend the acceptance for publication in Nature Microbiology.

1. The authors did state that UPEC is a common cause of bladder infection (~80%), but no data or references is provided to show how common human prostate tissue infection was caused by UPEC and the specific UPEC strains. The clinical significance of the paper would be further strengthened by those data.
2. all the experiments were performed with cystitis isolate UPEC UTI89 from the bladder, while the exact prostate isolate CP1 was not given. Considering the whole prostate infection study is based on one single bladder-derived strain (rather than a few prostate-derived strains), it remains questionable whether the proposed mechanism can be generalized and applicable to the real case of human patients.
3. The basal/luminal cell proportion in Figure 1b and 1c does not match the control sample. In the control, IF staining has almost no Cd24+ luminal signals, but scRNA-seq unveiled a noticeable portion of luminal cells. For the DHT 1nM sample, the IF image in Figure 1b does have a clear border of Cd24a+ cells versus Krt5+ cells. With the border and the discrepancy in mind, I am wondering whether the Control panel is indeed representative. The authors shall provide the quantification statistics whenever the IF images are provided (e.g. Figure 1b, Figure 2d, Figure 6, etc.).
4. It is unclear to me what the "differentiated model" and "undifferentiated model" (Line 182-183) refer to. Do the authors refer to DHT+ and Ctrl models? Here the authors intended to claim UPEC's preference towards luminal cells (Line 435). However, the DHT+ treatment could introduce unknown factors and, therefore, may not be the most appropriate control. Can the authors stain the luminal and basal cells and quantify the percentage of infected luminal and basal cells separately? To verify the authors' conclusion, direct evidence supporting that infected cells are mainly luminal is needed.
5. Following the Comment above. As the authors mentioned in the Discussion "the secreted isoform of PPAP might also act as a soluble FimH receptor". Do these potential soluble receptors transport between cells so that the luminal-cell-produced PPAP might make other cells capable of UPEC infection? It would be great if the authors could deepen their discussions.

Reviewer #2

(Remarks to the Author)

The authors identify a FimH binding protein in luminal prostate cells, prostatic acid phosphatase, that is linked to *E. coli* binding and internalization. The observation is novel and expands knowledge of prostate infections. The experiments are well designed and ample data are provided to support the observation. There are minor concerns, mainly with the overstated similarities between 3-D and 2-D cultures. They should modify statements in results and indicate the difference in 3-D and 2-D culture in their introduction and discussion. They provide data to support the use of 2-D culture for this investigation but overstate its similarity to 3-D culture. Additional gene expression of characteristic luminal cells such as Nkx3.1, probasin, PSCA, CD133, Trop1, etc should be included in the extended data analysis to further define similarities.

Reviewer #3

(Remarks to the Author)

This manuscript describes the development of a murine prostate organoid model and demonstrates its application to study UPEC infection. The authors show that the UPEC strain UT189 attaches to, invades, and replicates within luminal prostate cells. They then go on to identify the prostate-specific membrane protein PPAP as a receptor for the FimH adhesin of UPEC type 1 fimbriae, thus revealing a mechanism for UPEC adherence to the prostate epithelium.

Overall, the manuscript is well written and the work should be of interest to the field.

Major comments.

The authors start by characterising the ability of UPEC to infect prostate cells using their 2D organoid-based model. All of the data presented is performed using the well-characterised UT189 isolate, which was cultured from the urine of a woman with cystitis. I think an opportunity is missed here. It would be relevant to expand the analysis presented in Fig 2b to show how UPEC prostatitis isolates colonise the 2D organoid-based model over a 24h time course. Given the organoid model is established these experiments should not be difficult.

L192-193. Please define the criteria used to categorize tissues as 'healthy'.

Fig 4d (L262-263). I have a concern with this figure. The blot has been cut and is presented as two separate lanes. It does not look like it is the same blot. An explanation and new figure are required.

L269-287. I did not follow how the binding between rFimHLD and rPPAP was quantified in Fig 4g. What is meant by 'fold change over HA signal' as the value for the y-axis to Fig 4g.

Fig 5a-b. Invasion is measured at 1 hpi. This experiment could be performed as a time course, showing cfu's over 24h post infection.

Fig 6a. To demonstrate the images are representative of the overall data, the imaging analysis should be supported with quantitative data to show the number of adhered UPEC for the WT, fimH and WT+D-man assays across multiple fields.

Decision Letter:

11th December 2024

Dear Carmen,

Thank you for your patience while your manuscript "Prostate-specific membrane receptor PPAP facilitates E. coli invasion of luminal prostate cells via FimH binding" was under peer-review at Nature Microbiology. We got the final feedback very quickly so I am already able to pass on our decision about your manuscript. It has now been seen by 3 referees, whose expertise and comments you will find at the end of this email. They find your work of some potential interest, however they have also raised a number of concerns that will need to be addressed before we can consider publication of the work in Nature Microbiology.

In particular, several referees questioned whether these observations were more broadly relevant across UPEC strains and suggested that the work would be strengthened by analysis of prostate UPEC isolate(s). There were also several requests made for consolidation of the data with additional quantification and statistical analyses to validate conclusions drawn about the populations of luminal and basal cells and the extent to which these are infected by UPEC in the different models as well as to validate the 2D model itself further. Referee #2 particularly had concerns that there are differences between 2D and 3D models and similarity between the systems should not be overstated. Furthermore, Referee #3 had some concerns about the immunoblot data shown in Figure 4d, as well as requests for more quantification and time course analysis of the PPAP-FimH data. We feel that these are critical points which would need to be addressed for us to further consider a revised manuscript, alongside the remaining issues outlined in the referees' reports, which are clear and should be straightforward to address.

In further discussion with Referee #1, they also noted that the transcriptomic data provided to validate models could be improved with a quick demonstration of the basic statistics by providing steep curve analysis of UMI count, the gene detection and UMI statistics plot (two violin plots) in an extended data figure.

Should further experimental data allow you to address these criticisms, we would be happy to look at a revised manuscript.

We strongly support public availability of data. Please place the data used in your paper into a public data repository, if one exists, or alternatively, present the data as Source Data or Supplementary Information. If data can only be shared on request, please explain why in your Data Availability Statement, and also in the correspondence with your editor. For some data types, deposition in a public repository is mandatory - more information on our data deposition policies and available repositories can

be found at <https://www.nature.com/nature-research/editorial-policies/reporting-standards#availability-of-data>.

Please include a data availability statement as a separate section after Methods but before references, under the heading "Data Availability". This section should inform readers about the availability of the data used to support the conclusions of your study. This information includes accession codes to public repositories (data banks for protein, DNA or RNA sequences, microarray, proteomics data etc...), references to source data published alongside the paper, unique identifiers such as URLs to data repository entries, or data set DOIs, and any other statement about data availability. At a minimum, you should include the following statement: "The data that support the findings of this study are available from the corresponding author upon request", mentioning any restrictions on availability. If DOIs are provided, we also strongly encourage including these in the Reference list (authors, title, publisher (repository name), identifier, year). For more guidance on how to write this section please see: <http://www.nature.com/authors/policies/data/data-availability-statements-data-citations.pdf>

* If you have not done so already we suggest that you begin to revise your manuscript so that it conforms to our Article format instructions at <http://www.nature.com/nmicrobiol/info/final-submission>. Refer also to any guidelines provided in this letter.

When submitting the revised version of your manuscript, please pay close attention to our [href="https://www.nature.com/nature-portfolio/editorial-policies/image-integrity">Digital Image Integrity Guidelines.](https://www.nature.com/nature-portfolio/editorial-policies/image-integrity) and to the following points below:

Link Redacted

Note: This url links to your confidential homepage and associated information about manuscripts you may have submitted or be reviewing for us. If you wish to forward this e-mail to co-authors, please delete this link to your homepage first.

Nature Microbiology is committed to improving transparency in authorship. As part of our efforts in this direction, we are now requesting that all authors identified as 'corresponding author' on published papers create and link their Open Researcher and Contributor Identifier (ORCID) with their account on the Manuscript Tracking System (MTS), prior to acceptance. This applies to primary research papers only. ORCID helps the scientific community achieve unambiguous attribution of all scholarly contributions. You can create and link your ORCID from the home page of the MTS by clicking on 'Modify my Springer Nature account'. For more information please visit www.springernature.com/orcid.

If you wish to submit a suitably revised manuscript we would hope to receive it within 6 months. If you cannot send it within this time, please let us know. We will be happy to consider your revision, even if a similar study has been accepted for publication at Nature Microbiology or published elsewhere (up to a maximum of 6 months).

Yours sincerely,

Reviewer Expertise:

- Referee #1: transcriptomic validation
- Referee #2: prostate biology and disease
- Referee #3: UPEC-host interactions, infection biology

Reviewer Comments:

Reviewer #1 (Remarks to the Author):

In this study, Guedes et al. developed a mouse 2D prostate organoid model for uropathogenic *E. coli* (UPEC) infection studies. The authors characterized the key ligand-receptor pair between the *E. coli* FimH and luminal prostate cell membrane protein PAPP, as the direct mediator of the infection. Given that FimH small-molecule inhibitors and vaccines were already applied to a bladder infection, this result suggested the possibility that the same treatment could benefit patients with a prostate infection. The result is quite interesting and can potentially benefit patients with bladder infection, I have the following critiques/suggestions and one major concern for the manuscript. If the authors could address my major concern (bulletin #2 below), I would recommend the acceptance for publication in *Nature Microbiology*.

1. The authors did state that UPEC is a common cause of bladder infection (~80%), but no data or references is provided to show how common human prostate tissue infection was caused by UPEC and the specific UPEC strains. The clinical significance of the paper would be further strengthened by those data.
2. All the experiments were performed with cystitis isolate UPEC UT189 from the bladder, while the exact prostate isolate CP1 was not given. Considering the whole prostate infection study is based on one single bladder-derived strain (rather than a few prostate-derived strains), it remains questionable whether the proposed mechanism can be generalized and applicable to the real case of human patients.
3. The basal/luminal cell proportion in Figure 1b and 1c does not match the control sample. In the control, IF staining has almost no Cd24+ luminal signals, but scRNA-seq unveiled a noticeable portion of luminal cells. For the DHT 1nM sample, the IF image in Figure 1b does have a clear border of Cd24a+ cells versus Krt5+ cells. With the border and the discrepancy in mind, I am wondering whether the Control panel is indeed representative. The authors shall provide the quantification statistics whenever the IF images are provided (e.g. Figure 1b, Figure 2d, Figure 6, etc.).
4. It is unclear to me what the "differentiated model" and "undifferentiated model" (Line 182-183) refer to. Do the authors refer to DHT+ and Ctrl models? Here the authors intended to claim UPEC's preference towards luminal cells (Line 435). However, the DHT+ treatment could introduce unknown factors and, therefore, may not be the most appropriate control. Can the authors stain the luminal and basal cells and quantify the percentage of infected luminal and basal cells separately? To verify the authors' conclusion, direct evidence supporting that infected cells are mainly luminal is needed.
5. Following the Comment above. As the authors mentioned in the Discussion "the secreted isoform of PPAP might also act as a soluble FimH receptor". Do these potential soluble receptors transport between cells so that the luminal-cell-produced PPAP might make other cells capable of UPEC infection? It would be great if the authors could deepen their discussions.
6. A minor improvement for a quick demonstration of the basic statistics could be providing steep curve analysis of UMI count, the gene detection and UMI statistics plot (two violin plots) in an extended data figure. But the authors did provide the description of the detailed analysis and the data in the excel files. And you could add the highlighted suggestion to my review if you have not sent it out to them.

Reviewer #2 (Remarks to the Author):

The authors identify a FimH binding protein in luminal prostate cells, prostatic acid phosphatase, that is linked to *E. coli* binding and internalization. The observation is novel and expands knowledge of prostate infections. The experiments are well designed and ample data are provided to support the observation. There are minor concerns, mainly with the overstated similarities between 3-D and 2-D cultures. They should modify statements in results and indicate the difference in 3-D and 2-D culture in their introduction and discussion. They provide data to support the use of 2-D culture for this investigation but overstate its similarity to 3-D culture. Additional gene expression of characteristic luminal cells such as Nkx3.1, probasin, PSCA, CD133, Trop1, etc should be included in the extended data analysis to further define similarities.

Reviewer #3 (Remarks to the Author):

This manuscript describes the development of a murine prostate organoid model and demonstrates its application to study UPEC infection. The authors show that the UPEC strain UT189 attaches to, invades, and replicates within luminal prostate cells. They then go on to identify the prostate-specific membrane protein PAPP as a receptor for the FimH adhesin of UPEC type 1 fimbriae, thus revealing a mechanism for UPEC adherence to the prostate epithelium.

Overall, the manuscript is well written and the work should be of interest to the field.

Major comments.

The authors start by characterising the ability of UPEC to infect prostate cells using their 2D organoid-based model. All of the data presented is performed using the well-characterised UT189 isolate, which was cultured from the urine of a woman with cystitis. I think an opportunity is missed here. It would be relevant to expand the analysis presented in Fig 2b to show how UPEC prostatitis isolates colonise the 2D organoid-based model over a 24h time course. Given the organoid model is established these experiments should not be difficult.

L192-193. Please define the criteria used to categorize tissues as 'healthy'.

Fig 4d (L262-263). I have a concern with this figure. The blot has been cut and is presented as two separate lanes. It does not look like it is the same blot. An explanation and new figure are required.

L269-287. I did not follow how the binding between rFimHLD and rPPAP was quantified in Fig 4g. What is meant by 'fold change over HA signal' as the value for the y-axis to Fig 4g.

Fig 5a-b. Invasion is measured at 1 hpi. This experiment could be performed as a time course, showing cfu's over 24h post infection.

Fig 6a. To demonstrate the images are representative of the overall data, the imaging analysis should be supported with quantitative data to show the number of adhered UPEC for the WT, fimH and WT+D-man assays across multiple fields.

Version 1:

Reviewer comments:

Reviewer #1

(Remarks to the Author)

The revised manuscript has provided a substantial amount of new data to address my concerns and the important statistical data. I thank the authors for putting in these efforts to improve their manuscript. I will suggest acceptance for publication.

Reviewer #2

(Remarks to the Author)

The authors have adequately addressed reviewer concerns.

Reviewer #3

(Remarks to the Author)

This revised manuscript addresses the comments raised in my first review. The authors also appear to have addressed the key questions raised by the other reviewers, and they have added additional revisions that improve the clarity of the manuscript. Overall, I commend the authors on their attention to detail in the response and revised manuscript.

My only outstanding request is that the authors provide more detail to allow identification of E. coli strains P10 and P16. Are these strains E. coli 10 and E. coli 16 in Figure 1 of reference 9. If so, this should be stated. If not, at a minimum the Sequence Type of the isolates should be stated. It would be best if all strains used in the study are noted and referenced in the Methods section.

Decision Letter:

Our ref: NMICROBIOL-24103325A

8th August 2025

Dear Dr. Aguilar,

Thank you for submitting your revised manuscript "Prostate-specific membrane receptor PPAP facilitates E. coli invasion of luminal prostate cells via FimH binding" (NMICROBIOL-24103325A). It has now been seen by the original referees and their comments are below. The reviewers find that the paper has improved in revision, and therefore we'll be happy in principle to publish it in Nature Microbiology, pending minor revisions to satisfy the referees' final requests and to comply with our editorial and formatting guidelines.

Thank you again for your interest in Nature Microbiology Please do not hesitate to contact me if you have any questions.

Sincerely,

Reviewer #1 (Remarks to the Author):

The revised manuscript has provided a substantial amount of new data to address my concerns and the important statistical data. I thank the authors for putting in these efforts to improve their manuscript. I will suggest acceptance for publication.

Reviewer #2 (Remarks to the Author):

The authors have adequately addressed reviewer concerns.

Reviewer #3 (Remarks to the Author):

This revised manuscript addresses the comments raised in my first review. The authors also appear to have addressed the key questions raised by the other reviewers, and they have added additional revisions that improve the clarity of the manuscript. Overall, I commend the authors on their attention to detail in the response and revised manuscript.

My only outstanding request is that the authors provide more detail to allow identification of *E. coli* strains P10 and P16. Are these strains *E. coli* 10 and *E. coli* 16 in Figure 1 of reference 9. If so, this should be stated. If not, at a minimum the Sequence Type of the isolates should be stated. It would be best if all strains used in the study are noted and referenced in the Methods section.

Version 2:

Reviewer comments:

Reviewer #3

(Remarks to the Author)

I have reviewed the additional changes made to this manuscript. This was an unfortunate mistake that I am relieved has been picked up in review. The authors appear to have addressed this appropriately, however further attention is required regarding the presentation of the statistical methods. Although the Methods section describes the statistical analyses, Nature journals require that all error bars be clearly defined in the corresponding figure legends. This has not been done for the figures presented, and should be corrected. Moreover, upon reviewing other figures, it is evident that this issue affects all figures containing statistical analyses and should be corrected throughout.

I have an additional concern with Fig 4d, extending from comments made in my initial review. I reviewed this again and I still have concerns with the way this is presented, as the authors only show a single lane from two different blots. This is unsatisfactory, and the fact that a similar mode of presentation was used in another paper does not justify the approach.

In the authors response, they presented a ligand capture assay as 'biological replicate no. 2'. This contained a far western analysis of lysates from THP1 cells, prostate organoids, 5637 cells and rFimH LD, probed with rFimH LD in the absence of D-mannose (left panel) and in the presence of D-mannose (right panel), that were detected with anti-FLAG antibody. I would like to see this blot included in the manuscript instead of the current panel Fig 4d. It is highly appropriate to show the far western analysis performed using prostate organoids and 5637 cells, as both cell types are examined in the current study. The rFimH LD lane is an appropriate control. I am happy for the authors to decide whether or not the THP1 cell lane should be included.

Decision Letter:

16th October 2025

Dear Carmen,

Thank you for your patience while your manuscript "Prostate-specific membrane receptor PPAP facilitates *E. coli* invasion of luminal prostate cells via FimH binding" was under peer-review with one of the original reviewers. You will see from their comments below that they are happy with the changes made regarding the images of the P10 and P16 infections. However, they note two other points. These are to note what the error bars represent in the figure legends throughout the manuscript. They also note a continued concern with one of the western blots - however they make a suggestion to replace it with the blots shown in the previous rebuttal [rebuttal figure 3b; biological replicate 2]. We would ask you to make these minor changes, which don't require any additional experiments and so should be quick and easy to address. We are very interested in the possibility of

publishing your study in Nature Microbiology, so would ask you to make these changes. We do not see a need to return the manuscript to reviewers another time once we have received the revised version, but these changes are essential before we can make a final decision on publication.

If you have not done so already please begin to revise your manuscript so that it conforms to our Article format instructions at <http://www.nature.com/nmicrobiol/info/final-submission/>

The usual length limit for a Nature Microbiology Article is six display items (figures or tables) and 3,500 words, though there is margin for more complex articles up to 4,000 words. All Methods should be described in a separate section following the discussion, we do not place a word limit on Methods.

Nature Microbiology titles should give a sense of the main new findings of a manuscript, and should not contain punctuation. Please keep in mind that we strongly discourage active verbs in titles, and that they should ideally fit within 90 characters each (including spaces).

Please include a data availability statement as a separate section after Methods but before references, under the heading "Data Availability". This section should inform readers about the availability of the data used to support the conclusions of your study. This information includes accession codes to public repositories (data banks for protein, DNA or RNA sequences, microarray, proteomics data etc...), references to source data published alongside the paper, unique identifiers such as URLs to data repository entries, or data set DOIs, and any other statement about data availability. At a minimum, you should include the following statement: "The data that support the findings of this study are available from the corresponding author upon request", mentioning any restrictions on availability. If DOIs are provided, we also strongly encourage including these in the Reference list (authors, title, publisher (repository name), identifier, year). For more guidance on how to write this section please see: <http://www.nature.com/authors/policies/data/data-availability-statements-data-citations.pdf>

To improve the accessibility of your paper to readers from other research areas, please pay particular attention to the wording of the paper's opening bold paragraph, which serves both as an introduction and as a brief, non-technical summary in about 150 words. If, however, you require one or two extra sentences to explain your work clearly, please include them even if the paragraph is over-length as a result. The opening paragraph should not contain references. Because scientists from other sub-disciplines will be interested in your results and their implications, it is important to explain essential but specialised terms concisely. We suggest you show your summary paragraph to colleagues in other fields to uncover any problematic concepts.

If your paper is accepted for publication, we will edit your display items electronically so they conform to our house style and will reproduce clearly in print. If necessary, we will re-size figures to fit single or double column width. If your figures contain several parts, the parts should form a neat rectangle when assembled. Choosing the right electronic format at this stage will speed up the processing of your paper and give the best possible results in print. We would like the figures to be supplied as vector files - EPS, PDF, AI or postscript (PS) file formats (not raster or bitmap files), preferably generated with vector-graphics software (Adobe Illustrator for example). Please try to ensure that all figures are non-flattened and fully editable. All images should be at least 300 dpi resolution (when figures are scaled to approximately the size that they are to be printed at) and in RGB colour format. Please do not submit Jpeg or flattened TIFF files. Please see also 'Guidelines for Electronic Submission of Figures' at the end of this letter for further detail.

Figure legends must provide a brief description of the figure and the symbols used, within 350 words, including definitions of any error bars employed in the figures.

When submitting the revised version of your manuscript, please pay close attention to our [href="https://www.nature.com/nature-research/editorial-policies/image-integrity">Digital Image Integrity Guidelines](https://www.nature.com/nature-research/editorial-policies/image-integrity) and to the following points below:

EXTENDED DATA FIGURES

Finally, please ensure that you retain unprocessed data and metadata files after publication, ideally archiving data in perpetuity,

as these may be requested during the peer review and production process or after publication if any issues arise.

Please include a statement before the acknowledgements naming the author to whom correspondence and requests for materials should be addressed.

Finally, we require authors to include a statement of their individual contributions to the paper -- such as experimental work, project planning, data analysis, etc. -- immediately after the acknowledgements. The statement should be short, and refer to authors by their initials. For details please see the Authorship section of our joint Editorial policies at http://www.nature.com/authors/editorial_policies/authorship.html

* include a point-by-point response to any editorial suggestions and to our referees. Please include your response to the editorial suggestions in your cover letter, and please upload your response to the referees as a separate document.

* ensure it complies with our format requirements for Letters as set out in our guide to authors at www.nature.com/nmicrobiol/info/gta/

* state in a cover note the length of the text, methods and legends; the number of references; number and estimated final size of figures and tables

* resubmit electronically if possible using the link below to access your home page:

Link Redacted

*This url links to your confidential homepage and associated information about manuscripts you may have submitted or be reviewing for us. If you wish to forward this e-mail to co-authors, please delete this link to your homepage first.

Please ensure that all correspondence is marked with your Nature Microbiology reference number in the subject line.

Nature Microbiology is committed to improving transparency in authorship. As part of our efforts in this direction, we are now requesting that all authors identified as 'corresponding author' on published papers create and link their Open Researcher and Contributor Identifier (ORCID) with their account on the Manuscript Tracking System (MTS), prior to acceptance. This applies to primary research papers only. ORCID helps the scientific community achieve unambiguous attribution of all scholarly contributions. You can create and link your ORCID from the home page of the MTS by clicking on 'Modify my Springer Nature account'. For more information please visit www.springernature.com/orcid.

We hope to receive your revised paper within three weeks. If you cannot send it within this time, please let us know.

Yours sincerely,

Reviewer Expertise:

Referee #3: E coli infection

Reviewers Comments:

Reviewer #3 (Remarks to the Author):

I have reviewed the additional changes made to this manuscript. This was an unfortunate mistake that I am relieved has been picked up in review. The authors appear to have addressed this appropriately, however further attention is required regarding the presentation of the statistical methods. Although the Methods section describes the statistical analyses, Nature journals require that all error bars be clearly defined in the corresponding figure legends. This has not been done for the figures presented, and should be corrected. Moreover, upon reviewing other figures, it is evident that this issue affects all figures containing statistical analyses and should be corrected throughout.

I have an additional concern with Fig 4d, extending from comments made in my initial review. I reviewed this again and I still have concerns with the way this is presented, as the authors only show a single lane from two different blots. This is unsatisfactory, and the fact that a similar mode of presentation was used in another paper does not justify the approach.

In the authors response, they presented a ligand capture assay as 'biological replicate no. 2'. This contained a far western analysis of lysates from THP1 cells, prostate organoids, 5637 cells and rFimH LD, probed with rFimH LD in the absence of D-mannose (left panel) and in the presence of D-mannose (right panel), that were detected with anti-FLAG antibody. I would like to

see this blot included in the manuscript instead of the current panel Fig 4d. It is highly appropriate to show the far western analysis performed using prostate organoids and 5637 cells, as both cell types are examined in the current study. The rFimH LD lane is an appropriate control. I am happy for the authors to decide whether or not the THP1 cell lane should be included.

Version 3:

Decision Letter:

Our ref: NMICROBIOL-24103325C

5th November 2025

Dear Dr Aguilar,

Thank you for submitting your revised manuscript "Prostate-specific membrane receptor PPAP facilitates E. coli invasion of luminal prostate cells via FimH binding" (NMICROBIOL-24103325C). We'll be happy in principle to publish it in Nature Microbiology, pending minor revisions to comply with our editorial and formatting guidelines.

We are now finalising a checklist detailing our editorial and formatting requirements which we will send to you either tomorrow or in the next few days. Please do not upload the final materials and make any revisions until you receive this additional information from us.

Thank you again for your interest in Nature Microbiology Please do not hesitate to contact me if you have any questions.

Sincerely,

Version 4:

Decision Letter:

24th November 2025

Dear Dr Aguilar,

I am pleased to accept your Article "Uropathogenic Escherichia coli invade luminal prostate cells via FimH-PPAP receptor binding" for publication in Nature Microbiology. Thank you for having chosen to submit your work to us and many congratulations.

Over the next few weeks, your paper will be copyedited to ensure that it conforms to Nature Microbiology style.

You may wish to make your media relations office aware of your accepted publication, in case they consider it appropriate to organize some internal or external publicity. Once your paper has been scheduled you will receive an email confirming the publication details. This is normally 3-4 working days in advance of publication. If you need additional notice of the date and time of publication, please let the production team know when you receive the proof of your article to ensure there is sufficient time to coordinate. Further information on our embargo policies can be found here:

<https://www.nature.com/authors/policies/embargo.html>

Authors may need to take specific actions to achieve compliance with funder and institutional open access mandates. If your research is supported by a funder that requires immediate open access (e.g. according to [Plan S principles](https://www.springernature.com/gp/open-science/plan-s-compliance) or the [NIH public access policy](https://www.springernature.com/gp/open-science/us-federal-agency-compliance)) then you should select the gold OA route, and we will direct you to the compliant route where possible. Because authors warrant under our subscription licensing terms that they haven't committed to licensing any version of their article under a licence inconsistent with the terms of our agreement – including the applicable embargo period – publication under the subscription model isn't suitable for authors whose funders require no embargo.

With kind regards,

P.S. Click on the following link if you would like to recommend Nature Microbiology to your librarian <http://www.nature.com/subscriptions/recommend.html#forms>

** Visit the Springer Nature Editorial and Publishing website at http://editorial-jobs.springernature.com?utm_source=ejP_NMicro_email&utm_medium=ejP_NMicro_email&utm_campaign=ejp_NMicro for more information about our career opportunities. If you have any questions please click [here](mailto:editorial.publishing.jobs@springernature.com).

Reviewer Comments:

We are very grateful to the reviewers for their careful evaluation of our work and constructive comments. Please find below our point-by-point responses to the comments (in blue).

Reviewer #1:

In this study, Guedes et al. developed a mouse 2D prostate organoid model for uropathogenic *E. coli* (UPEC) infection studies. The authors characterized the key ligand-receptor pair between the *E. coli* FimH and luminal prostate cell membrane protein PAPP, as the direct mediator of the infection. Given that FimH small-molecule inhibitors and vaccines were already applied to a bladder infection, this result suggested the possibility that the same treatment could benefit patients with a prostate infection. The result is quite interesting and can potentially benefit patients with bladder infection, I have the following critiques/suggestions and one major concern for the manuscript. If the authors could address my major concern (bulletin #2 below), I would recommend the acceptance for publication in Nature Microbiology.

1. The authors did state that UPEC is a common cause of bladder infection (~80%), but no data or references is provided to show how common human prostate tissue infection was caused by UPEC and the specific UPEC strains. The clinical significance of the paper would be further strengthened by those data.

We thank the Reviewer for this insightful comment. Indeed, the clinical significance of our results would be strengthened by discussing this point.

Epidemiological studies support that *E. coli* is the predominant pathogen in acute bacterial prostatitis, with most of them belonging to the UPEC classification (Lipsky, Byren, and Hoey 2010; Krieger and Thumbikat 2016). This is supported by several lines of evidence:

- *E. coli* strains are categorised into one of four phylogenetic groups A, B1, B2 and D, with UPEC strains belonging mainly to phylogenetic group B2 and D (Clermont, Bonacorsi, and Bingen 2000), same as *E. coli* causing prostatitis (Krieger et al. 2011; Krieger and Thumbikat 2016).
- UPEC strains are distinguishable from non-UPEC strains by the presence of specific virulence factors, all of which can be found in prostatitis isolates too. Comparisons between prostatitis isolates and other UTI isolates (e.g., from cystitis or pyelonephritis) have shown variable results: while some studies report a higher prevalence of virulence factors in prostatitis isolates (e.g., *hly* or *cnf1* (Ruiz et al. 2002); *sfa*, *hly*, and *cnf1* (Terai et al. 1997), others show the opposite, with a higher presence of strains carrying *cnf1*, *sat*, *uspA*, and *hly* in cystitis isolates (Morales- Espinosa et al. 2016).
- Several studies have shown that *E. coli* isolates from prostatitis cases predominantly belong to UPEC and carry UPEC's key virulence genes (*hly*, *cdt1*, *clb*, *pap*, *sfa/foc*, *fyuA*, *iroN*, *kpsMT(II)*, and *traT*). One study concluded that the phylogenetic background and the accumulation of an exceptional repertoire of virulence genes indicate that prostatitis *E. coli* strains belong to a highly virulent subset of UPEC (Krieger et al. 2011). Of note, in those studies, type 1 fimbriae (*fimH*) was present in all prostatitis isolates (Krieger et al. 2011) or showed no difference in prevalence when compared across isolates from cystitis, pyelonephritis, or prostatitis (Ruiz et al. 2002).

In conclusion, although definitive data directly comparing prostatitis isolates to those causing cystitis or other UTIs are limited, the available literature indicates that most *E. coli* strains isolated from prostatitis cases fall within the UPEC pathotype.

As suggested by the Reviewer, we have added this information to the manuscript. The following text was added to the introduction section (line 54): “*Escherichia coli* (*E. coli*) strains, predominantly uropathogenic *E. coli* (UPEC), are the most common cause of bacterial prostatitis”, and the references Krieger and Thumbikat 2016, and Krieger et al 2011 were added.

2. all the experiments were performed with cystitis isolate UPEC UTI89 from the bladder, while the exact prostate isolate CP1 was not given. Considering the whole prostate infection study is based on one single bladder-derived strain (rather than a few prostate-derived strains), it remains questionable whether the proposed mechanism can be generalized and applicable to the real case of human patients.

Our initial thought was that, since bacterial prostatitis strains are mostly UPEC strains (see point above), using the UPEC reference strain UTI89, the most commonly used UPEC strain in the literature, would allow for better comparison of our results with those of others. However, we thank the Reviewer (and Reviewer #3, please see below) for suggesting this. We fully agree with them that validating our data using prostatitis isolates would improve the significance of our results. Hence, we have validated all major results using three prostatitis isolates:

- **CP1:** *E. coli* strain isolated from a chronic prostatitis patient (Rudick et al. 2011). Considered the reference strain for prostatitis and suggested by Reviewer #1.
- **P10:** *E. coli* strain isolated from an acute prostatitis case. Belongs to the phylogenetic group D. First described by Krieger et al. 2011.
- **P16:** *E. coli* strain isolated from an acute prostatitis case. Belongs to the phylogenetic group B2. First described by Krieger et al. 2011.

To evaluate whether the findings observed with UTI89 were consistent across the prostatitis isolates, we performed the following experiments:

- Flow cytometry analysis of infected cells with CP1 strain (**Extended Data Figure 9**, see Point #4 below).
- CFU time course and imaging of the infection with CP1, P10 and P16 (**Extended Data Figure 10a-f**).
- Incubation with human prostate tissue slides with CP1 (**Extended Data Figure 10g-h**).
- CFU analysis in the presence or absence of D-mannose with CP1, P10, and P16 (**Extended Data Figure 12a**).
- Incubation with human prostate tissue slides in the presence of D-mannose with CP1 (**Extended Data Figure 12b-c**).
- FimH protein sequence comparison among all four *E. coli* strains (**Extended Data Figure 16**).
- CFU quantification (1 and 24hpi) and imaging of the infection (1hpi) with the *Ppap* KO organoid lines with CP1, P10, and P16 (**Extended Data Figure 17**).
- Incubation with human prostate tissue slides with CP1 (**Extended Data Figure 18**).

This set of experiments demonstrated that the key observations made with the UPEC reference strain UTI89 are conserved in the prostatitis isolates used in our study as well. Specifically, we show that the prostatitis isolates:

- I. invade and replicate over time in the organoid-based model grown with DHT, and do so to a greater extent than in the model grown without DHT (**Extended Data Figure 10a-f**);

- II. preferentially bind to KRT18⁺ cells in human prostate tissue slides (**Extended Data Figure 10g-h**);
- III. are blocked from invading mouse prostate cells or attaching to human prostate tissue in the presence of D-mannose (**Extended Data Figure 12**);
- IV. show reduced invasion at 1 hpi and lower bacterial loads at 24 hpi in *Ppap*-knockout organoid lines (**Extended Data Figure 17**);
- V. preferentially bind to PPAP⁺ cells in human prostate tissue, a phenotype that is lost in the presence of D-mannose (**Extended Data Figure 18**).

The only set of experiments we did not repeat with the prostatitis isolates was the one involving the use of the recombinant FimH protein. Although we generated the recombinant protein from the strain UTI89, we compared the FimH sequences of CP1, P10, and P16 (as well as other UTI reference strains: NU14, 536, and CFT073) and found no mutations in the binding pocket (**Extended Data Figure 14**, highlighted in blue). Only common SNPs previously described as polymorphisms in UPEC strains were identified (Hung et al. 2002). Notably, all strains expressed FimH at levels comparable to UTI89 under our culture conditions (**Extended Data Figure 12d**).

In summary, these results suggest that the urogenital pathogenic *E. coli* strains used in our study, including UPEC (UTI89), chronic prostatitis (CP1), and acute prostatitis isolates (P10 and P16), preferentially invade luminal prostate cells in a FimH- and PPAP-dependent manner, and that this preferential binding to luminal cells is also observed in human prostate tissue. All of this data has been incorporated into the manuscript, both in the figures and the main text.

3. The basal/luminal cell proportion in Figure 1b and 1c does not match the control sample. In the control, IF staining has almost no Cd24⁺ luminal signals, but scRNA-seq unveiled a noticeable portion of luminal cells. For the DHT 1nM sample, the IF image in Figure 1b does have a clear border of Cd24a⁺ cells versus Krt5⁺ cells. With the border and the discrepancy in mind, I am wondering whether the Control panel is indeed representative. The authors shall provide the quantification statistics whenever the IF images are provided (e.g. Figure 1b, Figure 2d, Figure 6, etc.).

We thank the Reviewer for bringing this oversight to our attention. Upon re-examining our data, we recognised that the initially selected images were not representative and have now been replaced with more accurate examples (**Figure 1b**). Moreover, we have added the quantification of CD24a⁺ and KRT5⁺ cells from the confocal images (**Extended Data Figure 1a**), which showed that, while the scRNA-seq data indicated approximately 50% luminal cells in the control condition, immunofluorescence (IF) images showed only $27.2 \pm 12.6\%$ of luminal cells.

To further investigate this discrepancy, we performed flow cytometry using a surface marker for luminal cells (CD24a, as in the IF analysis, but using a flow cytometry-specific antibody, see **Methods** section) and basal cells (CD49f). The results followed the same trend as the IF data, with a higher proportion of basal cells in the control condition and a higher proportion of luminal cells in the DHT-treated condition (**Extended Data Figure 1b**).

For the Reviewer's convenience, we have summarised the data from all three approaches (scRNA-seq, IF, and flow cytometry) in a comparative table (see below). As expected, the absolute values vary somewhat between methods, as each measures cell-type proportions differently: scRNA-seq assesses transcript-level expression of marker genes (a set of signature marker genes), while IF and flow cytometry detect protein-level expression, either total (in permeabilised cells for IF) or surface (for flow cytometry).

		Ctrl		DHT10	
		Basal cells	Luminal cells	Basal cells	Luminal cells
Exp. Method	scRNA-seq	49	50	16	84
	IF	51.6 ± 13.5	27.2 ± 12.6	15.5 ± 5.3	68.8 ± 8.9
	FC	80.1 ± 11	18.1 ± 10.8	25.4 ± 1.3	73.4 ± 0.9

Table 1. Quantification of luminal and basal cell populations by scRNA-seq, immunofluorescence (IF) and flow cytometry (FC). Data is shown as % of the total population.

In addition and as requested by the Reviewer, we have quantified all immunofluorescence images presented in the manuscript and included the corresponding quantification graphs with statistical significance (**Figure 2f, 6b, Extended Data Figure 1a, 10h, 12c, 18b-c**), along with a summary of the statistical analyses performed in **Supplementary Table 3**.

4. It is unclear to me what the “differentiated model” and “undifferentiated model” (Line 182-183) refer to. Do the authors refer to DHT+ and Ctrl models? Here the authors intended to claim UPEC’s preference towards luminal cells (Line 435). However, the DHT+ treatment could introduce unknown factors and, therefore, may not be the most appropriate control. Can the authors stain the luminal and basal cells and quantify the percentage of infected luminal and basal cells separately? To verify the authors’ conclusion, direct evidence supporting that infected cells are mainly luminal is needed.

We apologise for the confusion. The text has been corrected and now reads like this (line 204): “Interestingly, we observed that UPEC replicates more efficiently in the model grown in the presence of DHT, enriched for luminal cells, than in the model grown without DHT, enriched for basal prostate cells (**Figure 2c**)”. This correction has also been implemented throughout the manuscript (i.e. line 167: “In addition, an increased transepithelial electrical resistance (TEER) was measured in the 2D model grown with DHT (**Figure 1h**)”; line 188: “This differential staining showed that UPEC rapidly invaded the 2D model, with a greater number of bacteria invading cells grown in the presence of DHT compared to the control model grown without DHT (**Figure 2a**)”; line 228: “murine prostate model enriched with luminal cells and that bacterial binding is also more efficient in human luminal prostate cells”).

We fully agree with the Reviewer that DHT present in the culture medium could influence the infection. For this reason, we did not add DHT during the time when the bacteria was incubated with the cells. This information was already in the original manuscript in the discussion section (Original manuscript, line 377: “Notably, DHT had no effect on UPEC growth, and it was removed from the medium during the short incubation of bacteria with cells to eliminate any potential influence of the hormone on UPEC”. However, this comment made us realise this information was missing in the Methods section, and it has now been added (line 581: “No DHT was added to the medium during infection”). Moreover, we also showed that DHT (10nM) did not affect bacterial growth (**Extended Data Figure 11**).

Nevertheless, we agree with the Reviewer that providing direct evidence to support our proposed model in which UPEC preferentially invades luminal cells would strengthen the study. Generating such evidence has proven challenging due to limitations in available antibodies for immunostaining luminal cells in mouse tissue. In our hands, only the anti-CD24a antibody from Proteintech (Cat. #10600 1 AP) yielded reliable staining in mouse cells, and this was the antibody used for the immunofluorescence in **Figure 1b**. Unexpectedly, this antibody also stained the bacteria when used in infected samples (see images below). We contacted the manufacturer and confirmed the result using a different lot of the same antibody, but the issue persisted. Despite this limitation, we proceeded to use the antibody to quantify the identity of infected cells, distinguishing between luminal (CD24a-positive) and basal (KRT5-positive) cells, as shown in the immunofluorescence images and quantification graph below (**Rebuttal Figure 1**). This data has not been included in the manuscript.

Rebuttal Figure 1. Quantification of *E. coli*-infected cells per cell type. *a.* Representative images of 2D models infected with UPEC (UTI89, red) and stained for KRT5 (magenta), CD24a (green), and nuclei (blue). Scale bar 25 μ m. *b.* Image quantification graph showing the percentage of *E. coli*-infected cells in the CD24a⁺ (khaki) or CD49f⁺ (pink) population. *n* = 3.

To try to overcome this technical limitation, we also tested several alternative antibodies (Proteintech #84342-5-RR, Invitrogen #MA5-11828, BioLegend #101801), but none produced good results. Hence, we tried an alternative approach using flow cytometry to quantify infection rates in CD24a-positive (luminal) and CD49f-positive (basal) cell populations. Using an anti-CD24a antibody available for flow cytometry (used by Karthaus et al. 2020; Biolegend #101805), we were able to quantify CD24a populations in the models (naïve and infected). Results showed a significantly higher proportion of infected cells being CD24⁺ compared to CD49f⁺ cells in both models (Ctrl and DHT; **Extended Data Figure 1b** for naïve cells; **Figure 2b** for UTI89-infected cells, **Extended Data Figure 9** for CP1-infected cells). Gating strategy is shown in **Extended Data Figure 8**.

Notably, similar trends were observed in both immunofluorescence and flow cytometry analyses. For the Reviewer’s convenience, we have summarised the data in a table and accompanying graph to facilitate easier comparison between the two methods.

		UTI89				CP1			
		Ctrl		DHT		Ctrl		DHT	
		Cd49f/Krt5	Cd24a	Cd49f/Krt5	Cd24a	Cd49f/Krt5	Cd24a	Cd49f/Krt5	Cd24a
FC	Mean	6.86	68.08	12.17	41.12	5.26	73.09	13.61	66.29
	stdev	2.94	13.97	6.03	0.56	1.35	13.95	2.23	2.96
IF	Mean	10.57	51.22	19.67	68.20	20.43	54.60	15.80	56.97
	stdev	13.86	15.98	7.42	8.33	9.06	19.26	8.65	10.44

Table 2. Quantification of infected cells in the luminal (CD24a⁺ cells) and basal cell (KRT5⁺ or CD49f⁺) populations by immunofluorescence (IF) and flow cytometry (FC). Data is shown as % of infected cells in each population.

Rebuttal Figure 2. Both flow cytometry (FC) and immunofluorescence (IF) analyses show similar trends, with a higher percentage of infected cells in the CD24a⁺ cell population. The graph compares results obtained from FC and IF imaging.

In summary, both IF and FC data show that *E. coli* (either UTI89 or CP1 strain) preferentially invade CD24a⁺ cells. Data from the flow cytometry assay have been added to the manuscript in both the figures (**Extended Data Figure 1b, Figure 2b, Extended Data Figure 9, and Extended Data Figure 9**) and the text:

- Results section, line 193: “To test whether the increased bacterial invasion observed in the DHT model was due to the higher number of luminal cells, we measured the percentage of infected cells within the luminal (CD24⁺) and basal (CD49f⁺) populations in each model. Results showed a consistent preference of *E. coli* for luminal over basal cells, with 68±14% of luminal cells and 6.9±3% of basal cells infected in the Ctrl model, and 41±0.6% of luminal cells and 12.17±6% of basal cells infected in the DHT model (**Figure 2b, Extended Data Figure 8**). To ensure that this observation was not specific to the UTI89 strain, we validated the preference for luminal cells using the *E. coli* isolate CP1, originally isolated from a chronic prostatitis patient¹⁶ which showed the same preference for luminal cells (Cd24⁺; **Extended Data Figure 9**).”
- Method section, line 603 “Flow cytometry. Cells were seeded as described above. For analysis of naïve cells, 7-day-old 2D models were dissociated into single cells with TrypLE Express (12605028, Gibco) and washed with FACS buffer (PBS, 10% FBS, 50 mM EDTA). Single cell suspensions were then stained with anti-mouse CD24 (FITC, 1:500; #101805, Biolegend) and anti-mouse CD49f (Alexa647, 1:750; #313610, Biolegend) or isotype control (#400633, #400526, Biolegend) antibodies for 30 min at 4°C, measured with the NovoCyte Quanteon flow cytometer and analysed with NovoExpress software (Agilent; see **Extended Data Figure 8**). For infected cells, cells were infected as described above, washed 3 times with PBS after 1 hpi, and then stained following the same protocol.”

5. Following the Comment above. As the authors mentioned in the Discussion “the secreted isoform of PPAP might also act as a soluble FimH receptor”. Do these potential soluble receptors transport between cells so that the luminal-cell-produced PPAP might make other cells capable of UPEC infection? It would be great if the authors could deepen their discussions.

We fully agree with the Reviewer that this is a very interesting hypothesis.

It is plausible that secreted PPAP could influence neighbouring cells, increasing their susceptibility to *E. coli* infection. While we have not directly tested this hypothesis in our current study, the secreted isoform of PPAP could, in principle, diffuse within the glandular lumen or the surrounding microenvironment. The effect this could have on luminal cells is unknown. However, since secreted PPAP does not have a membrane anchor, it seems unlikely that the uptake of soluble PPAP (intracellular) would represent a benefit for *E. coli* invasion.

We rather think that, since secreted PPAP has no membrane anchor and is secreted by the cells, it could act as a soluble receptor for FimH. This could actually block the FimH binding pocket and prevent its binding to membrane-bound PPAP, which may be beneficial for the host. *E. coli* with FimH blocked by soluble PPAP would be eliminated during ejaculation, in a similar way as described for uromodulin in the bladder (Stsiapanava et al. 2022). On the other hand, it could also benefit the bacteria by promoting clump formation and/or biofilm development. This is indeed a fascinating topic for us and is currently under study in our lab.

As suggested by the Reviewer, we have extended our discussion on this and now it reads like this (line 472): “Nevertheless, the secreted isoform, which lacks a membrane anchor, could diffuse within the prostate lumen and potentially influence infection beyond its cell of origin. In particular, secreted PPAP might act as a decoy receptor by binding to FimH, thereby blocking bacterial adhesion to membrane-

bound PPAP and promoting bacterial clearance, similar to the role described for uromodulin in the bladder. Alternatively, it could facilitate bacterial aggregation or biofilm formation, enhancing bacterial survival. These scenarios underscore the dual potential role of secreted PPAP to either protect the host or support bacterial colonisation. Further work will be necessary to determine their relevance during infection.”

6. A minor improvement for a quick demonstration of the basic statistics could be providing steep curve analysis of UMI count, the gene detection and UMI statistics plot (two violin plots) in an extended data figure. But the authors did provide the description of the detailed analysis and the data in the excel files.

We thank the Reviewer for this suggestion. We have added the steep curve analysis of UMI count, the gene detection and UMI statistics plot for the scRNA-seq data in the new **Extended Data Figure 2**.

Reviewer #2:

The authors identify a FimH binding protein in luminal prostate cells, prostatic acid phosphatase, that is linked to E. coli binding and internalization. The observation is novel and expands knowledge of prostate infections. The experiments are well designed and ample data are provided to support the observation. There are minor concerns, mainly with the overstated similarities between 3-D and 2-D cultures. They should modify statements in results and indicate the difference in 3-D and 2D culture in their introduction and discussion. They provide data to support the use of 2-D culture for this investigation but overstate its similarity to 3-D culture. Additional gene expression of characteristic luminal cells such as *Nkx3.1*, *probasin*, *PSCA*, *CD133 (Prom1)*, *Trop1 (Epcam)*, etc should be included in the extended data analysis to further define similarities.

We appreciate the Reviewer’s comment and agree that the differences between 3D and 2D cultures should be explained more clearly. Indeed, we agree with the Reviewer that 3D organoids and 2D organoid-based models are very distinct. They show key differences, such as the proportion of luminal and basal cells. The 3D model resembles a more undifferentiated prostate epithelium, mainly composed of basal cells. In contrast, the 2D model is more similar to the in vivo prostate epithelium, which has a majority of luminal cells (**Figure 1, Extended Data Figure 3**). Another difference (understated in the original manuscript and corrected now) is the expression of luminal markers, which is higher in the 2D model compared to the 3D model. These differences have been further emphasized in the results section (see below). Moreover, as suggested by the Reviewer, gene expression plots for other luminal markers have been added to the **Extended Data Figure 3**. In particular, *Cd113 (Prom1)*, *Trop1 (Epcam)*, originally in Extended Data Figure 3 and 4, now moved to the new **Extended Data Figure 3**, and *PscA (Extended Data Figure 3)*. Unfortunately, no transcripts were found for *Nkx3.1* or *Pbsn*. This suggest that further adjustments will have to be made in the culture media to improve the expression of these markers. This information has been added to the Results section (see below).

Additionally, we have expanded the discussion on this observation (Discussion section, see below). We believe that the differences observed could potentially be due to the distinct physical and mechanical cues, as well as the presence of extracellular matrix (ECM) components, between the 3D and 2D models. More specifically, in 3D culture, stem cells are embedded in a relatively permissive ECM that allows them to grow in all three dimensions. While this has traditionally been considered a strength of 3D organoid cultures (supporting tissue-like architecture, e.g. intestinal crypts), in vivo stem and other epithelial cells, although experiencing some degree of 3D freedom, are often confined within limited spaces. This spatial restriction may explain why, in our model, simply seeding organoid-derived cells onto a 2D surface results in a higher degree of differentiation (e.g., proportion of luminal cells in 3D versus 2D), despite the use of the same culture medium.

Another possible factor is the presence of growth factors in the ECM matrix. In our case, we used Matrigel® (Corning), a complex mixture of more than 1,800 proteins (Hughes, Postovit, and Lajoie 2010), including laminin, collagen IV, and heparan sulfate proteoglycans, among others. It also contains growth factors such as TGF-beta and EGF. Since Matrigel is not used in 2D cultures, the absence of these growth factors may promote luminal differentiation in 2D, whereas their presence in 3D could inhibit it. This information has been added to the Discussion section (see below).

Results:

- Line 115: “we used 3D organoids as maintenance cultures to expand the primary prostate cells, which were then seeded onto a 2D surface to generate a 2D model with apical accessibility”
- Line 140: “However, we observed a much higher percentage of luminal cells in the 2D model compared to 3D organoids, independent of the absence or presence of DHT, indicating an overall higher degree of differentiation in the 2D versus the 3D model (50.5% versus 20.1% in Ctrl medium, and 83.9% versus 32.6% in DHT medium, respectively; **Figure 1f, Supplementary Table 1**) that was achieved just by seeding the 3D organoid cells in a 2D surface. Although media supplementation with 10nM DHT increased the number of luminal cells in both models, the effect was more evident in 2D (from 50.5 to 83.9%) than in 3D (20.1% to 32.6%; **Figure 1f, Supplementary Table 1**), and the expression of the luminal markers such as *Cd24a*, *Krt8*, *Epcam*, *Cd113/Prom1*, and *Krt18* was higher in the 2D model than in its 3D counterpart (**Figure 1e, Extended Data Figure 3c-f**). Of note, although luminal cells expressed characteristic luminal markers (i.e. *Cd24a*, *Krt18*, *Epcam*, and *Cd113/Prom1*), transcripts for other luminal markers such as *Pbsn* and *Nkx3.1* were not detected in either the 3D or 2D models.”
- Line 169: “Together, our data demonstrate that the 2D organoid-based model grown in the presence of 10nM DHT mimics the epithelial compartment of the murine prostate epithelium.”

Discussion:

- Line 389: “Our data show important differences between the 3D organoid and the 2D organoid-based model: while the 3D model contains a significantly higher proportion of undifferentiated basal stem cell-like cells, the 2D model contains a majority of differentiated luminal cells.”
- Line 395: “The fact that simply seeding the cells on a 2D surface leads to increased differentiation into prostate luminal cells, despite using the same culture medium, suggests that other factors may be at play. For instance, differences in physical/mechanical cues such as the low stiffness of the extracellular matrix (Matrigel®) in 3D culture compared to the high stiffness of tissue culture plastic, and the presence of extracellular matrix components and growth factors in 3D but not in 2D, may influence cell fate decisions. Further studies will be needed to clarify the contribution of these factors to epithelial differentiation.”

Reviewer #3:

This manuscript describes the development of a murine prostate organoid model and demonstrates its application to study UPEC infection. The authors show that the UPEC strain UTI89 attaches to, invades, and replicates within luminal prostate cells. They then go on to identify the prostate-specific membrane protein PAPP as a receptor for the FimH adhesin of UPEC type 1 fimbriae, thus revealing a mechanism for UPEC adherence to the prostate epithelium.

Overall, the manuscript is well written and the work should be of interest to the field.

Major comments:

The authors start by characterising the ability of UPEC to infect prostate cells using their 2D organoid-based model. All of the data presented is performed using the well-characterised UTI89 isolate, which was cultured from the urine of a woman with cystitis. I think an opportunity is missed here. It would be relevant to expand the analysis presented in Fig 2b to show how UPEC prostatitis isolates colonise

the 2D organoid-based model over a 24h time course. Given the organoid model is established these experiments should not be difficult.

We thank the Reviewer for this great suggestion, which was also brought up by Reviewer #1. We initially chose the UPEC reference strain UTI89 because bacterial prostatitis strains are often UPEC strains, and UTI89 is the most commonly used strain in the literature. This made it easier to compare our results with previous (and future) studies. However, we thank the Reviewer (and Reviewer #1, see Point #2 above) for this valuable suggestion. We fully agree that using prostatitis isolates strengthens our findings. Therefore, we have validated all major results, including the one from Figure 2b, using three prostatitis isolates as suggested by the Reviewer. Please see above (Point #2, Reviewer #1) for a complete summary of the data generated and incorporated into the manuscript.

Concerning **Figure 2b** (now **2c**), we have performed a CFU analysis over 24hpi with all three prostatitis isolates (CP1, P10 and P16), and analysed by fluorescence microscopy the intracellular bacterial replication (**Extended Data Figure 10a-f**). As observed for the strain UTI89, all prostatitis isolates were able to replicate within the prostate cells in the 2D model grown in the presence of DHT (DHT condition).

L192-193. Please define the criteria used to categorize tissues as 'healthy'.

We thank the Reviewer for bringing this inaccuracy to our attention. We have corrected the term "healthy" to "non-cancerous" (line 636) and have added this clarification in the Methods section, which now reads as follows (Line 634): "To stain and image the human prostate tissue, tissue explants were obtained from patients undergoing surgery (i.e. biopsy) at the University Hospital of Würzburg. Only tissues identified as non-cancerous by the pathologist were included in the analysis."

Fig 4d (L262-263). I have a concern with this figure. The blot has been cut and is presented as two separate lanes. It does not look like it is the same blot. An explanation and new figure are required. We thank the Reviewer for this comment and the opportunity to clarify our experimental approach. The blot was intentionally cut as a necessary step in the protocol. For the ligand capture assay, protein samples are separated by SDS-PAGE, transferred onto a PVDF membrane, blocked with 1% BSA/1% milk in TBS-T, and incubated with the recombinant FimH protein (rFimH-FLAG-tagged). Then, host proteins (transferred to the PVDF membrane) interacting with the rFimH are revealed by using an anti-FLAG antibody. Since D-mannose is known to bind FimH and block its interaction with other proteins (see **Figure 4a-c**), we used it here as a control, following a published protocol (Eto et al. 2007). To do so, we ran in duplicate a set of samples on the same gel, each set separated by a lane containing a protein ladder (see **Rebuttal Figure 3a**). After transfer, the membrane was cut over the protein ladder lane (**Rebuttal Figure 3a**, red start) to separate the two sets of samples. One set was incubated with rFimH alone (left membrane), and the other with rFimH plus 2.5% D-mannose (right membrane). Because D-mannose could also act as a partial blocking agent, the background signal on that membrane was lower compared to the one probed with rFimH alone. This has also been reported by others (see Figure 1 in Eto et al., 2007, where the same approach was used to identify FimH partners in bladder epithelial cells). To compensate for the reduced signal and ensure no binding was missed, we used a longer exposure for the D-mannose blot (see **Rebuttal Figure 3a**, right panel). We have also included images from a second independent replicate, which shows similar results (**Rebuttal Figure 3b**). Uncropped images of immunoblots are included as Source Data.

As the reviewer may notice, some lanes include samples from other models (bladder organoids, 5637 and THP-1 cell lines). While interesting, these are part of a separate ongoing project and are not shown in the main figure, as the protein bands have not been identified yet for the other samples. We hope this explanation fully addresses the Reviewer's concern.

Rebuttal Figure 3. Ligand capture assay. *a. Biological replicate no. 1. Black rectangles mark the lanes shown in the main figure of the manuscript. The red star shows the lane (protein ladder) where the membrane was cut. b. Biological replicate no. 2. rFimH_{LD} was run in the last lane as a control for the western blot using the anti-FLAG antibody.*

L269-287. I did not follow how the binding between rFimHLD and rPPAP was quantified in Fig 4g. What is meant by ‘fold change over HA signal’ as the value for the y-axis to Fig 4g.

We apologise for the confusion. **Figure 4g** shows the quantification of rFimH-FLAG binding to different PPAP-HA mutants. Recombinant PPAP proteins tagged with HA were immobilised on anti-HA agarose beads, then incubated with rFimH-FLAG and eluted. The amount of rFimH-FLAG bound was assessed by western blot (representative blot is shown in **Figure 4f**). To account for potential variation in the amount of immobilised PPAP-HA (the bait) as a consequence of the mutations, we normalised the rFimH-FLAG signal to the HA signal (PPAP-HA) for each mutant or WT. Thus, the y-axis represents the FimH-FLAG signal normalised to the PPAP-HA signal, which we originally referred to as “fold change over HA signal.” To clarify this, we have now corrected the label to “normalised to rPPAP-HA signal.” Moreover, we have added a short paragraph in the Methods section to explain how this quantification was performed (line 762: “For quantification, band intensities from Western blots were measured using ImageJ. The rFimH_{LD}-FLAG signal (FLAG blot) was normalised to the rPPAP-HA (HA blot) for each WT or mutant sample to account for potential differences in bait loading to the beads”).

Fig 5a-b. Invasion is measured at 1 hpi. This experiment could be performed as a time course, showing cfu's over 24h post infection.

We thank the Reviewer for this suggestion. Since the primary function of the FimH adhesin is to mediate bacterial attachment and facilitate invasion, and because we hypothesised that PPAP serves as the host receptor involved in this process, we initially focused on the 1 hpi time point to assess its potential role in invasion. However, we agree that analysing later time points could provide insights into whether the lack of PPAP has consequences at later time points of the infection.

Given that we expanded the analysis to include three additional prostatitis isolates, we opted to measure CFUs at 24 hpi, which we considered the most informative late time point beyond 1 hpi. These data are now included in **Extended Data Figure 17**. As expected, we observed a marked reduction in CFUs at 24 hpi for all strains (UTI89, CP1, P10, and P16) in *Ppap* knockout (KO) cells compared to WT controls, most likely due to the decrease in invasion (**Figure 5, Extended Data Figure 17**). Interestingly, bacteria that managed to invade *Ppap* KO cells did not replicate to the same extent as in WT cells (**Figure 5c-d, Extended Data Figure 17b**). This suggests that PPAP may either influence bacterial replication, or more likely, that in the absence of PPAP, bacteria invade less permissive cell types (e.g. basal cells) via alternative receptors, where intracellular replication does not occur.

These new results have been incorporated into the manuscript as **Figure 5bc-d** and **Extended Data Figure 17** and described in the text (line 485: "A decrease in bacterial intracellular replication was also observed in the *Ppap* KO cells, which could not be explained by reduced invasion alone. This suggests that PPAP may either influence bacterial replication or, more likely, that in the absence of PPAP, bacteria invade less permissive cell types (e.g. basal cells) via alternative receptors, where intracellular replication may not occur.").

Fig 6a. To demonstrate the images are representative of the overall data, the imaging analysis should be supported with quantitative data to show the number of adhered UPEC for the WT, *fimH* and WT+D-man assays across multiple fields.

We thank the Reviewer for this helpful suggestion. We agree that showing quantification of all immunofluorescence images strengthens our findings. As also suggested by Reviewer #1 (see Point #3 above), we have now done this not only for Figure 6a, but for all images shown in the manuscript, except for the rFimH_{LD} staining, where quantification at the single-cell level is challenging.

Concerning Figure 6a, we have quantified all the images and presented the data in two types of graphs: one showing the percentage of infected cells per image (or field of view, shown only for WT, as very few bacteria were observed in the *fimH* mutant or WT + D-mannose; **Figure 6b**), and another showing the absolute number of bacteria per image (**Extended Data Figure 18c**), where the low bacterial counts in the *fimH* mutant and WT + D-mannose conditions are evident. These data have been added to the manuscript as **Figure 6b** and **Extended Data Figure 18c**. The same approach was used to quantify colocalization of PPAP⁺ cells with the prostatitis isolate CP1 (**Extended Data Figure 18a–b**).

We hope the Reviewer agrees that the quantification supports our conclusion that bacteria (UTI89 or CP1) are found mainly in PPAP⁺ cells. This is dependent on FimH, as the effect is lost in the presence of D-mannose.

References

- Clermont, O, S Bonacorsi, and E Bingen. 2000. "Rapid and Simple Determination of the Escherichia Coli Phylogenetic Group." *Applied and Environmental Microbiology* 66 (10): 4555–58. <https://doi.org/10.1128/AEM.66.10.4555-4558.2000>.
- Eto, Danelle S, Tiffani A Jones, Jamie L Sundsbak, and Matthew A Mulvey. 2007. "Integrin-Mediated Host Cell Invasion by Type 1-Piliated Uropathogenic Escherichia Coli." *PLoS Pathogens* 3 (7): e100. <https://doi.org/10.1371/journal.ppat.0030100>.
- Hughes, Chris S, Lynne M Postovit, and Gilles A Lajoie. 2010. "Matrigel: A Complex Protein Mixture Required for Optimal Growth of Cell Culture." *Proteomics* 10 (9): 1886–90. <https://doi.org/10.1002/pmic.200900758>.
- Hung, Chia-Suei, Julie Bouckaert, Danielle Hung, Jerome Pinkner, Charlotte Widberg, Anthony DeFusco, C Gale Auguste, et al. 2002. "Structural Basis of Tropism of Escherichia Coli to the Bladder during Urinary Tract Infection." *Molecular Microbiology* 44 (4): 903–15. <https://doi.org/10.1046/j.1365-2958.2002.02915.x>.
- Karthaus, Wouter R, Matan Hofree, Danielle Choi, Eliot L Linton, Mesruh Turkekul, Alborz Bejnood, Brett Carver, et al. 2020. "Regenerative Potential of Prostate Luminal Cells Revealed by Single-Cell Analysis." *Science* 368 (6490): 497–505. <https://doi.org/10.1126/science.aay0267>.
- Krieger, John N, Ulrich Dobrindt, Donald E Riley, and Eric Oswald. 2011. "Acute Escherichia Coli Prostatitis in Previously Health Young Men: Bacterial Virulence Factors, Antimicrobial Resistance, and Clinical Outcomes." *Urology* 77 (6): 1420–25. <https://doi.org/10.1016/j.urology.2010.12.059>.
- Krieger, John N, and Praveen Thumbikat. 2016. "Bacterial Prostatitis: Bacterial Virulence, Clinical Outcomes, and New Directions." *Microbiology Spectrum* 4 (1). <https://doi.org/10.1128/microbiolspec.UTI-0004-2012>.
- Lipsky, Benjamin A, Ivor Byren, and Christopher T Hoey. 2010. "Treatment of Bacterial Prostatitis." *Clinical Infectious Diseases* 50 (12): 1641–52. <https://doi.org/10.1086/652861>.
- Morales-Espinosa, Rosario, Rigoberto Hernandez-Castro, Gabriela Delgado, Jose Luis Mendez, Armando Navarro, Angel Manjarrez, and Alejandro Cravioto. 2016. "UPEC Strain Characterization Isolated from Mexican Patients with Recurrent Urinary Infections." *Journal of Infection in Developing Countries* 10 (4): 317–28. <https://doi.org/10.3855/jidc.6652>.
- Rudick, Charles N, Ruth E Berry, James R Johnson, Brian Johnston, David J Klumpp, Anthony J Schaeffer, and Praveen Thumbikat. 2011. "Uropathogenic Escherichia Coli Induces Chronic Pelvic Pain." *Infection and Immunity* 79 (2): 628–35. <https://doi.org/10.1128/IAI.00910-10>.
- Ruiz, Joaquim, Karine Simon, Juan P Horcajada, Maria Velasco, Margarita Barranco, Gloria Roig, Antonio Moreno-Martínez, et al. 2002. "Differences in Virulence Factors among Clinical Isolates of Escherichia Coli Causing Cystitis and Pyelonephritis in Women and Prostatitis in Men." *Journal of Clinical Microbiology* 40 (12): 4445–49. <https://doi.org/10.1128/JCM.40.12.4445-4449.2002>.
- Stsiapanava, Alena, Chenrui Xu, Shunsuke Nishio, Ling Han, Nao Yamakawa, Marta Carroni, Kathryn Tunyasuvunakool, et al. 2022. "Structure of the Decoy Module of Human Glycoprotein 2 and Uromodulin and Its Interaction with Bacterial Adhesin FimH." *Nature Structural & Molecular Biology* 29 (3): 190–93. <https://doi.org/10.1038/s41594-022-00729-3>.
- Terai, A, S Yamamoto, K Mitsumori, Y Okada, H Kurazono, Y Takeda, and O Yoshida. 1997. "Escherichia Coli Virulence Factors and Serotypes in Acute Bacterial Prostatitis." *International Journal of Urology* 4 (3): 289–94. <https://doi.org/10.1111/j.1442-2042.1997.tb00192.x>.

Response to concern in Extended Data Figure 17a:

Comment: Overlapping area between the microscopy images for the P10 infection and P16 infection shown in Extended Data Fig 17. There is an image section that overlaps between the image of P10 and P16 (the top section of the image of P10 is the same as a bottom section of the P16). These are supposed to be 2 independent wells infected with different bacterial strains (P10 and P16).

We sincerely apologize for the error and are very grateful to the editorial team for bringing it to our attention. After reviewing the raw data, we suspect that the issue arose from mislabelling of the P16 image during acquisition at the confocal microscope (LSCM). The image labelled as P16 was likely in fact from the well infected with P10. This was an honest mistake, but nonetheless unacceptable. We greatly appreciate the opportunity to correct it.

Since we could not confirm the mislabeling with absolute certainty, we re-imaged all three original biological replicates of the experiment. In addition, we performed an additional biological replicate after the issue was brought to our attention.

All re-taken images are presented in **Source Data for Reviewer 1–3** (corresponding to replicates 1–3). Images from replicate 4 are presented in **Source Data for Reviewer 4** (See below). Raw data (Leica files and TIFFs) have been provided via GigaMove (link: <https://gigamove.rwth-aachen.de/en/download/c36d6dbc02883db81626638fbf7e1a28>).

We have updated data in **Extended Data Figure 17a** using the new images taken from the biological replicate 3. The figure legend has also been revised to include the number of biological replicates ($n = 4$, shown in green, see below). No changes were required in the main manuscript, as the results remain the same, and the experimental conditions (methods section) were unchanged.

All biological replicates were performed under the same experimental conditions. Experimental dates and other experimental details are shown in the table below. As requested, we provide below the technical details regarding the experimental setup:

Cell seeding and in vitro infection: Seven-day-old 3D mouse prostate organoids were seeded onto ibidi μ -Slide 18 Well (81816, Ibidi). A total of 6,250 cells were seeded per well (half the number used for 48wp, corresponding to half the surface area of an ibidi well). Passage numbers of the 3D organoids for each replicate are listed in the table below. Organoid cells were cultured for 7 days in the presence of 10 nM DHT, with medium changes on days 4 and 7 (before infection). On day 7, prior to infection, one well per condition (WT and KO) was trypsinised and counted to calculate MOI. Cells were then incubated with *E. coli* strains (CP1, P10, and P16; see below details for bacteria preparation) at an MOI of 100 for 1 h. After that hour (considered time point 0 h), cells were washed and the medium was replaced with fresh medium containing 50 μ g/ml gentamicin for 30 min, followed by medium containing 10 μ g/ml gentamicin for another 30 min. Wells were then washed three times with PBS and fixed with 4% paraformaldehyde for 15 min at room temperature, permeabilized with 0.5% Triton X-100, and nuclei were counterstained with Hoechst 33342 (1:5,000; H3570, Life Technologies).

Bacteria preparation for infection: For each infection, bacteria were streaked from -80 °C cryostocks onto LB agar plates with ampicillin (P10, P16) or chloramphenicol (CP1) and incubated overnight at 37 °C. The following day, a single colony was picked and grown in 2 ml of LB with ampicillin or chloramphenicol under static conditions at 37 °C. After 15–18 h, a fresh culture was started from the overnight culture at an initial OD600 of 0.05 in LB with ampicillin (P10 and P16) or chloramphenicol (CP1) and incubated statically at 37 °C for 24 h. In all cases, OD600 measurement after 24 h was between 1.7–2.0. OD600 1 was harvested by centrifugation and resuspended in organoid medium. MOI100 was then calculated, and the required number of bacteria was added to each well.

Imaging: Three images per well were acquired using a Leica Stellaris 5 confocal microscope with Leica Application Suite Advance Fluorescence (LAS AF Leica Microsystems, v.2.7.3.9723). For each image, a

z-stack was taken with a step size of 0.5 μm . Maximal projection images were generated from the stacks, and TIFFs were exported from LAS AF Lite.

No. Biological Replicate	Date Cell Seeding	Passage Organoid line (WT and KO)	Date infection	OD 24h bacterial culture			MOI	Time point (hour post infection, hpi)	Date Imaged
				CP1	P10	P16			
1	21.03.2025	9	28.03.2025	2	1.7	2	100	1hpi	23.08.2025
2	25.03.2025	11	01.04.2025	1.9	1.7	1.9	100	1hpi	23.08.2025
3	01.04.2025	12	08.04.2025	1.8	1.7	1.8	100	1hpi	23.08.2025
4	20.08.2025	30	27.08.2025	1.7	1.8	1.8	100	1hpi	27.08.2025/18.09.2025

Table 1. Experimental conditions for the biological replicates 1-4 corresponding to Extended Data Figure 17a.

We are grateful to the reviewer and the editorial team for their careful evaluation of this matter and their understanding, and we look forward to their feedback.

Extended Data Figure 17

Extended Data Figure 17. PPAP is necessary for maximal invasion of *E. coli* CP1, P10 and P16 into prostate cells. a. Representative confocal microscopy images of *Ppap* KO mouse organoid cells (*Ppap* KO M#1, grown in 10 nM DHT) infected with CP1, P10 or P16 (red) for 1h. Nuclei were counterstained using Hoechst 33342 (blue). Scale bar: 25 μ m. $n = 4$. **b-d.** CFU quantification of two *Ppap* KO mouse organoid lines infected with CP1 (**b**), P10 (**c**) or P16 (**d**) for 1h and 24h. $n = 6$.

Source Data for Reviewer 1

Biological replicate 1

E. coli Nuclei

Source Data for Reviewer 1. Raw images from the biological replicate #1 of *Ppap* KO and WT mouse organoid cells (*Ppap* KO M#1, grown in 10 nM DHT) infected with CP1, P10 or P16 (red) for 1h. Three images were acquired per well. Nuclei were counterstained using Hoechst 33342 (blue). Scale bar: 25 μ m.

Source Data for Reviewer 2

Biological replicate 2

Source Data for Reviewer 2. Raw images from the biological replicate #2 of *Ppap* KO and WT mouse organoid cells (*Ppap* KO M#1, grown in 10 nM DHT) infected with CP1, P10 or P16 (red) for 1h. Three images were acquired per well. Nuclei were counterstained using Hoechst 33342 (blue). Scale bar: 25 μ m.

Source Data for Reviewer 3

Biological replicate 3

Source Data for Reviewer 3. Raw images from the biological replicate #3 of *Ppap* KO and WT mouse organoid cells (*Ppap* KO M#1, grown in 10 nM DHT) infected with CP1, P10 or P16 (red) for 1h. Three images were acquired per well. Nuclei were counterstained using Hoechst 33342 (blue). Scale bar: 25 μ m. This experiment was used as representative for the Extended Data Figure 17a.

Source Data for Reviewer 4

Biological replicate 4

E. coli Nuclei

Source Data for Reviewer 4. Raw images from the biological replicate #4 of *Ppap* KO and WT mouse organoid cells (*Ppap* KO M#1, grown in 10 nM DHT) infected with CP1, P10 or P16 (red) for 1h. Three images were acquired per well, except for the condition WT infected with CP1, where three additional images were acquired after observing incomplete z-stack for image 1 and 2 (see raw data folder; total number of images = 6). Nuclei were counterstained using Hoechst 33342 (blue). Scale bar: 25 μ m.